# TOWARDS THE CHARACTERIZATION OF REPRESENTATIONS LEARNED VIA CAPSULE-BASED NETWORK ARCHITECTURES

**Paper# 2884 (Anonymous authors)**

## ABSTRACT

Capsule Networks (CapsNets) have been re-introduced as a more compact and interpretable alternative to standard deep neural networks. While recent efforts have proved their compression capabilities, to date, their interpretability properties have not been fully assessed. Here, we conduct a systematic and principled study towards assessing the interpretability of these types of networks. We pay special attention towards analyzing the level to which *part-whole* relationships are encoded within the learned representation. Our analysis in the MNIST, SVHN, PASCAL-part and CelebA datasets on several capsule-based architectures suggest that the representations encoded in CapsNets might not be as disentangled nor strictly related to *parts-whole* relationships as is commonly stated in the literature.

## 1 INTRODUCTION

Capsule Networks (CapsNets) Sabour et al. (2017) were recently re-introduced as a more compact and interpretable alternative to deep neural networks. This motivated their introduction in different critical applications including healthcare Deepika et al. (2022); Afriyie et al. (2022), object detection Lin et al. (2022); Yu et al. (2021), hyperspectral imaging Deng et al. (2018); Wang et al. (2018), autonomous driving Heylen et al. (2018); Grigorescu et al. (2020); Dulian & Murray (2021), and finance Sezer et al. (2020). Different from standard convolutional neural networks (CNNs), which arrange neurons in a predefined 2-dimensional manner, a capsule is a group of neurons whose activity vector represents the instantiation parameters of a specific type of entity such as an object or an object part. Then, by explicitly providing a mechanism to link specific capsules at neighboring layers, *part-whole* relationships are modelled. Based on these mechanisms ensembles of capsules are capable of modelling higher level semantic concepts, e.g. objects and scenes, with a proper spatial arrangement. Recent efforts Mukhometzianov & Carrillo (2018); Jung et al. (2020) have shown that CapsNets are capable of obtaining comparable results with respect to their convolutional counterparts while requiring less parameters. Thus, confirming their compression capabilities. However, while it is theoretically expected that the activities of the neurons within an active capsule represent relevant properties of the data (e.g. deformation, velocity, hue, texture, etc.), to date, this built-in interpretable characteristic has not been systematically assessed.

Efforts towards improving the intelligibility of systems based on deep models have been oriented in two fronts. Either, by providing insights into what a model has actually learned (model interpretation), or by providing justifications for the predictions made by these models (model explanation). In the last decade, a significant amount of work Fong & Vedaldi (2017); Grün et al. (2016); Hendricks et al. (2016); Zeiler & Fergus (2014) has been conducted towards explanation. In contrast, the amount of efforts Bau et al. (2017); Oramas et al. (2019) around the interpretation task is much more reduced. A more critical trend has been observed in the context of CapsNets, where efforts toward their interpretation are almost non-existent. Putting the previous points together paints a worrying picture. CapsNets are a type of model that is proving effective and is gaining attention in the context of several critical applications. However, when compared with its CNN-based counterparts, this type of model is not well studied and their inner-working are not necessarily well understood.

Starting from these observations, we aim to experimentally assess the interpretability capabilities of CapsNets. More specifically, we verify whether the internal representations of CapsNets do encode features that are both relevant for the data they were trained on, and critical for the performance on the

task at hand, e.g. classification. This paper puts forward the following contributions: i) We propose a principled methodology for assessing the interpretation capabilities of capsule networks. This aims at a complete understanding by looking at the inner workings of the CapsNet across all its layers. ii) We conduct an empirical analysis to assess the level to which CapsNet-based representations do encode *part-whole* relationships. To the best of our knowledge, this is one of the first efforts conducting this type of study. iii) As part of our methodology, we propose two methods for the extraction of relevant units in CapsNets. These units lead to comparable performance.

## 2 RELATED WORK

Understanding how deep complex models operate internally have received significant attention in recent years Zeiler & Fergus (2014); Simonyan et al. (2014); Grün et al. (2016); Selvaraju et al. (2017). Most of the methods in the literature operate in a post-hoc manner, i.e. they provide interpretation/explanation capabilities on a pre-trained model. More recently a new trend has emerged which focuses on the design of methods or learning algorithms that aim to produce models that are interpretable. Thus producing models that are interpretable-by-design Zhang et al. (2018).

Related to this trend, Sabour et al. (2017) proposed CapsNets. A characteristic aspect of this model is that the learned representations encode *part-whole* relationships from features present in the data. Li et al. (2019) analyzed their proposed CapsNet-based recommendation model (CARP) which verifies whether it was able to detect suitable reasons and their effects. This was done by retrieving the top-$k$ phrases which are used to interpret rating behavior. In a similar manner, Wang et al. (2020) proposed a multi-head attention layer with capsules that capture the semantic aspects which resemble the global interpretation with respect to the entire dataset. Moreover, they explain the output of the network for a given input by referring to the primary capsule that strongly agrees with the class capsule.

In the medical context, Shen & Gao (2018) proposed a modified CapsNet model for automatic thoracic disease detection. Grad-CAM Selvaraju et al. (2017) visualizations were used to inspect regions of interest that are considered critical for assessing the location of the disease. In the domain of hyperspectral imaging, Shi et al. (2021) proposed a two stage model for vegetation recognition. The interpretability of their model was improved by using a capsule-based stage for learning a hierarchical representation from inputs. This capsule-based stage was composed of enriched low-level feature representations computed during the first stage. The interpretation capabilities of this model were assessed by measuring the correlation between intermediate features in the model and annotations in auxiliary datasets. de Jesus et al. (2018) proposed a CapsNet-based method for protein classification and prediction. Outputs produced by this model are explained by following an input-modification method in which information about some "atoms" of a given input is modified and its effect on the output is measured. Shahroudnejad et al. (2018) analyzed the behavior of CapsNets (Sabour et al. (2017)) to identify the relevant activation path defined by the bi-product of the routing procedure and use it as an explanation for the network. Jung et al. (2020) proposed the interpretable iCaps model which produces explanations of classification predictions based on relevant information in active capsules. This is achieved by an additional supervised approach for representation disentanglement and a regularizer that ensures low redundancy on the concepts encoded in the model. Previously Bhullar (2020); Shahroudnejad et al. (2018) introduced interpretation methods to verify the path based on the dynamic routing procedure introduced in Sabour et al. (2017). More recently, new efforts emerged, questioning the hierarchical relationships encoded in CapsNets. Along this line, Mitterreiter et al. (2023) argues that CapsNets do not exhibit any theoretical properties suggesting the emerging of parse trees in their encoded representation. They discussed how a parse tree structure is crucial when capsules are expected to serve as nodes and their connections function as edges.

Inspired by Bhullar (2020); Shahroudnejad et al. (2018), we follow a similar approach in our experiments to obtain and interpret the relevant connections between layers. In line with Wang et al. (2020) we consider local and global interpretation from models trained on visual data. Different from it, we analyze the internal behavior including the convolution layer, and we also analyzed forward and backward path estimation alternatives. Similar to Shen & Gao (2018) we also explain model predictions, visually, by producing heatmaps. Different from iCaps Jung et al. (2020) and all previous efforts, we analyze the internal behavior of the CapsNet architectures. We analyzed the drop in performance when relevant filters and capsules are removed in such a network. Finally, similar to Mitterreiter et al. (2023), we introduce a systematic study towards assessing the interpretability of

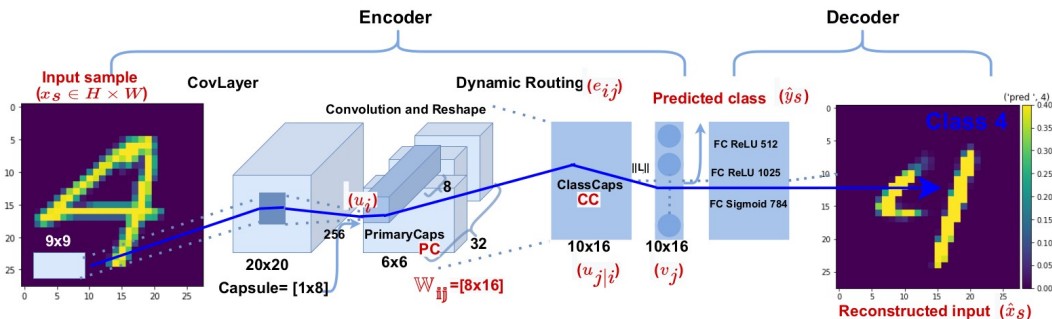

Figure 1: Considered capsule network architecture by Sabour et al. (2017).

several Capsule Architectures. However, we propose a more general protocol that extends beyond examining the connections between various levels of capsule layers. Our protocol consists of perturbation analysis and extracts the relevant features that define the relevant paths connecting the inputs and outputs. We also assess the emergence of the *part-whole* relationship

## 3 OVERVIEW: CAPSULE NETWORKS

Fig. 1 presents an overview of the CapsNet architecture proposed by Sabour et al. (2017). In a CapsNet architecture, each capsule is defined by a set of neurons (represented by the activity vectors denoted by $u_i \in \mathbb{R}^{d^l \times d^{l+1}}$, where $d^l$ and $d^{l+l}$ are the dimensions of the primary and class capsule layers respectively) along with instantiation parameters $T$. These parameters represent the features of the entities in the desired dataset such as pose (position, size, orientation), lighting, deformation, and velocity. The length of the output vector $v_j$ indicates the likelihood of occurrence of an entity. A standard CapsNet architecture (as proposed in Sabour et al. (2017)) consists of two parts (Fig. 1). First; an *encoder* which is composed of three layers; a convolutional layer (*Conv*), a primary capsule layer (*PC*), and a class capsule layer (*CC*). Second, a *decoder* which aims a reconstructing the input.

The *encoder* part starts with a standard *Conv* layer which aims to obtain the initial features (e.g., edges and lines) from an input image $x_s \in \mathbb{R}^{H \times W}$. The *Conv* layer is followed by a *ReLU* activation function. The *PC* layer is represented by 32 *PC* capsules which aim to detect high-level features (e.g., part of an object) based on features extracted by the *Conv* layer. *PCs* perform 8 convolution operations each time. Each $PC_i$ in the *PC* layer ($i \in [1 \le i \le N]$, with $N$ the number of *PC* capsules) encodes $T$ by vector $u_i$. Consequently, $CC^j$ in the *CC* layer ($j \in [1 \le j \le M]$, with $M$ being the number of discrete classes to be predicted (e.g., $M=10$ in case of the MNIST digits). The *CC* layer receives an input from the *PC* layer whose size depends on the dataset being used (e.g., $6 \times 6 \times 8 \times 32$ in case of the MNIST digits). Then, an affine transformation is applied via the transformation matrix $\mathbb{W}_{ij} \in \mathbb{R}^{d^l \times d^{l+1}}$. This type of transformation is a unique procedure to CapsNet since it considers both the missing spatial relationships and the relationships between parts in relation to the object. A product between the transformation matrix $W_{ij}$ and capsules in the *PC* layer is computed. This transforms $u_i$ to a vote/predicted vector $\hat{u}_{j|i} \in \mathbb{R}^{d^{l+1}}$ towards the $j^{th}$ capsule. The predicted vector $\hat{u}_{j|i}$ indicates how the $i^{th}$ capsule from the *PC* layer contributes to the $j^{th}$ capsule in the *CC* layer. Coupling coefficients $e_{ij} \in \mathbb{R}^{d^l \times p^{d+1}}$ are added, and further computed, to indicate the forward path/flow (links) from $PC_i$ capsule at $l$ to the $CC_j$ capsule at $l+1$. $e_{ij} \in [0,1]; i \in \{1,2,...,n\}$. In the literature, this procedure is usually referred to as *dynamic routing* Sabour et al. (2017). In CapsNets, *CC* is the classification layer, where each capsule vector $v_j$ indicates a single class. The predicted class $\hat{y}_s$ is obtained by looking at the highest predicted probability $v_j \in \mathbb{R}^{d^{l+1}}$ (see Fig. 1). The *decoder* consists of several fully connected layers. It aims to reconstruct the input example $x_s$ from the vector $v_j$ that is produced by the CC layer. The decoder output is the reconstruction $\hat{x}_s$.

## 4 METHODOLOGY

The proposed methodology for assessing the interpretability of the representation encoded in a CapsNet consists of two parts. First, a hollistic perturbation analysis in which various parts of the architecture are ablated to assess their impact on classification performance. Second, during the

forward pass, the activation paths linking input and output are analyzed with the goal of verifying the most relevant features in each layer and their influence on modelling *part-whole* relationships.

## 4.1 PERTURBATION ANALYSIS

Given a trained CapsNet model $F$ that takes an input $x$ and produces a class label $\hat{y}$ as output, i.e a classifier. The first step is to push every example $x_{s,c}$ belonging to class $c$ and extract the activations $a_{s,c}^l \in \mathbb{R}^{w \times h \times d}$ for every internal layer $l$. Then, after computing the activations $a_{s,c}^l$ of every example $s$ in the dataset, flattening them and concatenating them on top of each other, a matrix $A_c^l = [a_{1,c}^l; a_{2,c}^l; ...; a_{s,c}^l]$ is defined with $A_c^l \in \mathbb{R}^{[S\prime \times A\prime]}$ where $S\prime$ refers to number of training examples corresponding to a class $c$ and $A\prime$ refers to total number of (1D) flattened activations.

With $A_c^l$ in place, first-order statistics $\eta_c^l = [min(A_c^l); max(A_c^l); mean(A_c^l); std(A_c^l)] \in \mathbb{R}^{[4 \times A\prime]}$ are computed in a column-wise manner. To complement this, a similar $\eta$ matrix is composed of the first-order statistics across the whole dataset $A_{all}^l = [A_1^l; A_2^l; ...; A_M^l] \in \mathbb{R}^{[D \times A\prime]}$, where $D$ refers to the number of examples in the dataset. From here, we define the interval $\alpha = [min(A_{all}^l), max(A_{all}^l)]$ which represents the empirical range of the activation space of a given unit at layer $l$ in the CapsNet. By estimating the range $\alpha$ in this way, we ensure the proper analysis of the activation space of different units located at different layers of the model. This differs significantly from previous efforts Sabour et al. (2017); Amer & Maul (2020); Shahroudnejad et al. (2018), which use heuristically-defined $\alpha$ values. We apply a perturbation analysis on $v_j$ (output of the *CC* layer) based on $\alpha$ for a given class $c$. We systematically replace each dimension of $v_j$ by an uniformly-sampled value $\xi$ in the range $\alpha$, which produces a new vector $v_j' \in \mathbb{R}^{d^{l+1}}$. Then, $v_j'$ is fed to the decoder to produce a reconstruction $\hat{x}_{s,c}$. In our analysis, the reconstructed input $\hat{x}_{s,c}$ has two purposes. It is used to conduct a systematic qualitative assessment on how variations in one of the dimensions of $v_j$ lead to different reconstructions. In addition, the effects of these variations can be quantitatively assessed by measuring the difference in classification performance that is obtained when $\hat{x}_{s,c}$ are fed to the CapsNets as inputs.

## 4.2 LAYER-WISE RELEVANT UNIT SELECTION

This method is aimed at detecting the relevant features/units that define activation paths in a given network. This is achieved by probing one layer at a time and verifying how the selection/suppression of internal units in such layers affects classification performance.

For *Conv* layers, global average pooling (GAP) Lin et al. (2014) is computed on the output channel produced by each filter. The filters with the highest GAP values, i.e. with the highest average activation, are assumed as the most relevant in the layer. Given this relevance rank, as defined by the GAP score of each filter, we select the top-$k$ most relevant filters via cross-validation. More precisely, given trained CapsNet and a validation set, we feed every example and gradually increase the number ($k$) of considered filters that are considered. During inference, the output/response of the selected filters is preserved while that of the rest is suppressed, i.e. set to zero. After repeating this procedure for several $k$ values, the lowest $k$ value which led to the highest classification accuracy is adopted. We consider these $k$ relevant filters, in the *Conv* layer, as the starting point of the activation path. For capsule-based layers, we apply a slightly different procedure. As mentioned earlier, CapsNets are characterized by the routing algorithms Mukhometzianov & Carrillo (2018) which determine how activations flow internally across capsule layers. Taking advantage of this, the "joint" capsules linking the *PC* and *CC* layers are identified. This is done based on the coupling coefficients $e_{ij}$ that determine the routes across capsule layers. Connecting the most relevant filters in the *Conv* layer with the "joint" capsules linking *PC* and *CC* produces a complete path between the input $x_s$ and the prediction $\hat{y}_s$. It is worth mentioning that during inference, some capsules from the *PC* layer that hold irrelevant features (for a given class) may appear to be active Bhullar (2020). However, following the routing algorithms, these capsules are eventually routed to inactive capsules in the *CC* layer. This prevents them from having a significant effect on the prediction $\hat{y}_s$. Similar as within the *Conv* layer, the number $k$ of selected units is estimated via cross-validation.

On the basis of the procedure described above, we propose two methods to define activation paths on CapsNets. These methods are characterized by the direction, i.e. *forward* or *backward*, that is followed to estimate the path. These directions are closely related to the awareness of the method on the class $\hat{y}_s$ predicted by the model.

**Class-Agnostic Forward Path Estimation.** During the forward pass, the routing procedure defines the path of the predicted vector $\hat{u}$ which represents the relationship between a child (part) $i$ and a possible parent (*whole*) $j$ capsule where we ignored the target classes. This is achieved by following the internal sequence of units (capsules) that maximize the internal activation flow-

Table 1: Mean classification accuracy of the original dense CapsNet and sparser versions based on the identified activation paths

| Model | MNIST | | SVHN | | PASCAL | | CelebA | |
|---|---|---|---|---|---|---|---|---|
| | valid | Test | Valid | Test | Valid | Test | Valid | Test |
| Dense (original) | 99.9 | 99.1 | 96.7 | 91.0 | 99.1 | 78.0 | 93.0 | 92.0 |
| Backward Path | 76.9 | 78.5 | 89.3 | 87.2 | 61.0 | 56.0 | 81.4 | 80.6 |
| Forward Path | 95.2 | 95.1 | 96.0 | 88.3 | 65.8 | 60.1 | 89.1 | 86.2 |

ing forward. In particular, we solve Eq. 1 by passing $x_s$ through the CapNet to obtain the corresponding coefficients $E^*$. By solving Eq. 1, the optimal $E^*$ indicates the agreement to the magnitude of $v_j$ is maximized. In other words, this optimization leads to identifying the relevant capsules that should be activated in the *PC* layer. Then, we find the relevant capsules of an example $x_s$ based on the coupling coefficients $e_{ij}$ with the highest values. The term $e_{iM}\hat{u}_{M|i}$ refers to coefficients that agree on what could be the predicted class for the input $x_s$.

**Class-Aware Backward Path Estimation.** In contrast to the previous method, which defines an activation

$$E^* = \sum_i^N (max(F(\hat{u}))) = \sum_i^N (max(\sum_i^N e_{iM}\hat{u}_{M|i})) \quad (1)$$

path in a class-agnostic fashion, here the coefficients $E^*$ are obtained based on the predicted class $\hat{y}_s$.

More specifically, each input example $x_s$ is fed to the CapsNet to produce a predicted class $\hat{y}_s$. Then, in a reverse manner, we extract the $e_{ij}$ of the prediction of

$$E^* = max(f(\hat{u})_{|\hat{y}_s}) = \sum_i^N (max(\sum_i^N e_{i\hat{y}}\hat{u}_{\hat{y}|i})_{|\hat{y}_s}) \quad (2)$$

class $\hat{y}_s$. Another difference with the previous method, which included all class coefficients, is that this method ignores the $e_{ij}$ that might be relevant to other classes.

In practice, this is a achieved by solving Eq. 2 given $\hat{y}_s$ where the term $(e_{i\hat{y}}\hat{u}_{\hat{y}|i})_{|\hat{y}_s}$ indicates agreement with respect to $\hat{y}_s$.

## 5 EXPERIMENTAL SETUP

**Datasets:** Following the settings of well established efforts Sabour et al. (2017); Gu & Tresp (2020); Ren et al. (2019); Ning et al. (2020); Shahroudnejad et al. (2018); Jung et al. (2020) we validate our methodology on the MNIST and SVHN datasets. In addition, we include the more complex PASCAL-Part Chen et al. (2014), CelebA Liu et al. (2015), and CelebAMask-HQ Lee et al. (2020) (referred to as "CPS") datasets. MNIST is a grayscale dataset depicting hand-drawn

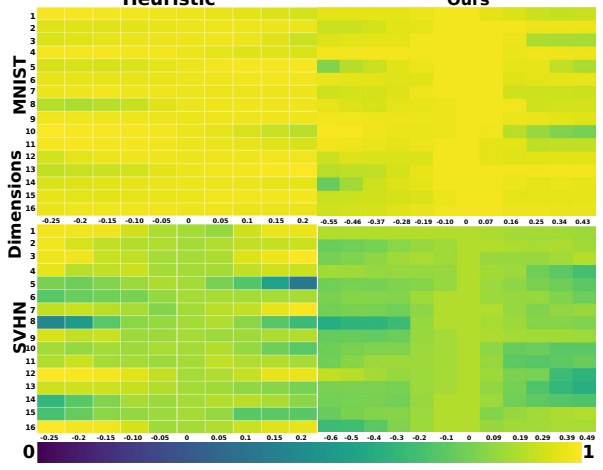

Figure 2: Mean classification Accuracy as perturbations are applied to the 16 dimensions of vector $v_j$.

digits. It is composed by 60k training images, from which 10k have been selected for validation, and 10k for testing. SVHN is an RGB dataset depicting number plates. It consists of 63k images for training, 10k for validation, and 26k for testing. The PASCAL-Part dataset is composed of 10k images covering 20 classes with detailed part-level annotations for each image. The dataset has been split into 1k images for both the validation and the test phases. CelebA is composed by 202k facial images consisting of two classes, *male* and *female*. For training the CapsNet, we follow the provided partitioning. CPS is a subset of CelebA with part-level annotations, 2842 test images were used to quantify the coverage of relevant parts.

**Implementation Details:** We consider the CapsNet architectures from Sabour et al. (2017) and Hinton et al. (2018). Both were trained on the MNIST and SVHN datasets for 50/32 epochs with a batch size of 32. The input size was set to 28×28 and 32×32 for the MNIST and SVHN datasets, respectively. On the PASCAL-part dataset, the models were trained for 50 epochs with a batch size of 14. In this case, the images were changed to grayscale. The input size was changed to 48×48. Similar to SVHN, CelebA and CPS were resized to an input size of 32×32. The CapsNet was trained for 7 epochs with a batch size of 32. In the training step, no data augmentation was applied, and batch size was determined empirically. Our experiments were implemented in PyTorch; code is publicly

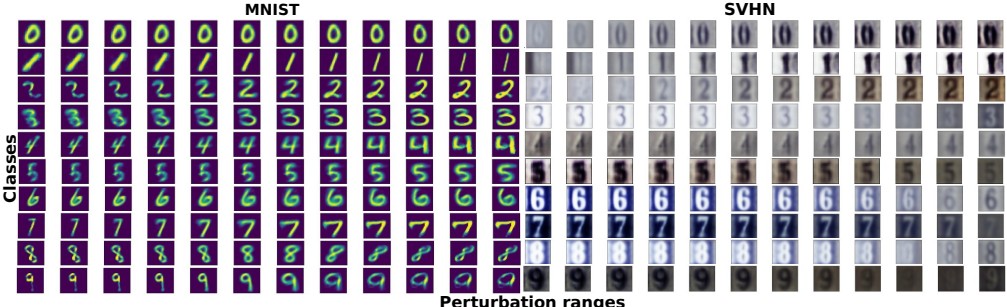

Figure 3: Examples of reconstructed inputs as the vector $v_j$ is perturbed. In some cases, multiple visual features are modified at the same time via the perturbation of a single dimension of the vector.

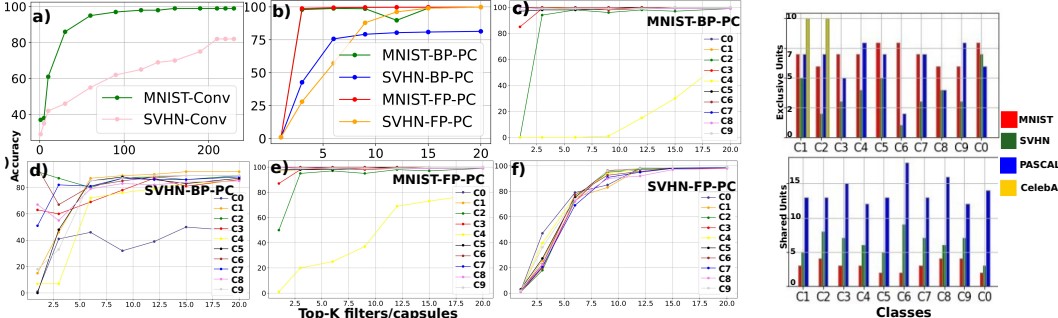

Figure 4: Changes in classification performance as the number of selected relevant units $k$ changes across *CapsNet* layers (left) where the relevant units $k$ increased gradually. Number of top-$k$ units exclusive for each class and shared w.r.t all the other classes (right)

available in the following link [1]. For brevity, we limit ourselves to report results on the original architecture from Sabour et al. (2017). Please refer to the supplementary material for results on the CapsNetEM architecture from Hinton et al. (2018).

## 6 EXPERIMENTS

**Classification.** Table 1 reports the mean classification accuracy of the trained models. In addition, we report the performance from sparser versions of the models based on the identified paths (Sec. 4.2). More specifically, when only the relevant activations were used to make the prediction. Worth noting is that while the performance using the dense model is on a par with that reported in previous works, the performance using the identified paths is lower. This is more critical for the backward path where it is much more reduced in comparison. As mentioned in Sabour et al. (2017), the backbone CapsNet did not perform well on complex datasets. This is also noticeable in the results from PASCAL.

### 6.1 PERTURBATION ANALYSIS

This analysis is conducted on the MNIST and SVHN datasets. Following Sec. 4.1, we estimate the empirical activation range $\alpha$ for the $v_j$ vector from the considered datasets. Then, a step size $\xi$ is computed (0.09 for MNIST, 0.1 for SVHN), in order to partition the $\alpha$ range into 12 perturbation possibilities per dimension; similar as in Sabour et al. (2017); Ning et al. (2020); Shahroudnejad et al. (2018). We replace the value of each dimension in $v_j$ producing the perturbed vector $v'_j$. This vector is pushed through the *decoder* in order to produce reconstructions $\hat{x}_s$ for each perturbed $v'_j$. These reconstructions are then pushed through the CapsNet and classification performance is computed.

**Results:** Fig. 3 presents some qualitative results in the form of reconstructions obtained by this procedure. A quick inspection to this figure reveals how the applied perturbations effectively provide some insights related to the features encoded by $v_j$. As can be noted, each dimension encodes various characteristics of the digits such as thickness, rotation, deformation, and scale. Moreover, the shape

---
[1]http://annonymized/github/URL

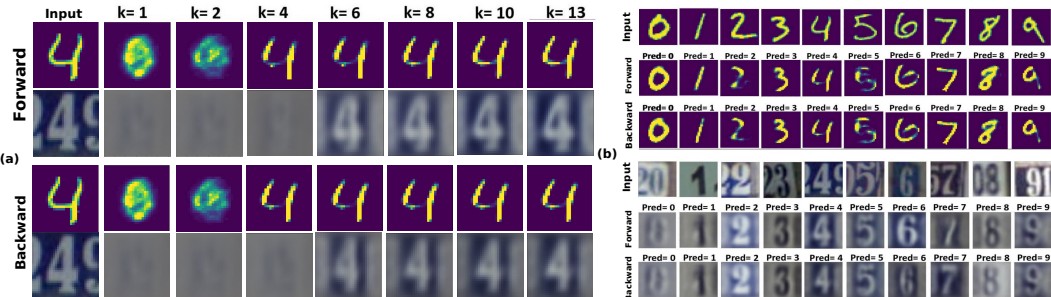

Figure 5: (a)Reconstructed inputs when only a small amount $k$ of units in the *PC* layer are considered. Note the difference on sharpness between the reconstructions for the SVHN dataset. (b) Reconstructed inputs when only relevant units of the network ($k$=35 *Conv* and $k$=10 *PC*) are considered.

(top and bottom regions in some digit classes) of $\hat{x}_s$ shows changes in different forms. It is clear that changes in specific elements of this feature space ($v_j$) are not exclusive to a single visual feature. As can be noted in Fig. 3 (MNIST digits 1-4), perturbations along a single dimension of $v_j$ introduce changes in both rotation and thickness of the generated digit. This suggests some level of feature entanglement. Moreover, we have observed that a given dimension in the representation space may encode different visual features for different classes. These observations are sufficient to conclude that CapsNets might be interpretable but the representations they encode are not disentangled.

To complement the qualitative analysis, Fig. 2 presents the classification performance that is obtained when each of the mentioned perturbations is applied. We show the output when the typical (heuristic) perturbation is applied (left), and the output of our perturbation analysis (right). As can be noted, overall performance is relatively high. However, our analysis indicates that the activation range to be considered should be wider than the usual one arbitrarily used in previous works Sabour et al. (2017); Shahroudnejad et al. (2018); Bhullar (2020). This observation further stresses that the range used for the analysis (via the $\alpha$ parameter in our case) should be properly estimated. Further experiments (Sec. E.1) show that CapsNetEM exhibit similar trends.

## 6.2 LAYER-WISE RELEVANT UNIT SELECTION

This experiment is also conducted on the MNIST and SVHN datasets. Following the methodology introduced earlier, we define the *forward (FP)* and *backward (BP)* activation paths based on the top-$k$ units on the CapsNet. The cross-validation procedure led to the selection of $k$=35 filters for the *Conv* layer and $k$=10 for the *PC* layer for both datasets. Due to the complexity of PASCAL, the number of internal capsules was increased. Therefore, the number of selected relevant units is set to $k$=1000. In Fig. 4(left) we report test classification performance as the number ($k$) of selected units is increased. We report results when only the *Conv* layer is modified (Fig. 4.a (left)), when only the *PC* layer is modified (Fig. 4.b (left)) and when both layers are modified at the same time, i.e. when the complete path is considered, for the MNIST (Fig. 4.c & e (left)) and SVHN (Fig. 4.d & f (left)) datasets.

**Convolutional Layer Ablation.** When only the *Conv* layer is modified (Fig. 4.a(left)) we notice that performance is quite acceptable for the case of MNIST. The story is different on SVHN, where there is a drop in performance when only considering the selected ($k$=35) units. This suggests a significant domain shift between the validation and test subsets of that dataset.

**Capsule Layer Ablation.** When only the *PC* layer is modified (Fig. 4.b(left)), we notice that performance is relatively high in most cases. For the selected units ($k$=10) we notice that the units selected via the *forward (FP)* path achieve higher performance than their *backward (BP)* counterparts on both datasets.

We looked for shared and exclusive units among the top-$k$ units selected for each class (Fig. 4(right)) by our layer-wise relevant unit selection method (Sec. 4.2). In MNIST, we noted that classes with similar appearance had a higher number of shared units. In particular, classes [1,7],

Table 2: Number (%) of shared top-$k$ units among two classes.

| Dataset | C1-C7 | C2-C8 | C3-C5 | C4-C8 | C6-C8 | C7-C9 | C0-C9 |
|---|---|---|---|---|---|---|---|
| MNIST | 4 (40) | 2 (20) | 3 (33) | 2 (20) | 1 (10) | 2 (20) | 1 (10) |
| SVHN | 4 (40) | 2 (20) | 5 (50) | 1 (10) | 4 (40) | 2 (20) | 2 (20) |

[3,5], [3,8], [8,9], with 4, 3, and 3 shared units, respectively (Table 2). In SVHN, on the quantitative side, we noted a higher number of shared units. On the qualitative side, the relation between class and the relevant unit was less pronounced, possibly due to the occurrence of parts of other digit instances

co-occurring in the input images. In this dataset, classes [0,2], [5,9], [6,8], [2,5], [3,4] had 6, 6, 5, and 5, shared units, respectively. For the PASCAL dataset, the classes dog-horse, person-tvmonitor, car-tvmonitor were the ones that had the highest number (5) of shared units. Worth mentioning is that it was observed across all the datasets, that the activation magnitude of the shared units was significantly low in comparison with the units exclusive to each class. For CelebA/CPS, all units were exclusive between class male and female. These are also observed on the qualitative results presented on Fig. 5(a). There it can be observed that, for both datasets, the quality of the reconstructed images stabilizes around the selection of $k$=10 units. Moreover, it is noticeable that, for the case of SVHN, reconstructions produced from units in the *forward* path are sharper that those from the *backward* path. These observations support the difference in performance across paths observed in Table 1.

**Network Ablation.** When the complete path is used, we notice that overall classification performance remains high. By inspecting the per-class performance, we observe that very few cases (digit-4 for MNIST and digit-0 for SVHN) suffer a significant drop when only considering the selected top-k units per layer. Fig. 5(b) shows qualitative reconstructions that were obtained with the selected k relevant units (filters/capsules). For MNIST, we notice that, while the digit sketches were not complete, the selected regions of the sketches seem to be sufficient to characterize the digit classes. For SVHN, we notice that background information, usually in the form of other digit regions, seem to be suppressed in the reconstructions. This suggests the selected units focus on the foreground objects, i.e. the digit of interest. These observations support our analysis on these relevant units for the search of possibly encoded parts. We anticipated similar behavior using CapsNetEM routing. However, we observed that the units do not seem to be class-specific (see Sec. E.2).

## 6.3 MEASURING *Part-Whole* RELATIONSHIP ENCODINGS

This experiment aims to measure the level to which features encoded in a CapsNet encode *part-whole* relationships. Towards this goal, inspired by methods proposed for CNNs Bau et al. (2017); Gonzalez-Garcia et al. (2018), we measure the spatial overlap between the internal responses of relevant capsules at layer $l$ (*Parts*) and the internal response at a layer $l+1$ (*Whole*). In our experiments, we measure this overlap between the *PC* ($h^l$) and *CC* ($h^{l+1}$) layers. The responses of relevant capsules were represented as heatmap $h$ Simonyan et al. (2014) computed by, first, estimating the prediction $\hat{y}_s$ produced by input $x_s$ when only the activations of a given unit are propagated forward during inference. Then, given the prediction $\hat{y}_s$ we compute the gradients of this prediction w.r.t. the input $x_s$.

The overlap between $h^l$ and $h^{l+1}$ is measured via the Relevance Mass Accuracy (*RMA*) metric Arras et al. (2022), which provides a value in the range [0,1] indicating the level of overlap. The *RMA* requires $h^{l+1}$ to be a binary matrix. To meet this requirement, we binarized the response $h^{l+1}$ from *CC*. Each $h^l$ was normalized in a per-unit basis by considering the min/max values from all the heatmaps related to that unit. We report results for various threshold values (0.1, 0.25 & 0.5). When measuring the overlap between $h^l$ and $h^{l+1}$, we focus our analysis on the top-$k$ ($k$=200) units selected following the *BP* (Sec. 6.2) extraction procedure. For PASCAL, from the 1000 units, a subset of the top 200 is considered to measure the level of overlap between the responses of *PC* and *CC*.

**Isolated Unit Analysis.** Table 3 (left) presents the mean *RMA* scores across all classes for different thresholds. For a given input example, overlap is computed for every response pair [$h^l$ (from *PC*), $h^{l+1}$ (from *CC*)] produced by each of the selected top-200 units, in isolation.

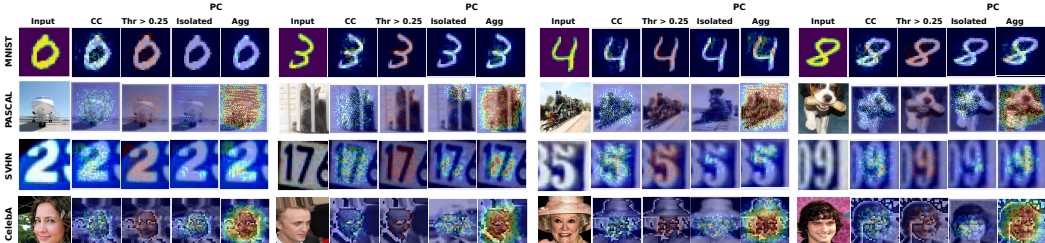

Figure 6: Qualitative examples of the responses considered for the computation of relevance mass accuracy (RMA) with 0.25 threshold (Thr) applied on *CC*. Heatmaps are generated based on top-$k$=1 relevant unit (isolated) and when top-$k$=200 (Aggregated).

Across datasets, we can notice that the *RMA* scores significantly drop as the threshold increases. This is to be expected since a higher threshold enforces a small region in $h^{l+1}$ (*CC*), which is harder to cover accurately. However, it is noticeable how, even for more lenient threshold values (0.1 and 0.25) the overlap scores remain relatively low. We notice that the scores related to PASCAL and SVHN seem to be the highest. Overall, the mean of *RMA* is lower than what was anticipated. This could be due to the inner-workings of the *RMA* metric. As we can see in Fig 6, the $h^{l+1}$ (*CC*) responses are sparse even when binarized with a relatively low threshold value (0.25). We notice a similar behavior on the $h^l$ responses from *PC*. For instance, as depicted in Fig 6 (4th column), for the class aeroplane from PASCAL, the $h^l$ (*PC*) response is not only sparse, but also covers a very small region. This affects the overlap score during the estimation. Moreover, the very focused nature of the *PC* responses, see Fig 6, suggests, first, that each of the relevant units model very specific details from the input data; and second, that compositions of all the $h^l$ responses might achieve higher coverage.

**Aggregated Unit Analysis.** Complementing the experiment above, we conduct a second analysis where the overlap is computed between aggregated $h^l$ responses from *PC* and the single $h^{l+1}$ response they produce at *CC*. This aggregation occurs at the pixel level by taking the maximum response per pixel location across the response maps produced by the considered top-200 relevant units.

A quick inspection of Fig. 6 *(Agg)* show two different trends. For the case of PASCAL and CelebA (Fig. 6, $2^{nd}$ & $4^{th}$ column), the effectiveness of the compositions *(Agg)* from different relevant units is evident. Here, the composition covers a significant region belonging to the objects of interest. On the contrary, for MNIST and SVHN (Fig. 6, $1^{st}$ & $3^{rd}$ column), considering additional units does not lead to higher coverage.

Table 3: RMA scores obtained by the top-200 relevant units when analyzed separately and aggregated over different thresholds.

| Dataset/Thr | Isolated | | | Aggregated | | |
|---|---|---|---|---|---|---|
| | 0.1 | 0.25 | 0.5 | 0.1 | 0.25 | 0.5 |
| MNIST | $31 \pm 6$ | $12 \pm 4$ | $3 \pm 1$ | $31 \pm 5$ | $12 \pm 3$ | $3 \pm 4$ |
| PASCAL | $45 \pm 5$ | $18 \pm 4$ | $3 \pm 1$ | $45 \pm 4$ | $16 \pm 3$ | $3 \pm 2$ |
| SVHN | $60 \pm 10$ | $26 \pm 8$ | $6 \pm 3$ | $55 \pm 10$ | $25 \pm 7$ | $7 \pm 2$ |
| CelebA | $44 \pm 7$ | $18 \pm 5$ | $4 \pm 2$ | $33 \pm 6$ | $12 \pm 3$ | $3 \pm 1$ |

These two observations might be pointing at the simplicity of such datasets and the inherent complexity of making predictions on them. More concretely, for more complex cases (PASCAL & CelebA) the units are steered towards learning complementary features; leading to larger coverage when aggregated. For the latter simpler, more constrained, case (MNIST & SVHN) a lower amount of features are needed. This results in more redundancy across the features encoded by the units, and might be the reason behind the reduced coverage from the aggregated responses.

Table 3 (right) shows the mean *RMA* scores related to this experiment. It is noticeable that for most of the cases even after aggregating a significant amount of relevant units into $h^l$, the *RMA* score remains low. Moreover, a significant drop is observed in the PASCAL and CelebA datasets. The reason for these low values may find its origin in the relationship between the aggregated response $h^l$ and the binarized response $h^{l+1}$ when considered in the RMA computation. As indicated earlier, for PASCAL and CelebA, a larger-coverage response $h^l$ is observed. This is clearly an over-estimation when compared to the binarized $h^{l+1}$ (*CC*) that is used as a reference. This is different in the case of MNIST and SVHN, where the compared responses are roughly the same. In our experiments with CapsNetEM routing (Sec. E.3) we noticed similar trends where the overlap is relatively low.

While the results presented in Table 3 suggest that CapsNets may not effectively encode *part-whole* relationships, it is worth noting that the observed low overlap may have its origin in other sources. More specifically, beyond an unsatisfactory overlap between the responses from *PC* and *CC*, the observed low RMA scores can also be attributed to the sparsity of the considered responses (see Fig. 6). As shown in Vandersmissen & Oramas (2023), RMA scores and other metrics for measuring overlap tend to favor smooth continuous heatmaps. In addition, we currently focused on the analysis of the top-200 relevant units due to the high computational costs required for a fine-grained analysis.

## 7 CONCLUSION

We propose a methodology to assess the interpretation properties of capsule networks. Our analysis is centered on the identification and ablation of relevant units in the network. Our results suggest that the representation encoded in CapsNets might not be as disentangled nor explictly related to *part-whole* features as is usually stated in the literature. Future work will concentrate in denser analysis of the architecture plus pinpointing the effect that the selection of k-top units has in the obtained insights. We hope the proposed methodology and discussed observations serve as a starting point for future efforts towards a deeper study and understanding of representations learned via CapsNets.

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

## A  SUPPLEMENTARY MATERIAL

## B  INTRODUCTION

This section consists of supplementary material. We introduce an extension of three sections where we provide extra details on the considered capsule network models and conducted experiments. First, we provide detailed information related to the layers of CapsNet architecture that are presented in the original manuscript. Afterwards, we introduce the algorithms that we followed in our methodology. Finally, we elaborate in more detail on our observed results.

## C  OVERVIEW: CAPSULE NETWORKS

**Dynamic Routing CapsNet:**  We introduced the first CapsNet architecture Sabour et al. (2017) in Sec. 3. This section provides more details of the model used in our methodology which consists of two parts, *encoder* and *decoder*. Table 4 presents details of the CapsNet proposed by Sabour et al. (2017). This table shows more details of the input and output shapes in both *encoder* and *decoder* parts.

Table 4: The input and the output shapes of the considered capsule network architecture (Dynamic Routing Sabour et al. (2017))

|  | *MNIST* | | *SVHN & CelebA* | | *PASCAL Parts* | |
|---|---|---|---|---|---|---|
| Layers | Input shape | Output shape | Input shape | Output shape | Input shape | Output shape |
| *Encoder:* | | | | | | |
| *Conv* $\Rightarrow$ relu | [1,28,28] | [20,20,256] | [3,32,32] | [24,24,256] | [1,48,48] | [32,32,256] |
| *PC* $\Rightarrow$ *CC* | [6,6,32] | [1152,8] | [6,6,32] | [2048,8] | [16,16,32] | [8192,8] |
| *CC* $\Rightarrow$ *decoder* | [1152,8] | [10,16] | [2048,8] | [10,16] | [8192,8] | [20, 16] |
| *Decoder:* | | | | | | |
| Linear1 $\Rightarrow$ relu | [1,160] | [1,512] | [1,160] | [1,512] | [1,160] | [1,512] |
| Linear2 $\Rightarrow$ relu | [1,512] | [1,1024] | [1,512] | [1,1024] | [1,512] | [1,1024] |
| Linear3 $\Rightarrow$ sigmoid | [1,1024] | [1,784] | [1,1024] | [1,3072] | [1,1024] | [1,2304] |

**Expectation Maximization (EM) Routing CapsNet:**  The second capsule architecture used in our proposed methodology is the matrix capsule with EM routing Hinton et al. (2018). The case is further validate the proposed methodology and obtained insights. The following describes the considered architecture in more detail.

Moving beyond the concept of vector outputs, Hinton et al. (2018) introduced matrix capsules (CapsNetsEM) that employ the EM routing algorithm (EMR), leveraging capsules to work effectively. At different capsule levels, each capsule is associated with a $4 \times 4$ pose matrix ($M$) and its activation probability ($a$). $M$ encodes the pose parameters where $M$ learns to form the relationship between the object and the pose (the viewer or some of the object), while $a$ represents the presence of a particular feature. The transformation matrices ($W$) are learned during the backpropagation step. The potential votes $V$ are associated with coupling coefficients ($e_{ij}$) where they are learned during specified iterations using EMR algorithm. Each capsule $i$ (the child) at layer $l$ will vote for $M$ of capsule $j$ at layer $l+1$ (the parent) as follows $V_{ij} = M_i W_{ij}$. By multiplying the $M_i$ and $W_{ij}$ between a given layer and the layer above that could learn to represent the hierarchical relationships between the parts and its whole. (please refer to Hinton et al. (2018) for more details of the inner workings of matrix capsules). To have consistency on our experiments, we added a similar decoder at the end of the considered architecture. Table 5 presents more details of Matrix capsules with EM routing. We show extra details of the input and output shapes in both the encoder and decoder parts.

Table 5: The input and the output shapes of the considered capsule network architecture (EM Routing Hinton et al. (2018)

| Layers | *MNIST* | | *SVHN & CelebA* | | *PASCAL Parts* | |
|---|---|---|---|---|---|---|
| | Input shape | Output shape | Input shape | Output shape | Input shape | Output shape |
| *Encoder:* | | | | | | |
| $Conv \Rightarrow$ BatchNorm | [1,28,28] | [14,14,32] | [3,32,32] | [16,16,32] | [3,48,48] | [24,24, 32] |
| BatchNorm $\Rightarrow$ relu | [14,14,32] | [14,14,32] | [16,16,32] | [16,16,32] | [24,24, 32] | [24,24, 32] |
| $PC \Rightarrow ConvC1$ | [14,14,544] | [14,14,544] | [16,16,32] | [16,16,544] | [24,24, 32] | [24,24,544] |
| $ConvC1 \Rightarrow ConvC2$ | [14,14,544] | [6,6,544] | [16,16,544] | [7,7,544] | [24,24, 544] | [11,11, 544] |
| $ConvC2 \Rightarrow ClassC$ | [6,6,544] | [4,4,544] | [7,7,544] | [5,5,544] | [11,11, 544] | [9,9, 544] |
| $ClassC \Rightarrow decoder$ | [4,4,544] | [512, 10,17] | [5,5,544] | [(10 or 2),17] | [9,9, 544] | [20,17] |
| *Decoder:* | | | | | | |
| Linear1 $\Rightarrow$ relu | [1,160] | [1,512] | [1,160] | [1,512] | [1,160] | [1,512] |
| Linear2 $\Rightarrow$ relu | [1,512] | [1,1024] | [1,512] | [1,1024] | [1,512] | [1,1024] |
| Linear3 $\Rightarrow$ sigmoid | [1,1024] | [1,784] | [1,1024] | [1,3072] | [1,1024] | [1,6912] |

# D  METHODOLOGY

In order to assess the learned representations that are encoded in both capsule architectures, a perturbation analysis was conducted on different parts of both capsule architectures.

**Perturbation Analysis:** we introduce our perturbation protocol in Sec. 4.1 which aims to define the perturbation interval based on computing the first order statistics including all layers across the entire datasets. We conducted the perturbation analysis by following the defined procedure shown in Alg. 1. The algorithm is general enough for both capsule architectures.

---

**Algorithm 1** Perturbation Analysis

---

0: $a_{s,c}$ the internal activations
0: $v_j$ output vector
0: $d$ dimensions
0: $f$ features
0: $\alpha$ perturbation ranges
0: **while** $x_s$ **do** in dataset
0:     for all filters in a given layer: extract(f)
0:     flatten(f)
0:     $a_{s,c} \leftarrow$ concatenations the flatten filters
0:     repeat finding $a_{s,c}$ for all classes
0:     find(min, max, mean, and std of $(a_{s,c})$) column-wise in a given class
0:     $A_c \leftarrow$ first order statistics(min, max, mean, and std of $(a_{s,c})$) column-wise in all classes
0:     $A_{all} \leftarrow$ statistics(min, max, mean, and std of $(A_c)$) column-wise in all classes
0: **end while**
0:     **while** $d < 16$ **do**
0:         $\hat{v}_j = $ copy$(v_j)$
0:         **while** $r$ within $\alpha$ **do**
0:             $\hat{v}_j[d] = \hat{v}_j[\alpha$ values$]$
0:         **end while**
0:     **end while**
   =0

---

## D.1  LAYER-WISE RELEVANT UNIT SELECTION

This section aims at further extending the description of our methods presented in the original manuscript (see Sec. 4.2). The following methods were defined to detect the relevant features/units that define activation paths in a given pre-trained CapsNets.

**Conv:** In the case of the *Conv* layer, GAP was computed in filter level. This step aimed at obtaining the top-ranking filters based on the highest GAP values which indicated the highest average activations. Through a cross-validation step, top-*k* values were selected as the most relevant filters in *Conv* layer. In the case of *CapsNetEM* architecture, we followed the same procedure for *Conv layer* since they share similar characteristics. Additionally, the primary capsule layer in *CapsNetEM* is only used for preparing the pose matrix *M* and the activation *a* for the EM routing algorithm. Therefore, we followed the same procedure to determine the top-ranking filters by considering the activations *a* at the primary capsule layer for this experiment. We followed the procedure defined in Alg. 2 for both capsule architectures.

---

**Algorithm 2** *Conv* Layer Units Selection

---

0: $x_m \leftarrow$ input example
0: $k \leftarrow$ top-$k$
0: $f \leftarrow$ filters
0: D $\leftarrow$ filters with ascending orders
0: **while** $x_m$ in dataset **do**
0:     D $\leftarrow$ ranked(GAP(filters))
0:     $k \leftarrow$ select(top-$k$(D))
0:     **while** $k <$ length(D) **do**
0:         start $k \leftarrow 1$
0:         **if** $k$ in D **then**
0:             $Conv[f] \leftarrow Conv[k]$
0:         **else if**
0:             **then**$Conv[f] \leftarrow 0$
0:         **end if**
0:         gradually increased($k$)
0:     **end while**
0: **end while**=0

---

**Algorithm 3** Class-Agnostic Forward Path Estimation

---

0: $x_m \leftarrow$ input example
0: $k \leftarrow$ top-$k$ relevant units
0: $e_{ij} \leftarrow$ coupling coefficients
0: $E^* \leftarrow$ optimal coupling coefficients
0: **while** $x_m$ in dataset **do**
0:     extract($e_{ij}$) for all classes
0:     ranked($e_{ij}$) with corresponding capsules
0:     **if** $e_{ij} > 0.3$ **then**
0:         $E^* \leftarrow$ index[$e_{ij}$]
0:         $k \leftarrow$ select(top-$k$($E^*$))
0:     **end if**
0: **end while**=0

---

**Class-Agnostic Forward Path Estimation:** In contrast with the *Conv* layer, the relevant units were selected based on the different procedures in the capsule layers. We extract the units considering the routing algorithms (both algorithms, dynamic routing and EM routing) which determine how $i^{th}$ capsule flows to the $j^{th}$ capsule during the forward pass. We followed the procedure outlined in Alg. 3 for both capsule architectures. These steps are designed to determine the optimal routings $E^*$. We select the top-$k$ units that flow forward starting from the input layer until we obtain the output prediction. We added an extra step directly after we obtained the optimal routings $E^*$. We needed to recompute the routing assignment probabilities by multiplying these assignments with the input probabilities. Then, we multiply the results with the potential votes. This means we are reconsidering only those votes with higher assignment routing coefficients.

**Class-Aware Backward Path Estimation:** Different from the previous method, we followed the procedure described in Alg. 4 for the top-$k$ selection. $E^*$ were obtained based on a given predicted class $\hat{y}$. We also repeated the same step by reconsidering only those votes with higher assignment

properties as that were followed in **Class-Agnostic Forward Path Estimation** process.

---

**Algorithm 4** Class-Aware Backward Path Estimation

---

0: $x_m \leftarrow$ input example
0: $k \leftarrow$ top-$k$ relevant units
0: $\hat{y}$ the predicted label
0: $E^* \leftarrow$ optimal coupling coefficients
0: **while** $x_m$ in *dataset* **do**
0:      $\hat{y} \leftarrow$ CapsNet($x_s$)
0:      $e_{ij} \leftarrow$ extract(coefficients given $\hat{y}$))
0:      $e_{ij} \leftarrow$ ranked($e_{ij}$)
0:      $E^* \leftarrow$ select(top-$k(e_{ij})$)
0: **end while** =0

---

Putting the previous steps together, we defined the full path along with all layers based on the steps defined in Alg. 5. In the case of *CapsNetEM* architecture Alg. 6, we assume the child layer is *ConvCaps2* and the parent layer is *ClassCaps*. For the case of the *ConvCaps1* and the *ConvCaps2*, we adopted a fixed number of relevant capsules by considering the minimal number of capsules that gave us a higher classification performance. We previously defined those steps in the **Class-Agnostic Forward Path Estimation** and **Class-Aware Backward Path Estimation** sections.

---

**Algorithm 5** Full Path Units Selection

---

0: $x_m \leftarrow$ input example
0: $k \leftarrow$ top-$k$ relevant units
0: D$\leftarrow$ filters with ascending orders
0: $E^* \leftarrow$ optimal coupling coefficients
0: **while** $x_m$ in *dataset* **do**
0:      start $k \leftarrow 1$
0:      **if** $k$ in D **then**
0:          $Conv[f] \leftarrow Conv[k]$
0:      **else**
0:          $Conv[f] \leftarrow 0$
0:      **end if**
0:      **if** $k$ in $E^*$ **then**
0:          $PC[i^{th}] \leftarrow PC[k]$
0:      **else**
0:          $PC[i^{th}] \leftarrow 0$
0:      **end if**
0:      gradually increased($k$)
0: **end while** CapsNet(path) =0

---

## E    EXPERIMENTS

**Classification Performance:** On the one hand, in Fig. 7 we depict the classification performance and loss values during the training step when we consider the CapsNet model from Sabour et al. (2017). On the other hand, in Fig. 8 we show the classification performance and loss curves during the training step and we inspect the training process of our CapsNetEM model Hinton et al. (2018) to ensure it is trained properly. For the CapsNetEM Hinton et al. (2018), the performance using the forward path is higher than the dense model and the backward path as stated in Table 6.

### E.1    PERTURBATION ANALYSIS

This step was conducted to analyze the impact of perturbing the dimensions of $v_j$ that were systemically replaced by values in a range of $\alpha$ with fixed steps $\xi$. To this end, the qualitative

---

**Algorithm 6** Full Path Units Selection considering EM routing

---

0: $x_m \leftarrow$ input example
0: $k \leftarrow$ top-$k$ relevant units
0: D$\leftarrow$ filters with ascending orders
0: $E^* \leftarrow$ optimal coupling coefficients
0: **while** $x_m$ in *dataset* **do**
0:    start $k \leftarrow 1$
0:    **if** $k$ in D **then**
0:      $Conv[f] \leftarrow Conv[k]$
0:    **else**
0:      $Conv[f] \leftarrow 0$
0:    **end if**
0:    **if** $k$ in D **then**
0:      $PC[f] \leftarrow PC[k]$
0:    **else**
0:      $PC[f] \leftarrow 0$
0:    **end if**
0:    **if** $k$ in $E^*$ **then**
0:      $ConvCaps1[i^{th}] \leftarrow ConvCaps1[k]$
0:    **else**
0:      $ConvCaps1[i^{th}] \leftarrow 0$
0:    **end if**
0:    **if** $k$ in $E^*$ **then**
0:      $ConvCaps2[i^{th}] \leftarrow PC[k]$
0:    **else**
0:      $ConvCaps2[i^{th}] \leftarrow 0$
0:    **end if**
0:    **if** $k$ in $E^*$ **then**
0:      $ClassCaps[i^{th}] \leftarrow ClassCaps[k]$
0:    **else**
0:      $ClassCaps[i^{th}] \leftarrow 0$
0:    **end if**
0:    gradually increased($k$)
0: **end while** CapsNet(path) =0

---

experiments were conducted w.r.t. the provided classes in the considered datasets. In Fig. 11 (the top), 12, 14, 16, 18, and 20 (the top), we show a few qualitative results in the form of reconstructions obtained from the perturbations introduced in Alg. 1. It can be noted that several features (e.g., rotation and thickness) of the reconstructed class get modified.

Similarly, we followed the same procedure to conduct our experiments considering CapsNetEM. We systemically perturb the dimensions of the pose matrix $M$ by replacing the original value of a given dimension with values in a range of $\alpha$ with fixed steps $\xi$.

In the case of CapsNetEM, we found the $\alpha$ values between [-1,0.1]. Therefore, we used fixed steps $\xi$=0.08. In Fig 11 (the bottom), 13, 15, 17, 19, and 20 (the bottom), we show some of our

Table 6: Mean classification accuracy on the MNIST, SVHN, PASCAL-Parts, and CelebA datasets for the original dense CapsNet and a sparser version based on the identified activation path

| Capsule Type | | MNIST | | | SVHN | | | PASCAL | | | CelebA | | |
|---|---|---|---|---|---|---|---|---|---|---|---|---|---|
| | | Train | Valid | Test | Train | Valid | Test | Train | Valid | Test | Train | Valid | Test |
| **Dynamic Routing** | Dense (original) | 99.8 | 99.9 | 99.1 | 96.9 | 96.7 | 91.0 | 99.5 | 99.1 | 78.0 | 92.2 | 93.0 | 92.0 |
| | Backward Path | 74.9 | 76.9 | 78.5 | 89.0 | 89.3 | 87.2 | 59.0 | 61.0 | 56.0 | 81.1 | 81.4 | 80.6 |
| | Forward Path | 95.2 | 95.2 | 95.1 | 96.0 | 96.0 | 88.3 | 66.7 | 65.8 | 60.1 | 88.2 | 89.1 | 86.2 |
| **Em Routing** | Dense (original) | 99.1 | 98.1 | 98 | 82.7 | 81.4 | 80 | - | - | - | 85.9 | 85.7 | 85.7 |
| | Backward Path | 98.5 | 98.6 | 97.6 | 83.9 | 84.3 | 78.7 | - | - | - | 86.7 | 85.6 | 85.1 |
| | Forward Path | 98.9 | 98.8 | 97.8 | 84 | 83.3 | 78.9 | - | - | - | 86.9 | 85.7 | 85.2 |

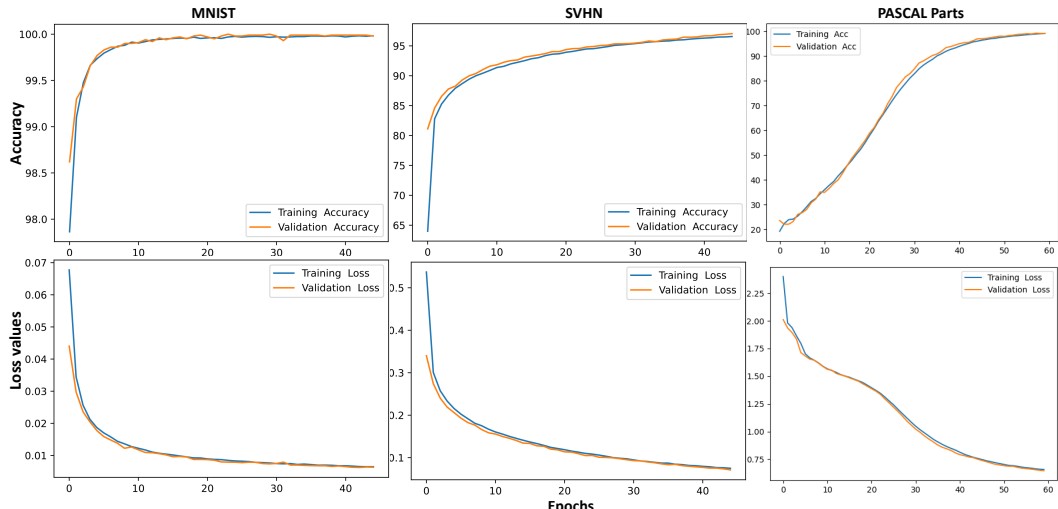

Figure 7: Mean classification performance on the train and validation of the MNIST, SVHN, and PASCAL-Parts dataset for the original dense CapsNet

reconstruction results when we perturb the Pose matrix $M$ following the algorithm (see Alg. 1) that we introduced earlier.

## E.2 LAYER-WISE RELEVANT UNITS SELECTION

**Convolutional Layer Ablation:** This section is extended from the original manuscript. This section shows more results regarding the selected top-$k$ values from the cross-validation procedure. In (Fig. 21.a & 21.b), we show the impact of classification performance. We notice the results in the case of MNIST are better than in the case of SVHN.

For the case of *CapsNetEM*, we followed the same procedure and in Fig 22 we show the impact of the classification performance on several level of layers in the architecture. We consider both datasets MNIST (the left) and SVHN (the right).

We noticed the same behavior on both models such that the classification performance significantly dropped when we removed relevant filters/units. For *CapsNetEM*, we notice that when we move

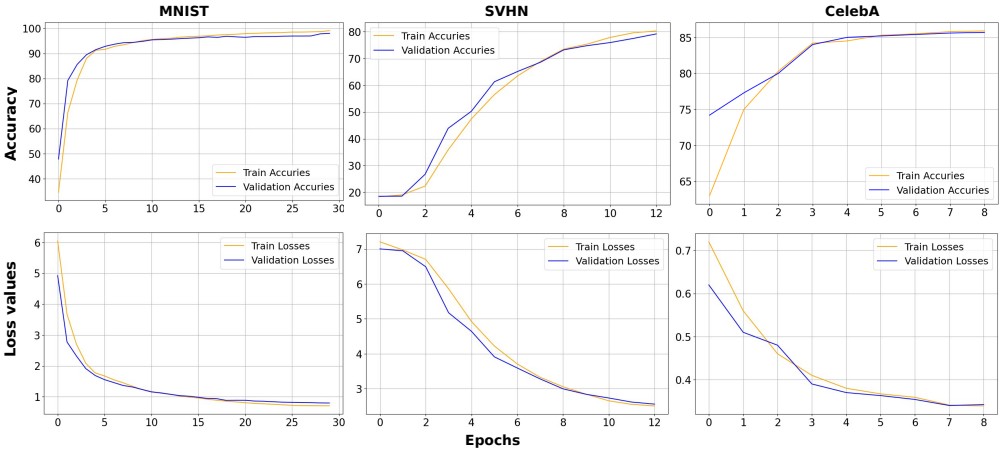

Figure 8: Mean classification performance on the train and validation of the MNIST and SVHN of EM matrix dense CapsNet

**MNIST**                    **SVHN**

**Class-Agnostic Perturbation Ranges (MNIST) [-0.55, 0.5] of (0.09) steps, (SVHN [-0.6, 0.6]) of (0.1) steps**

**Class-Agnostic Perturbation Range on both datasets [-0.3, 0.3] of (0.05) steps**

**Classes**

**Class-Agnostic Perturbation Ranges and Steps Respectively:**
**(MNIST) Class-2 [-0.2, 0.3] (0.045), Class-8 [-0.45, 0.22] (0.06), Class- [-0.27, 0.29] (0.05)**
**(SVHN) Class-2 [-0.25, 0.28] (0.048), Class-8 [-0.2, 0.25] (0.04), Class- [-0.4, 0.4] (0.07)**

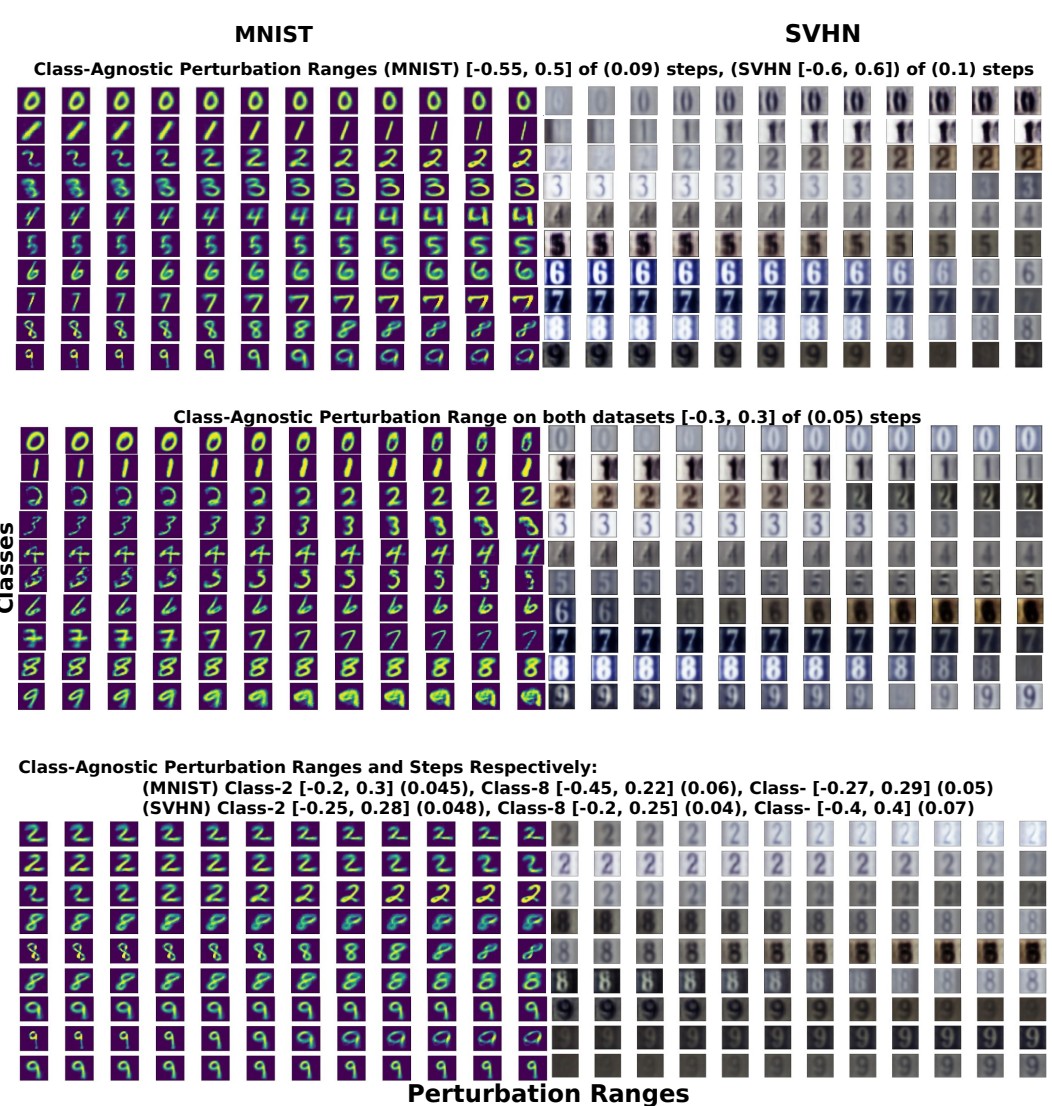

**Perturbation Ranges**

Figure 9: Qualitative Examples of reconstructed inputs as the vector $v_j$ is perturbed Considering CLass-Agnostic and Class-Specific.

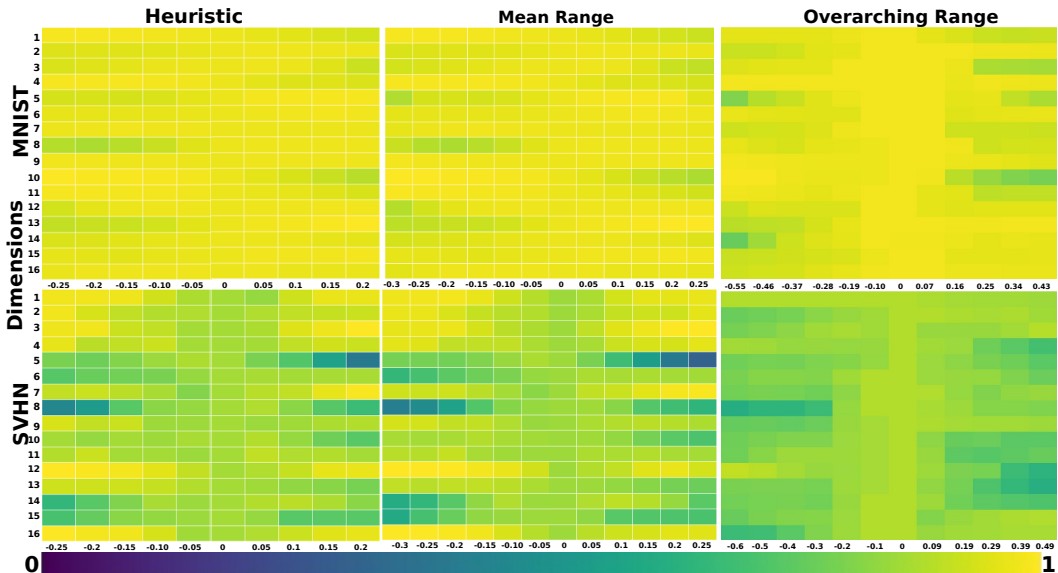

Figure 10: Mean classification Accuracy as perturbations are applied to the 16 dimensions of vector $v_j$ on different intervals. We present results for the standard perturbation approach followed in the literature (1st column), the Mean Range across classes obtained from variant of the proposed method (2nd column) and the largest overarching range covering the activation ranges of all the classes of interest (3rd column).

toward the end of the architecture, the classification performance significantly increases when we consider a few relevant units.

**Capsule Layer Ablation:** Given top-$k$ relevant units that were defined previously, when the *PC* is only modified. In Fig. 21.c and Fig. 21.d show the impact of the classification performance in the case of relevant $k$ units were increased gradually. In Fig. 23, 24, 25,and & Fig. 26, we show the distribution of the magnitude of $v_j$ over classes. These experiments were conducted in both the MNIST and SVHN datasets. In Fig. 27, 31, 29 & 33, we present additional qualitative reconstruction results of CapsNets (Dynamic routing) when we considered the top-$k$ relevant units that were defined previously. We started from top-$k$=1 and we increased the number of top-$k$ units gradually to reconstruct some input examples of different classes.

Similarly, we show further qualitative reconstruction results of *CapsNetsEM* (EM routing) when we considered the top-$k$ relevant units that were defined previously. We started from top-$k$=1 and we increased the number of top-$k$ units gradually to reconstruct some input examples of different classes. We present in Fig. 28, 32, 30 & 34 reconstructions of *CapsNetsEM* on MNIST and SVHN. For the case of CapsNetsEM, as we mentioned earlier, there are three capsule layers. In this case, we fixed the number of relevant units on both layers; the *ConvCaps1* and the *ConvCaps2* where top-$k$=10 as we defined earlier based on the impact of the classification performance in Sec. 4.2. Therefore, we conduct our experiments on the *ClassCaps* layer. We noticed that when we used the forward path, the reconstructions were better compared to identifying the backward path.

**Network Ablation:** In Fig. 35, we show qualitative reconstructions based on the path that was defined in our proposed protocol. In a similar manner, we show the reconstructed input examples in the case of *CapsNetEM* in Fig. 36

We noticed that we correctly reconstructed the input examples with their predictions when only using a small number of relevant units on both datasets for both capsule architectures.

Similarly, we looked for shared and exclusive units among the top-$k$ units selected for each class by the selection method we have introduced in Sec. 4.2. See Fig. 37, in the case of CapsNetsEM by Hinton et al. (2018), we noticed that there are less exclusive units on both datasets.

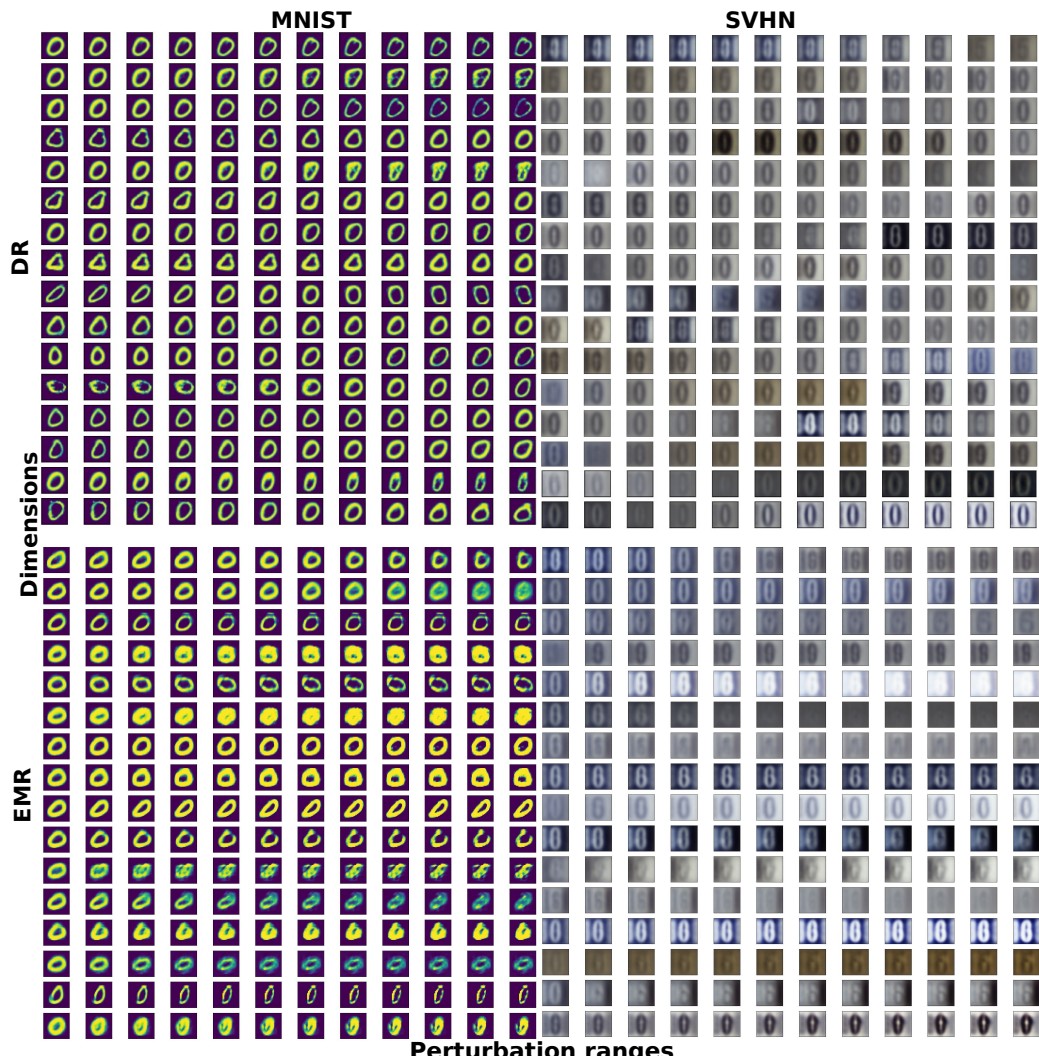

Figure 11: Qualitative example of reconstructed inputs as the vector $v_j$ the pose matrix $M$ are perturbed - class 0 on both architectures. The top shows DR reconstructions (DR refers to the dynamic routing) and the button shows EMR reconstructions (EMR refers to matrix capsule with EM routing)

Table 7: Number (%) of shared top-$k$ units among two classes.

| Dataset | C1-C6 | C3-C8 | C4-C5 | C0-C8 | C5-C8 | C7-C5 | C6-C9 |
|---------|-------|-------|-------|-------|-------|-------|-------|
| MNIST   | 2 (20) | 3 (30) | 1 (10) | 3 (30) | 3 (30) | 3 (30) | 0 (0) |
| SVHN    | 0 (0)  | 0 (0)  | 1 (10) | 1 (10) | 0 (0)  | 0 (0)  | 1 (10) |

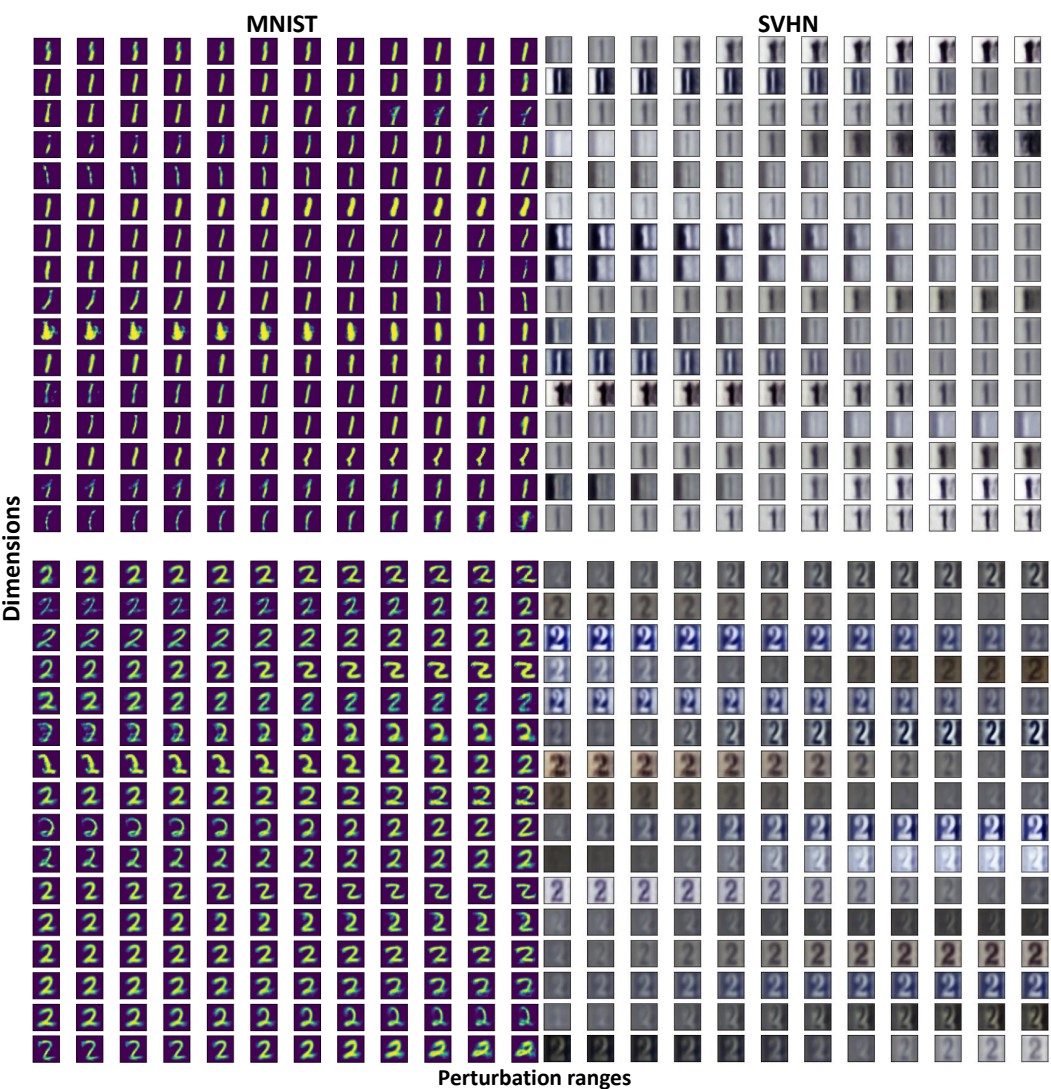

Figure 12: Qualitative example of reconstructed inputs as the vector $v_j$ is perturbed - classes 1&2.

### E.3 MEASURING *Part-Whole* RELATIONSHIP ENCODINGS

This section is an extension to the Sec. 6.3 that aims to measure the *part-whole* relationship encodings. We define an algorithm that shows the procedure we followed for measuring the level to which features encoded in a CapsNet encode *part-whole* relationships. In Alg. 7, we show the steps we followed to find our results. measure the spatial overlap between the internal responses of relevant capsules at layer $l$ (*Parts*) and the internal response at a layer $l+1$ (*Whole*) using Relevance Mass Accuracy (RMA) Arras et al. (2022). Tables 8, 9, and 10 show the first-order statistics of RMA scores when we considered *CapNets* proposed by Sabour et al. (2017). we compute the overlapping of the $h^l$ on the top of the aggregated relevant units $K$=5,15,30,200. We used several thresholds $Thr$=0.1, 0.25 & 0.5 respectively.

Similarly, we repeated the same experiments when we considered *CapsNetsEM* proposed by Hinton et al. (2018). We show the RMA results on the same thresholds in Tables 11, 12, and 13. Additionally, we show the mean of RMA scores when we consider only $k$=1 using the same thresholds and we

MNIST SVHN

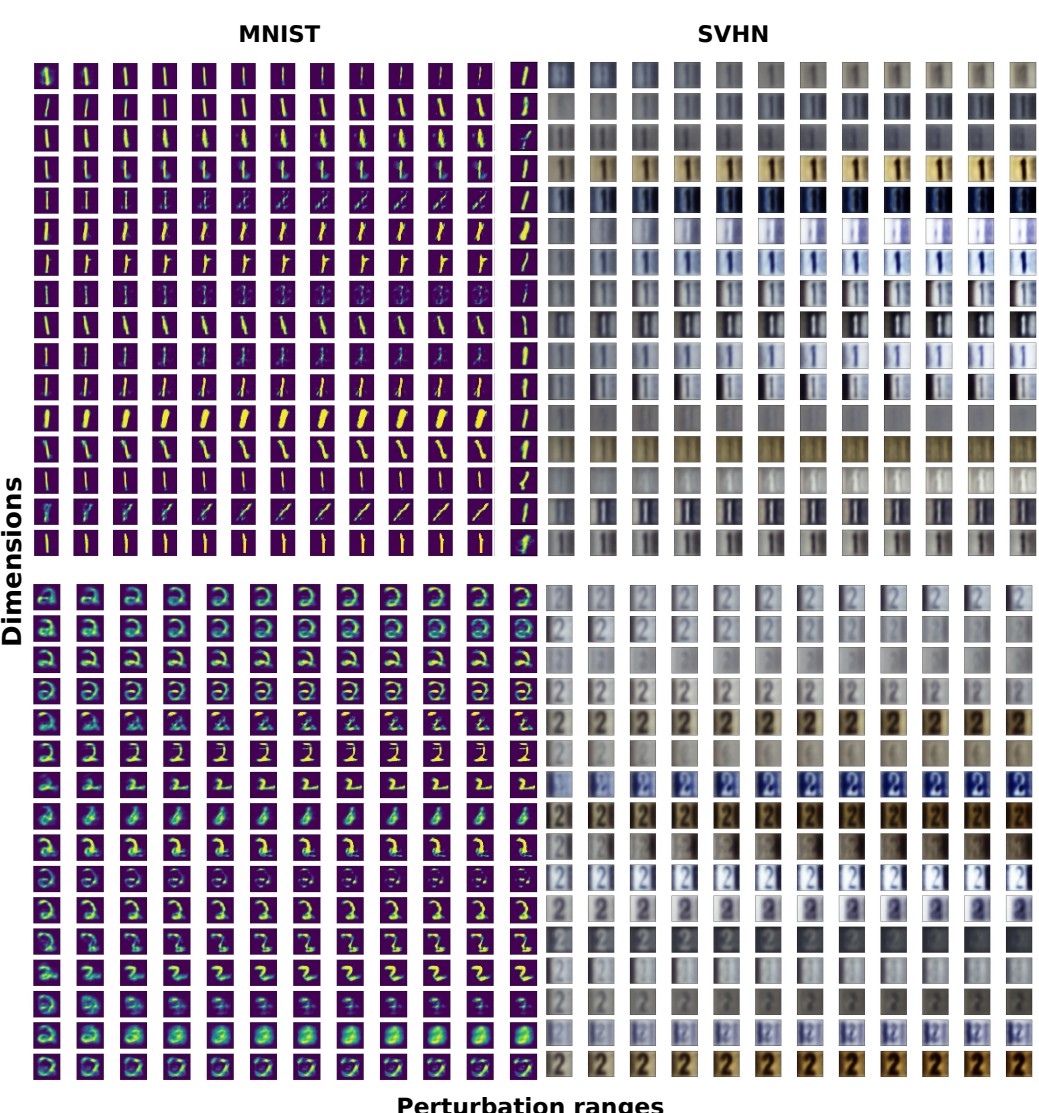

**Dimensions**

**Perturbation ranges**

Figure 13: Qualitative example of reconstructed inputs as the vector $v_j$ is perturbed - classes 1&2 (EM Routing).

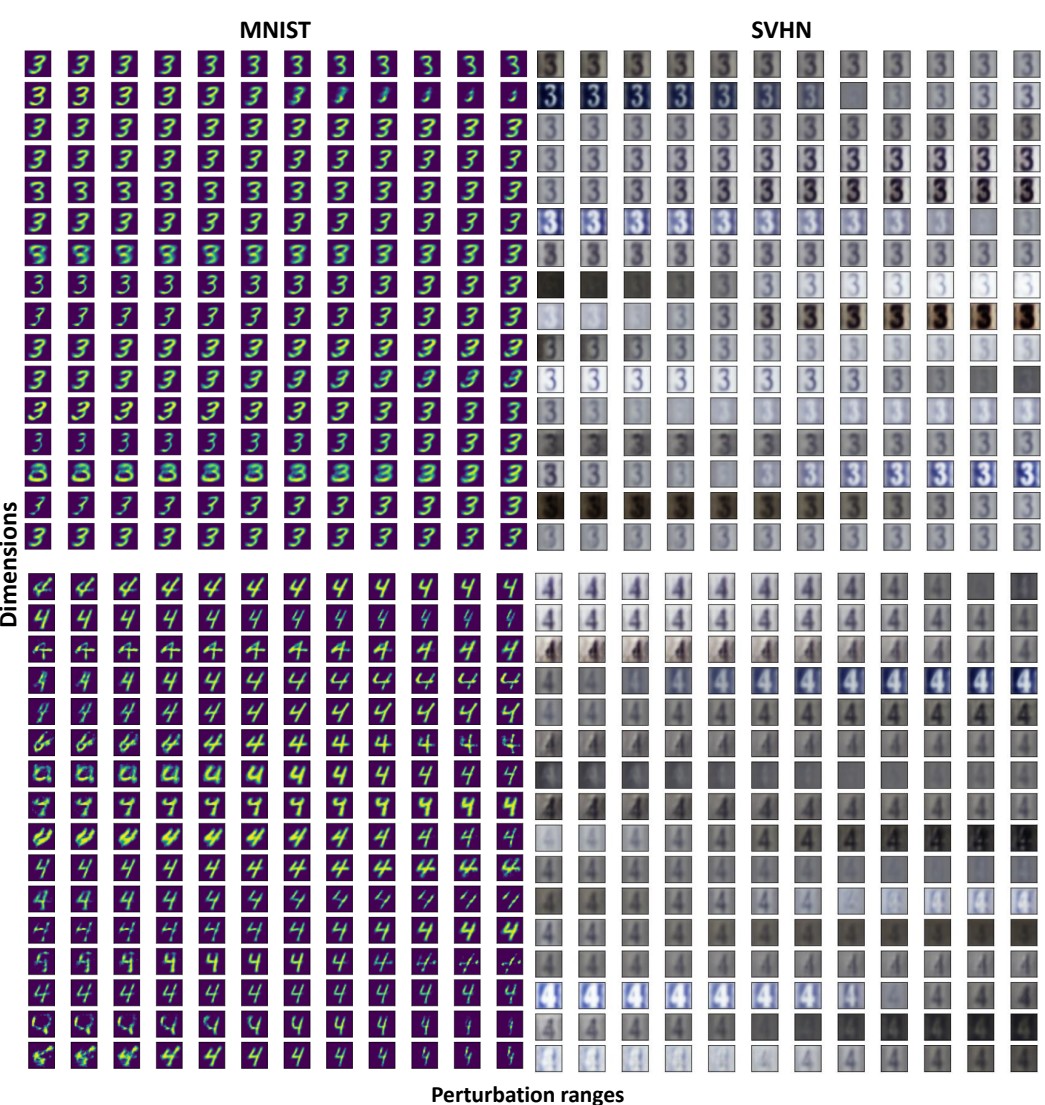

Figure 14: Qualitative example of reconstructed inputs as the vector $v_j$ is perturbed - classes 3&4.

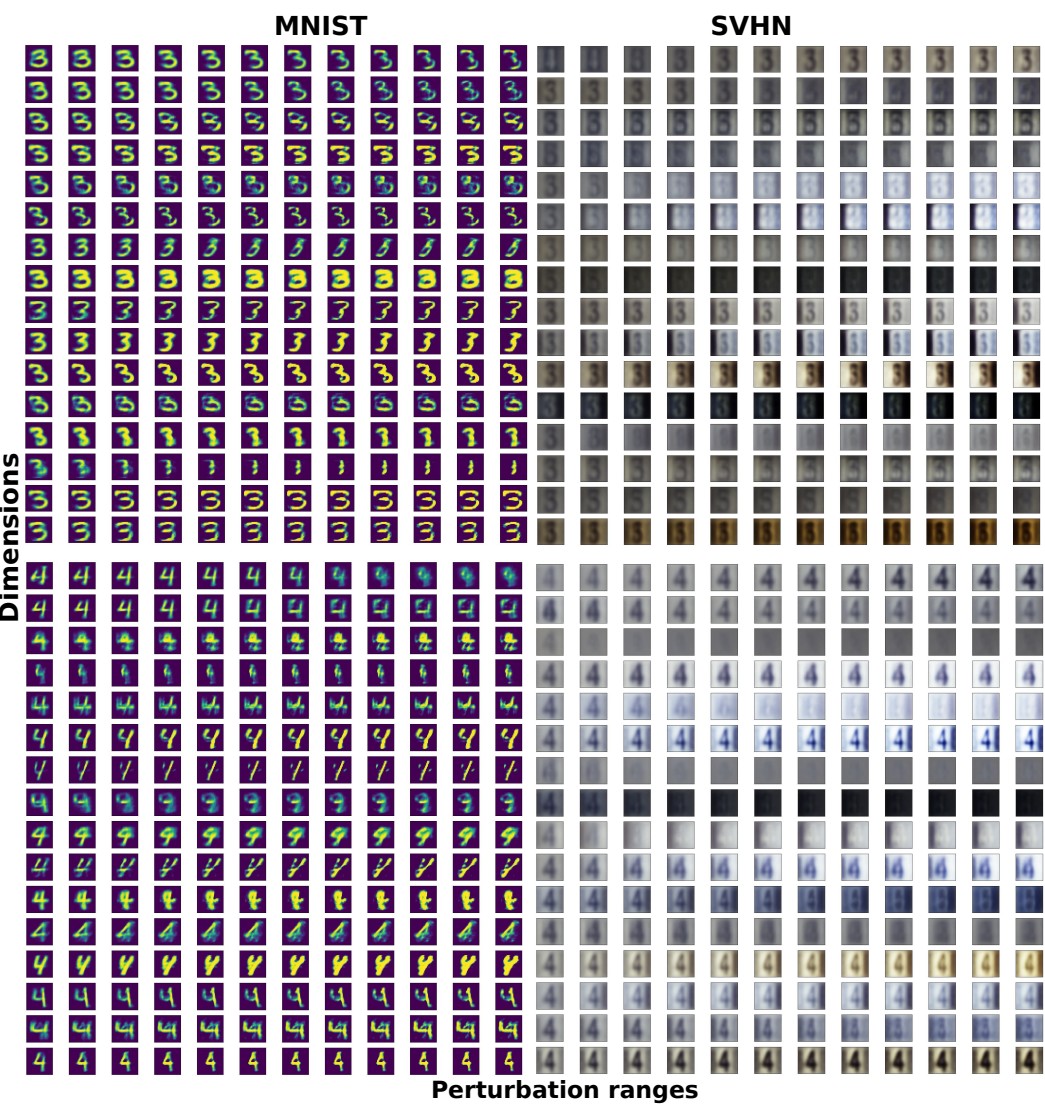

Figure 15: Qualitative example of reconstructed inputs as the vector $v_j$ is perturbed - classes 3&4 (EM Routing).

MNIST                                                    SVHN

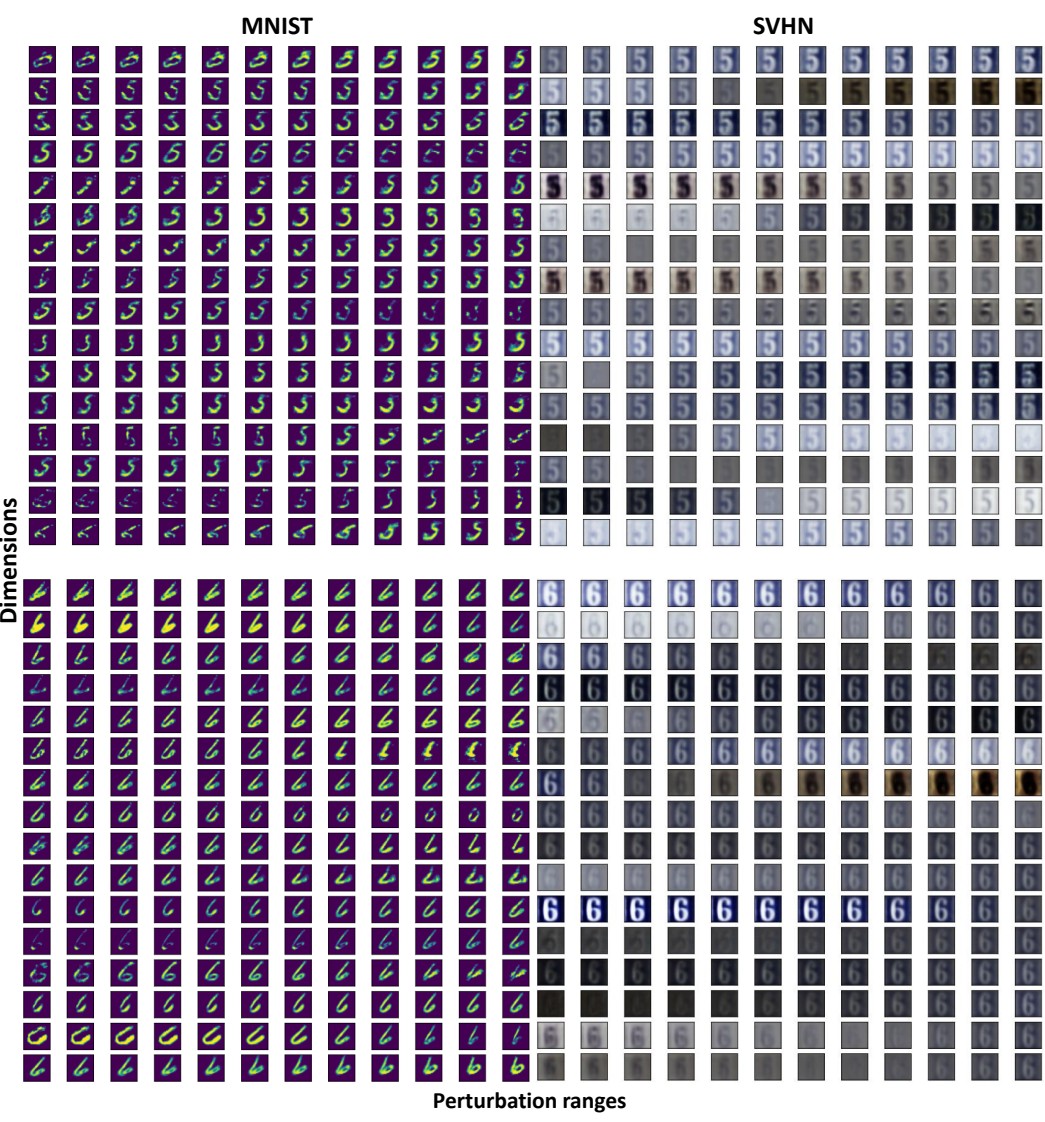

Figure 16: Qualitative example of reconstructed inputs as the vector $v_j$ is perturbed - classes 5&6.

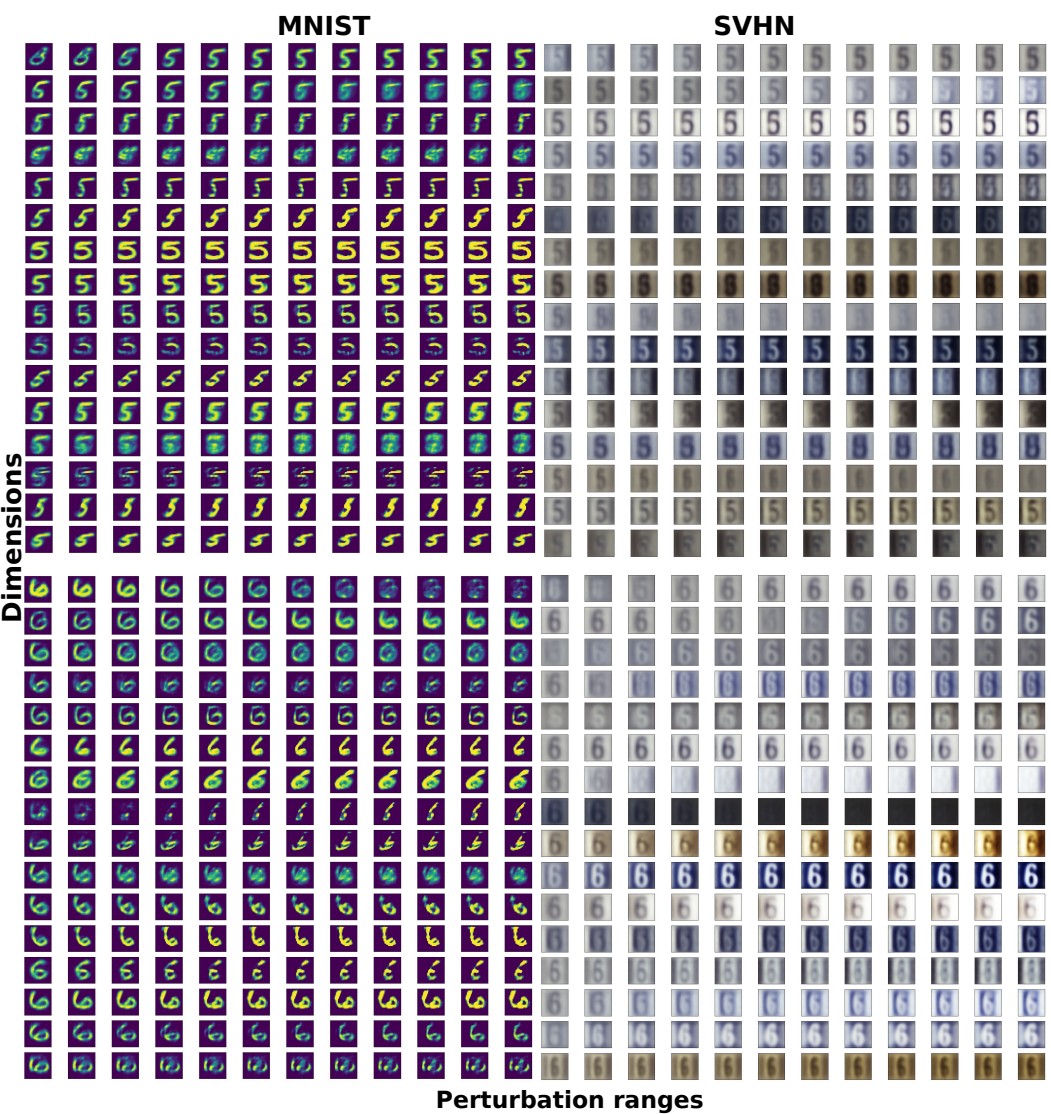

Figure 17: Qualitative example of reconstructed inputs as the vector $v_j$ is perturbed - classes 5&6 (EM Routing).

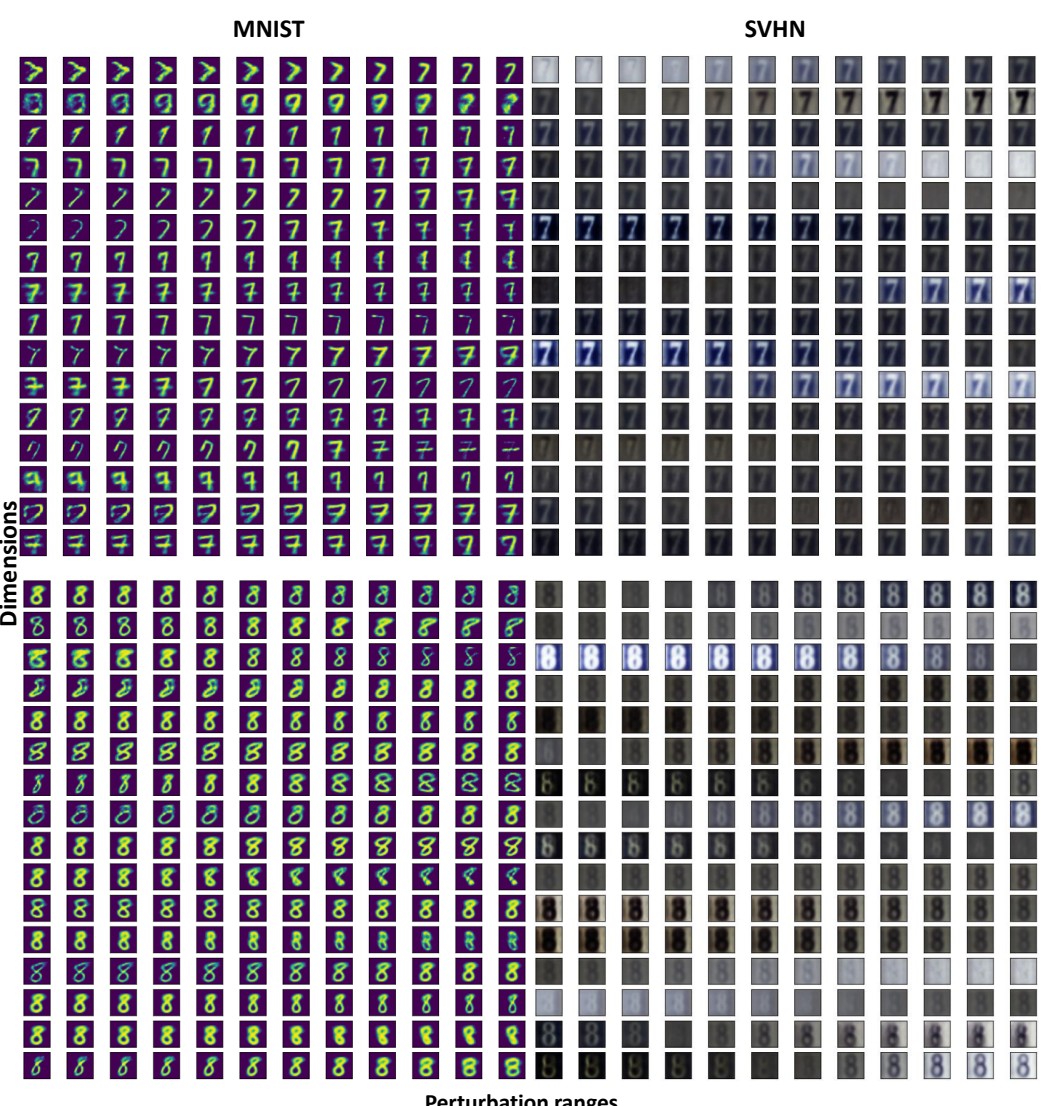

Figure 18: Qualitative example of reconstructed inputs as the vector $v_j$ is perturbed - classes 7&8 (Dynamic Routing).

**MNIST**  **SVHN**

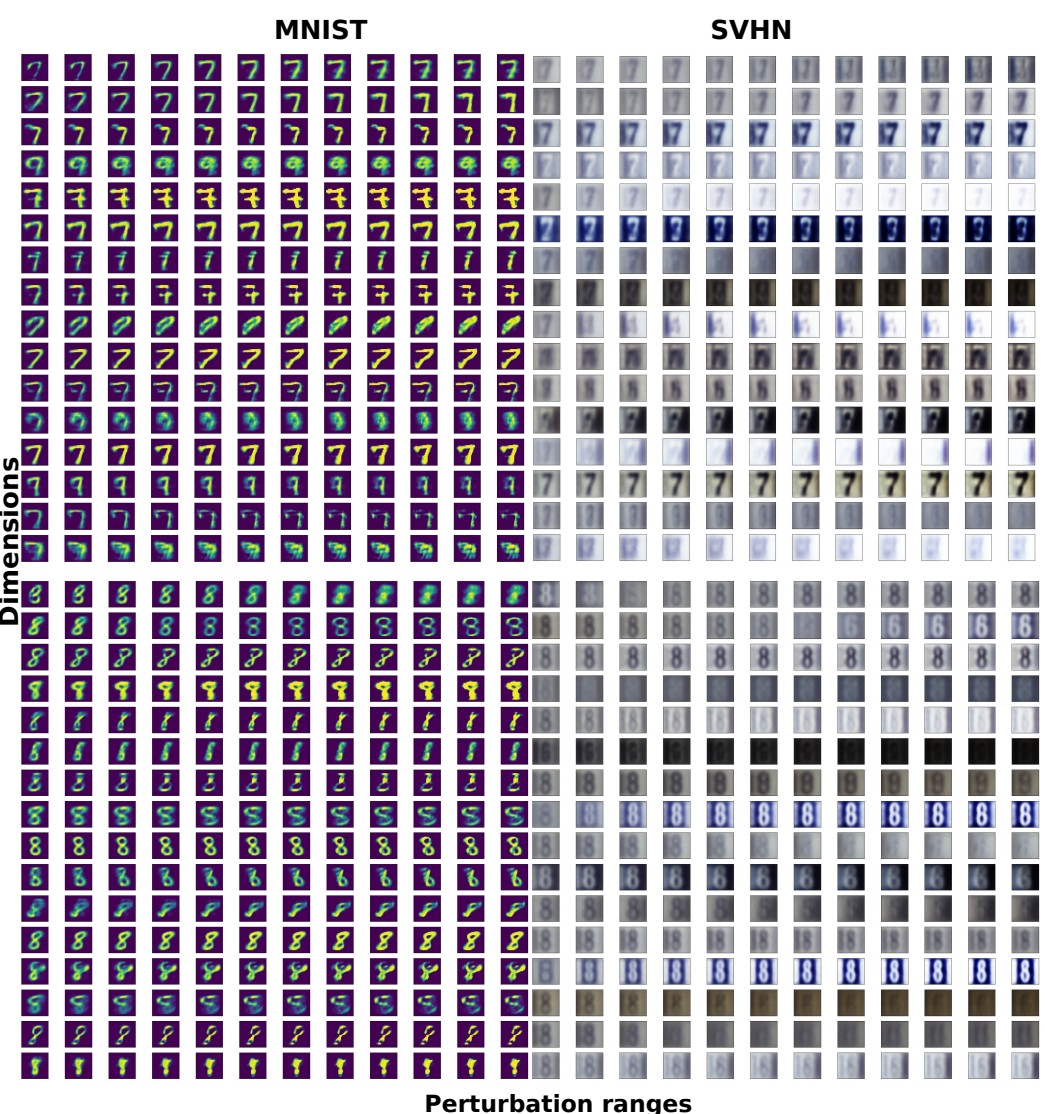

**Dimensions**

**Perturbation ranges**

Figure 19: Qualitative example of reconstructed inputs as the vector $v_j$ is perturbed - classes 7&8 (EM Routing).

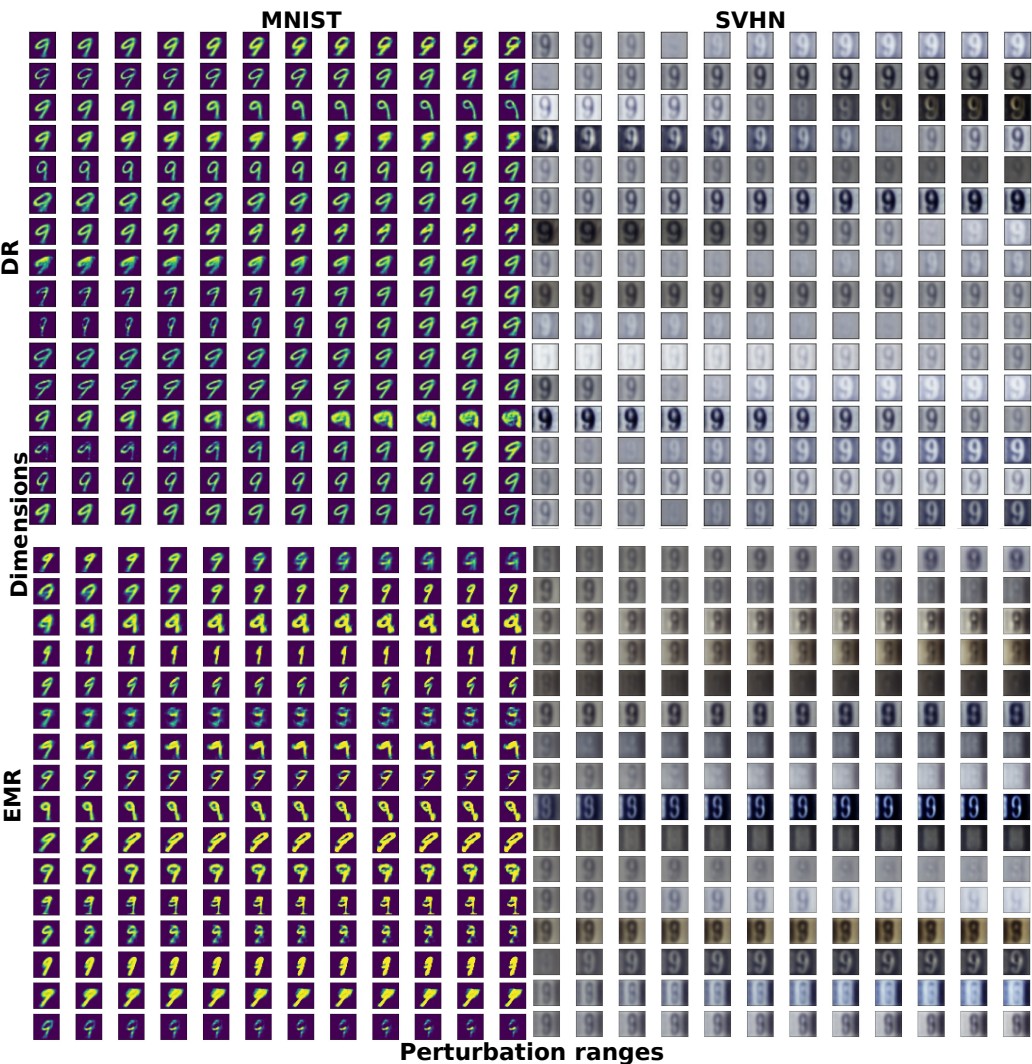

Figure 20: Qualitative example of reconstructed inputs as the vector $v_j$ the pose matrix $M$ are perturbed - class 9 on both architectures. The top shows DR reconstructions (DR refers to the dynamic routing) and the button shows EMR reconstructions (EMR refers to matrix capsule with EM routing).

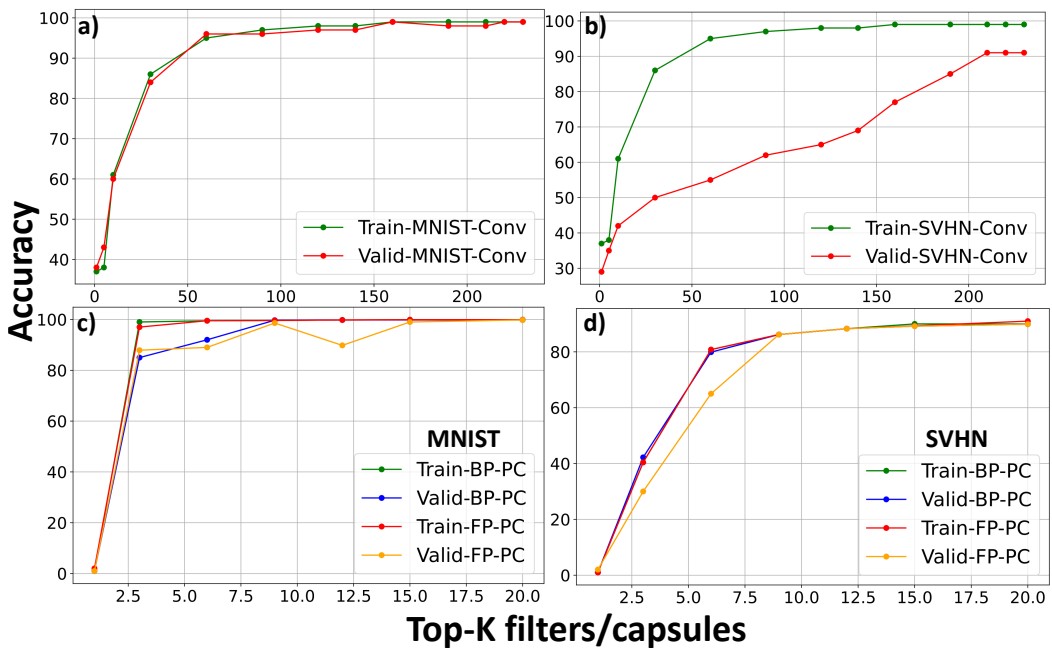

Figure 21: Mean classification accuracy on the MNIST, SVHN, and PASCAL-Parts dataset for the original CapsNet

reported the reults in Table. 14.

Fig. 38 presents the results of the responses considered for the computation of relevance mass accuracy (RMA). We followed a similar set-up to the results obtained by Sabour et al. (2017). We show the results when only $Thr$=0.25 and we considered the top relevant unit when $K$=1 using the same threshold $K$=5,15,30,200.

To summarize the observed results, we present the final results obtained from both capsule architectures in Table. 15. As we noticed in Fig. 38 and Table. 15, we anticipated similar trends on both capsule architectures and we noticed that CapsNetEM by Hinton et al. (2018) had a lower mean of RMA scores compared to CapsNets by Sabour et al. (2017).

Please note that the final results of CelebA and PASCAL are being computed and will be eventually added when available.

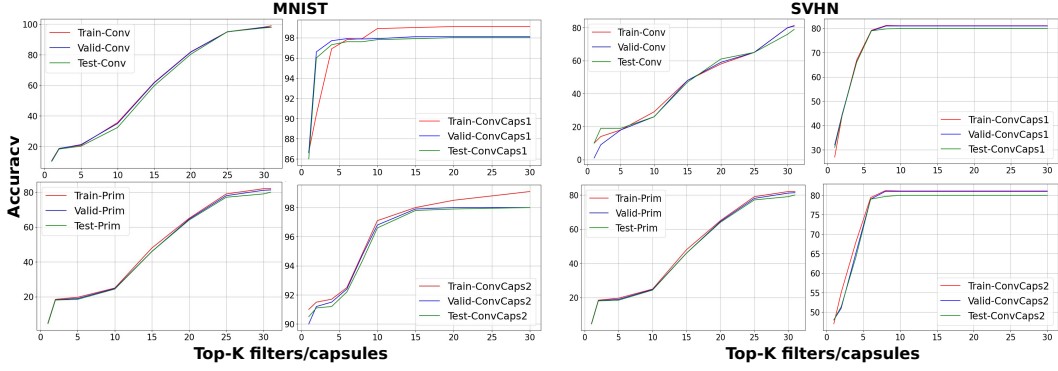

Figure 22: Mean classification accuracy on the MNISTdataset of EM matrix CapsNet

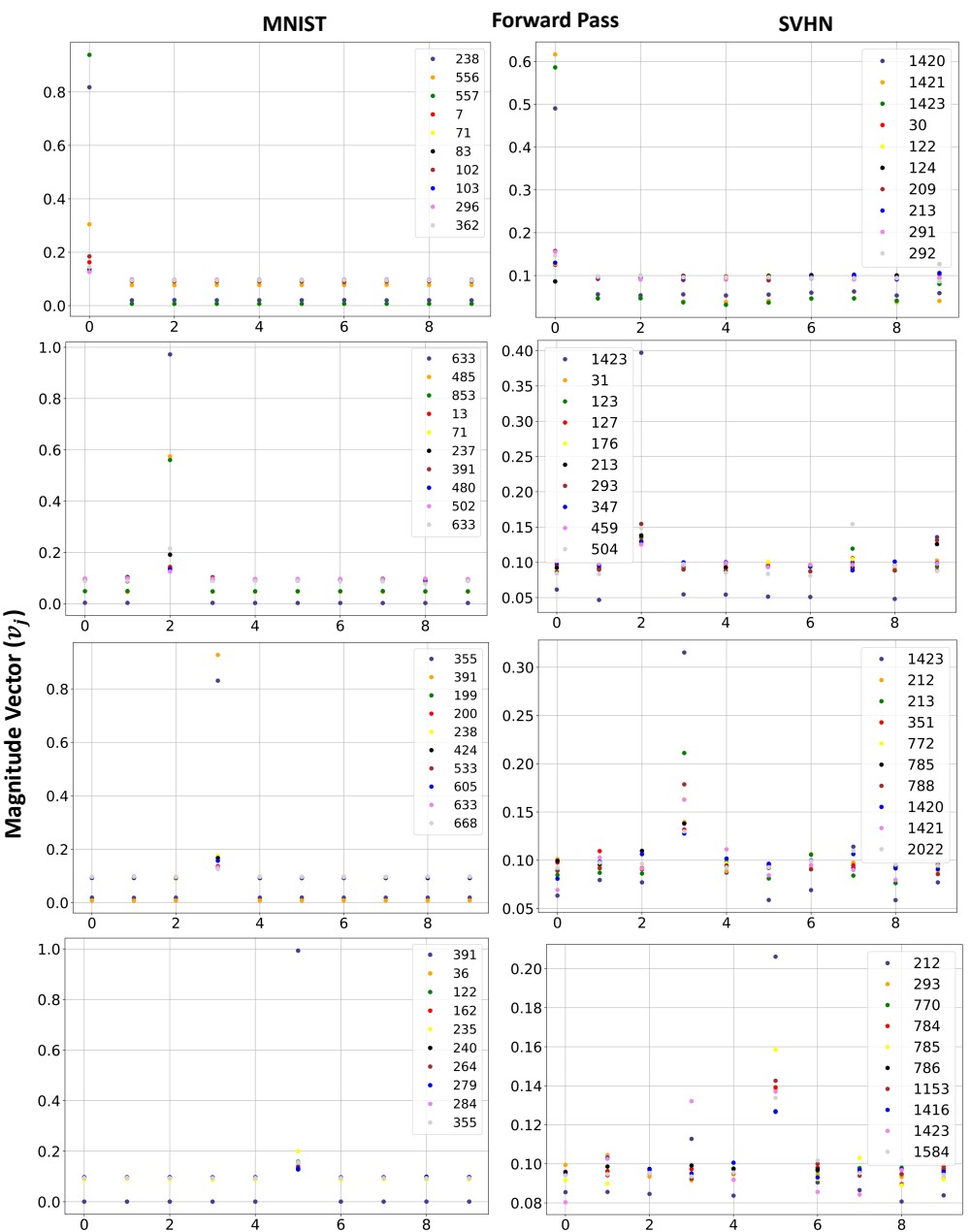

Figure 23: Vector Magnitude ($v_j$) for the selected $k$ units from the *forward* path.

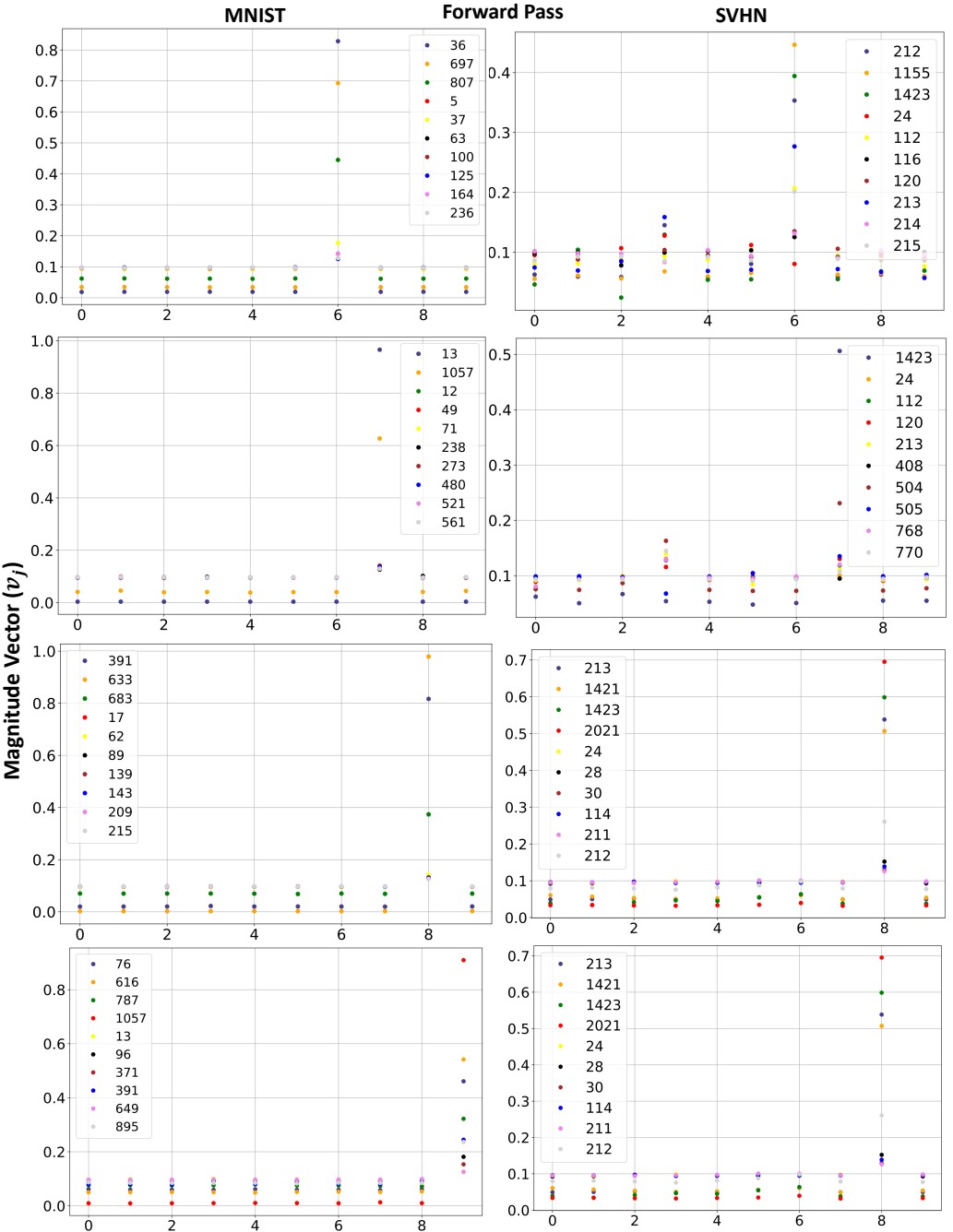

Figure 24: Vector Magnitude ($v_j$) for the selected $k$ units from the *forward* path

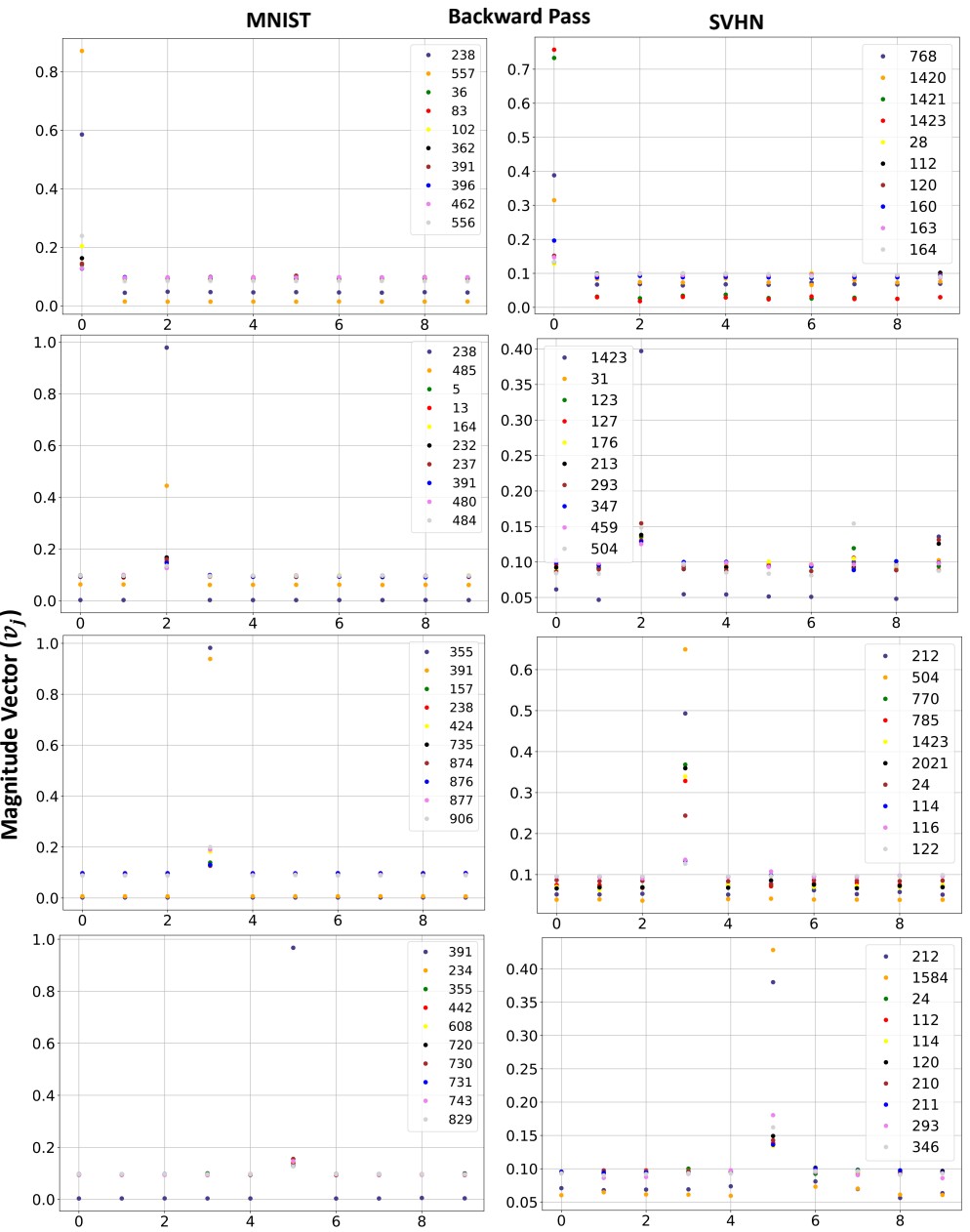

Figure 25: Vector Magnitude ($v_j$) for the selected $k$ units from the *backward* path

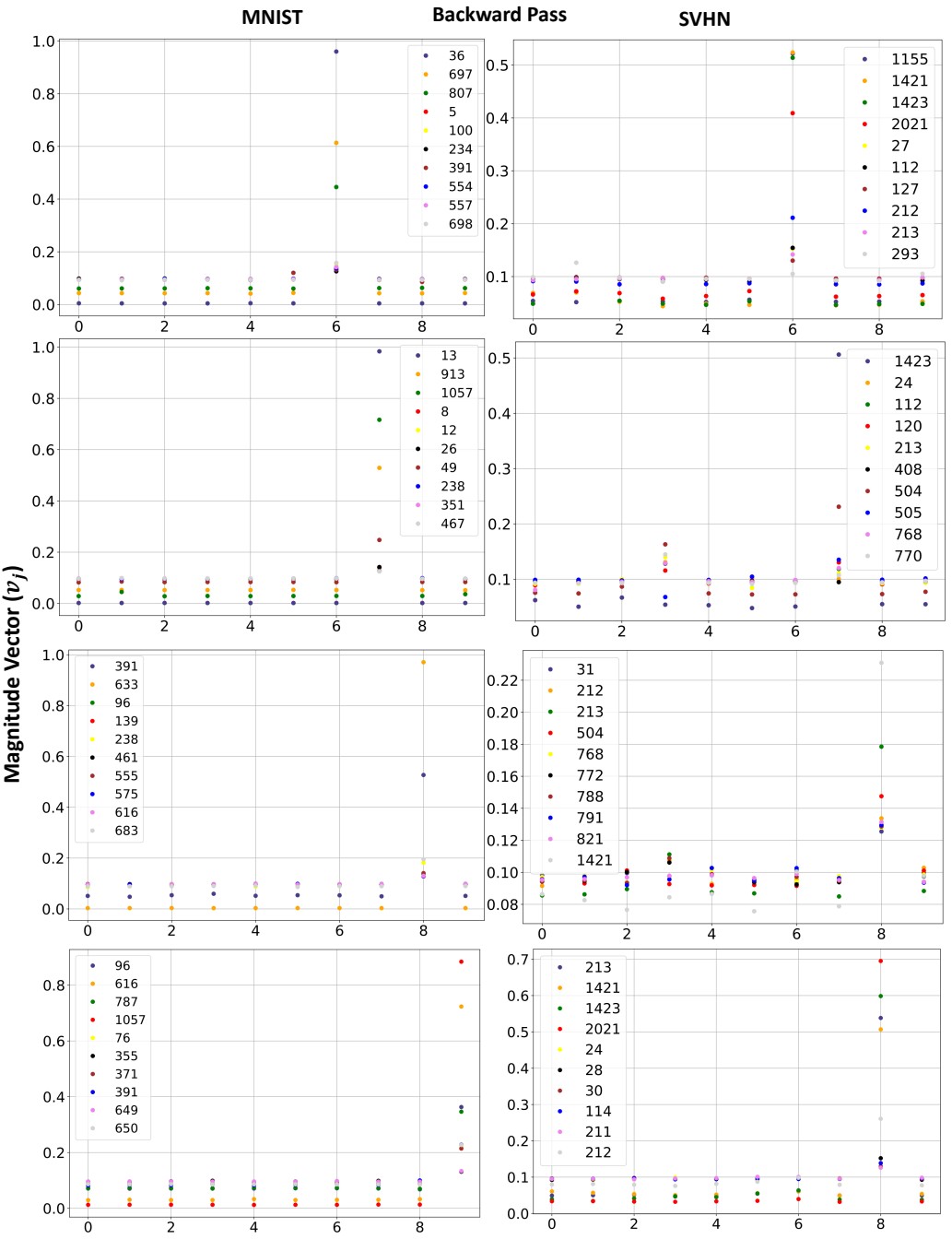

Figure 26: Vector Magnitude ($v_j$) for the selected $k$ units from the *backward* path

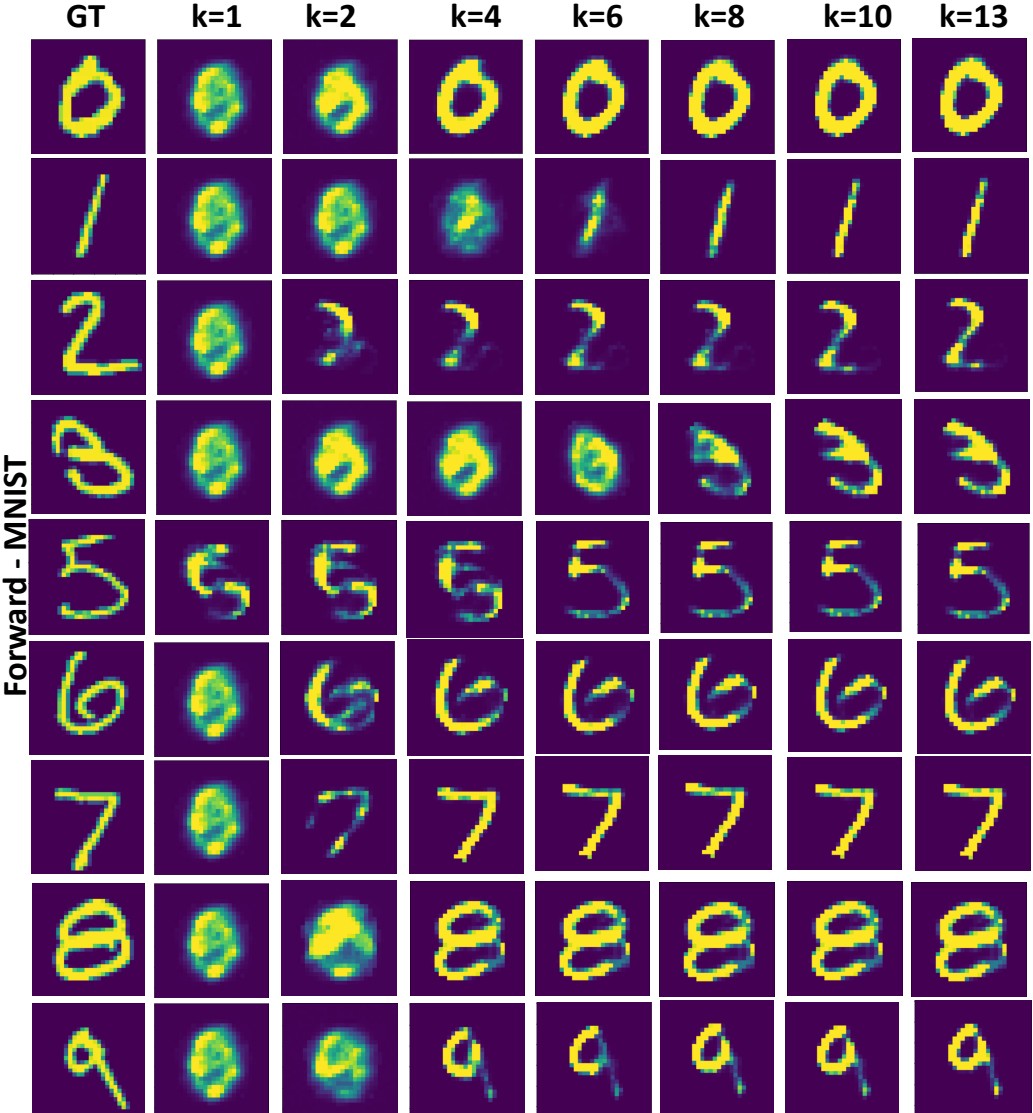

Figure 27: Reconstructed inputs when only a small amount *k* of units in the *PC* layer are considered

Table 8: Quantitative analysis of RMA scores obtained by **aggregating** the top-*k* (5, 15, 30, and 200) the relevant units across all datasets over the threshold of 0.1.

| Class | MNIST | | | | PASCAL | | | | SVHN | | | | CelebA | | | |
|---|---|---|---|---|---|---|---|---|---|---|---|---|---|---|---|---|
| | 5 | 15 | 30 | 200 | 5 | 15 | 30 | 200 | 5 | 15 | 30 | 200 | 5 | 15 | 30 | 200 |
| 0 | 35 ± 5 | 34 ± 5 | 34 ± 4 | 34 ± 5 | 46 ± 5 | 45 ± 4 | 45 ± 4 | 46 ± 6 | 60 ± 8 | 59 ± 7 | 59 ± 7 | 59 ± 8 | 45 ± 5 | 43 ± 6 | 43 ± 7 | 43 ± 6 |
| 1 | 39 ± 5 | 37 ± 6 | 37 ± 6 | 38 ± 7 | 48 ± 6 | 47 ± 5 | 47 ± 4 | 47 ± 4 | 54 ± 7 | 53 ± 6 | 53 ± 6 | 53 ± 6 | 44 ± 5 | 43 ± 5 | 44 ± 4 | 43 ± 5 |
| 2 | 34 ± 4 | 32 ± 5 | 33 ± 4 | 32 ± 5 | 48 ± 4 | 47 ± 4 | 46 ± 4 | 47 ± 5 | 63 ± 7 | 62 ± 6 | 62 ± 5 | 62 ± 5 | | | | |
| 3 | 40 ± 3 | 39 ± 3 | 38 ± 5 | 38 ± 5 | 44 ± 4 | 42 ± 3 | 42 ± 4 | 42 ± 2 | 61 ± 8 | 61 ± 7 | 60 ± 7 | 60 ± 6 | | | | |
| 4 | 33 ± 4 | 33 ± 4 | 31 ± 3 | 32 ± 5 | 45 ± 5 | 44 ± 4 | 44 ± 4 | 45 ± 3 | 60 ± 7 | 58 ± 5 | 57 ± 8 | 58 ± 6 | | | | |
| 5 | 32 ± 3 | 31 ± 3 | 30 ± 4 | 30 ± 5 | 41 ± 5 | 41 ± 4 | 40 ± 5 | 41 ± 4 | 62 ± 8 | 62 ± 7 | 61 ± 7 | 62 ± 6 | | | | |
| 6 | 26 ± 4 | 28 ± 4 | 26 ± 4 | 27 ± 5 | 41 ± 5 | 40 ± 4 | 40 ± 4 | 40 ± 5 | 62 ± 7 | 61 ± 6 | 61 ± 6 | 61 ± 5 | | | | |
| 7 | 30 ± 4 | 30 ± 5 | 28 ± 4 | 29 ± 5 | 45 ± 5 | 45 ± 4 | 44 ± 5 | 44 ± 4 | 59 ± 9 | 57 ± 7 | 57 ± 6 | 57 ± 5 | | | | |
| 8 | 32 ± 4 | 30 ± 5 | 29 ± 4 | 30 ± 5 | 46 ± 8 | 46 ± 5 | 45 ± 4 | 45 ± 4 | 61 ± 7 | 61 ± 6 | 60 ± 6 | 60 ± 6 | | | | |
| 9 | 28 ± 4 | 25 ± 4 | 27 ± 3 | 26 ± 5 | 45 ± 9 | 44 ± 8 | 44 ± 6 | 44 ± 5 | 64 ± 10 | 60 ± 9 | 60 ± 8 | 60 ± 6 | | | | |
| Avg | 33 ± 4 | 32 ± 4 | 31 ± 4 | 31 ± 5 | 45 ± 6 | 44 ± 5 | 44 ± 4 | 44 ± 4 | 61 ± 8 | 61 ± 7 | 59 ± 7 | 55 ± 10 | 45 ± 5 | 43 ± 6 | 44 ± 6 | 43 ± 6 |

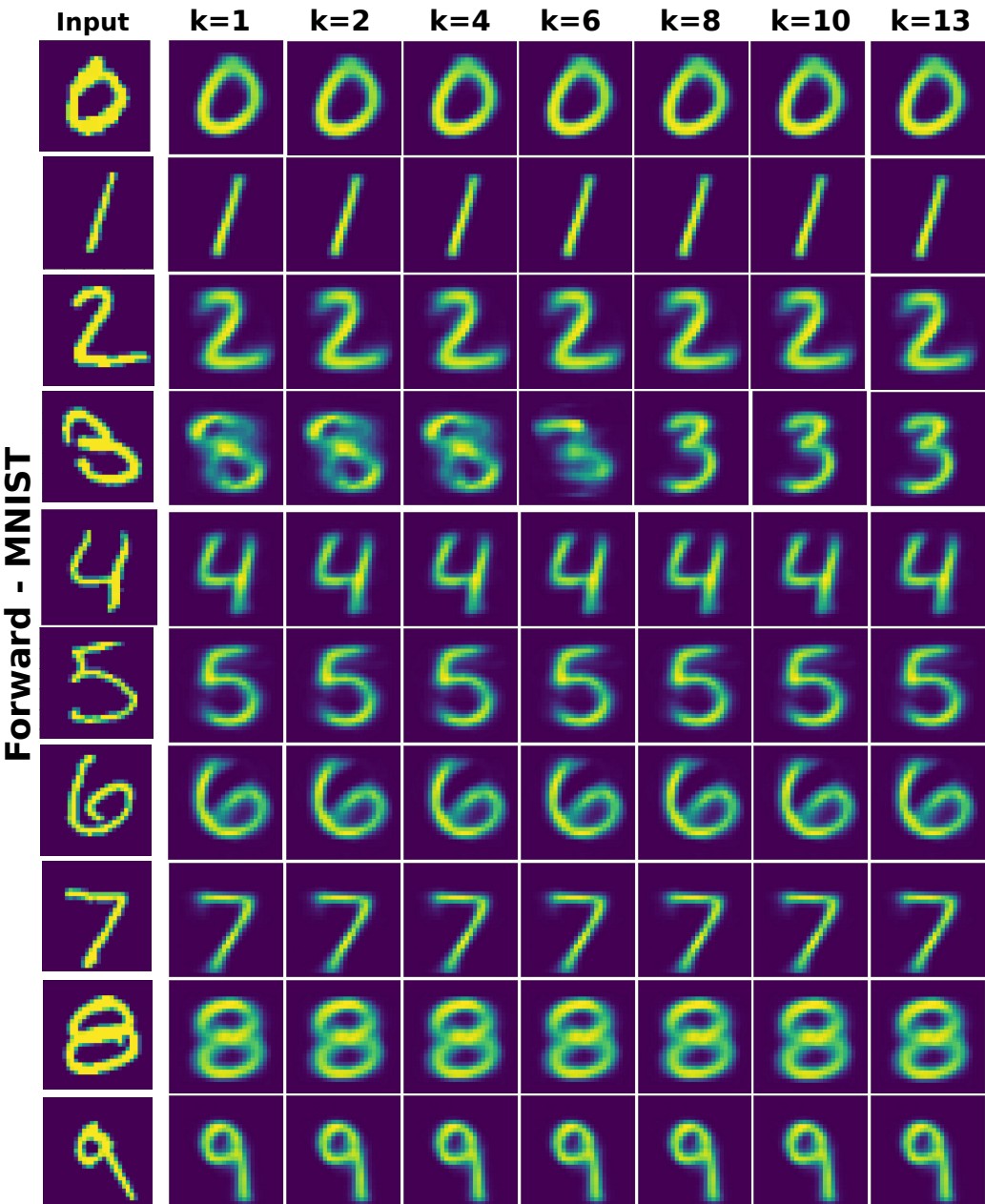

Figure 28: Reconstructed inputs when only a small amount *k* of units in the *PC* layer are considered (EM Routing)

Table 9: Quantitative analysis of RMA scores obtained by **aggregating** the top-*k* (5, 15, 30, and 200) the relevant units across all datasets over the threshold of 0.25.

| Class | MNIST | | | | PASCAL | | | | SVHN | | | | CelebA | | | |
|---|---|---|---|---|---|---|---|---|---|---|---|---|---|---|---|---|
| | 5 | 15 | 30 | 200 | 5 | 15 | 30 | 200 | 5 | 15 | 30 | 200 | 5 | 15 | 30 | 200 |
| 0 | 14 ± 5 | 13 ± 4 | 13 ± 4 | 14 ± 4 | 17 ± 4 | 17 ± 4 | 17 ± 4 | 17 ± 4 | 24 ± 8 | 23 ± 7 | 24 ± 8 | 23 ± 7 | 19 ± 5 | 18 ± 5 | 19 ± 5 | 12 ± 3 |
| 1 | 14 ± 5 | 14 ± 5 | 14 ± 5 | 14 ± 4 | 17 ± 4 | 18 ± 4 | 18 ± 5 | 17 ± 4 | 21 ± 7 | 21 ± 7 | 20 ± 7 | 22 ± 8 | 17 ± 4 | 18 ± 4 | 18 ± 4 | 12 ± 3 |
| 2 | 13 ± 4 | 13 ± 4 | 13 ± 4 | 13 ± 4 | 16 ± 2 | 17 ± 3 | 14 ± 4 | 17 ± 2 | 24 ± 8 | 23 ± 7 | 23 ± 6 | 27 ± 7 | | | | |
| 3 | 14 ± 4 | 14 ± 4 | 13 ± 5 | 14 ± 4 | 14 ± 2 | 15 ± 2 | 15 ± 4 | 14 ± 2 | 25 ± 7 | 24 ± 7 | 25 ± 7 | 25 ± 7 | | | | |
| 4 | 14 ± 4 | 12 ± 3 | 11 ± 3 | 12 ± 3 | 17 ± 3 | 17 ± 3 | 17 ± 4 | 17 ± 2 | 25 ± 7 | 24 ± 6 | 25 ± 7 | 25 ± 7 | | | | |
| 5 | 12 ± 3 | 11 ± 4 | 12 ± 4 | 12 ± 3 | 15 ± 4 | 16 ± 4 | 17 ± 4 | 17 ± 3 | 26 ± 7 | 26 ± 7 | 26 ± 7 | 26 ± 7 | | | | |
| 6 | 9 ± 3 | 9 ± 4 | 9 ± 3 | 10 ± 3 | 15 ± 4 | 14 ± 3 | 16 ± 4 | 17 ± 0.09 | 25 ± 6 | 26 ± 6 | 26 ± 7 | 26 ± 7 | | | | |
| 7 | 11 ± 3 | 11 ± 3 | 11 ± 3 | 11 ± 3 | 16 ± 5 | 16 ± 4 | 17 ± 5 | 17 ± 2 | 24 ± 4 | 24 ± 6 | 24 ± 7 | 25 ± 7 | | | | |
| 8 | 12 ± 4 | 13 ± 3 | 13 ± 3 | 12 ± 3 | 17 ± 5 | 17 ± 4 | 15 ± 4 | 16 ± 4 | 25 ± 8 | 25 ± 7 | 25 ± 8 | 25 ± 6 | | | | |
| 9 | 11 ± 3 | 11 ± 3 | 11 ± 3 | 11 ± 3 | 16 ± 5 | 16 ± 6 | 15 ± 7 | 16 ± 5 | 25 ± 8 | 25 ± 7 | 25 ± 8 | 25 ± 7 | | | | |
| Avg | 12 ± 4 | 12 ± 4 | 12 ± 4 | 12 ± 3 | 16 ± 4 | 16 ± 4 | 16 ± 5 | 16 ± 3 | 24 ± 7 | 24 ± 7 | 24 ± 7 | 25 ± 7 | 18 ± 5 | 18 ± 5 | 19 ± 5 | 12 ± 3 |

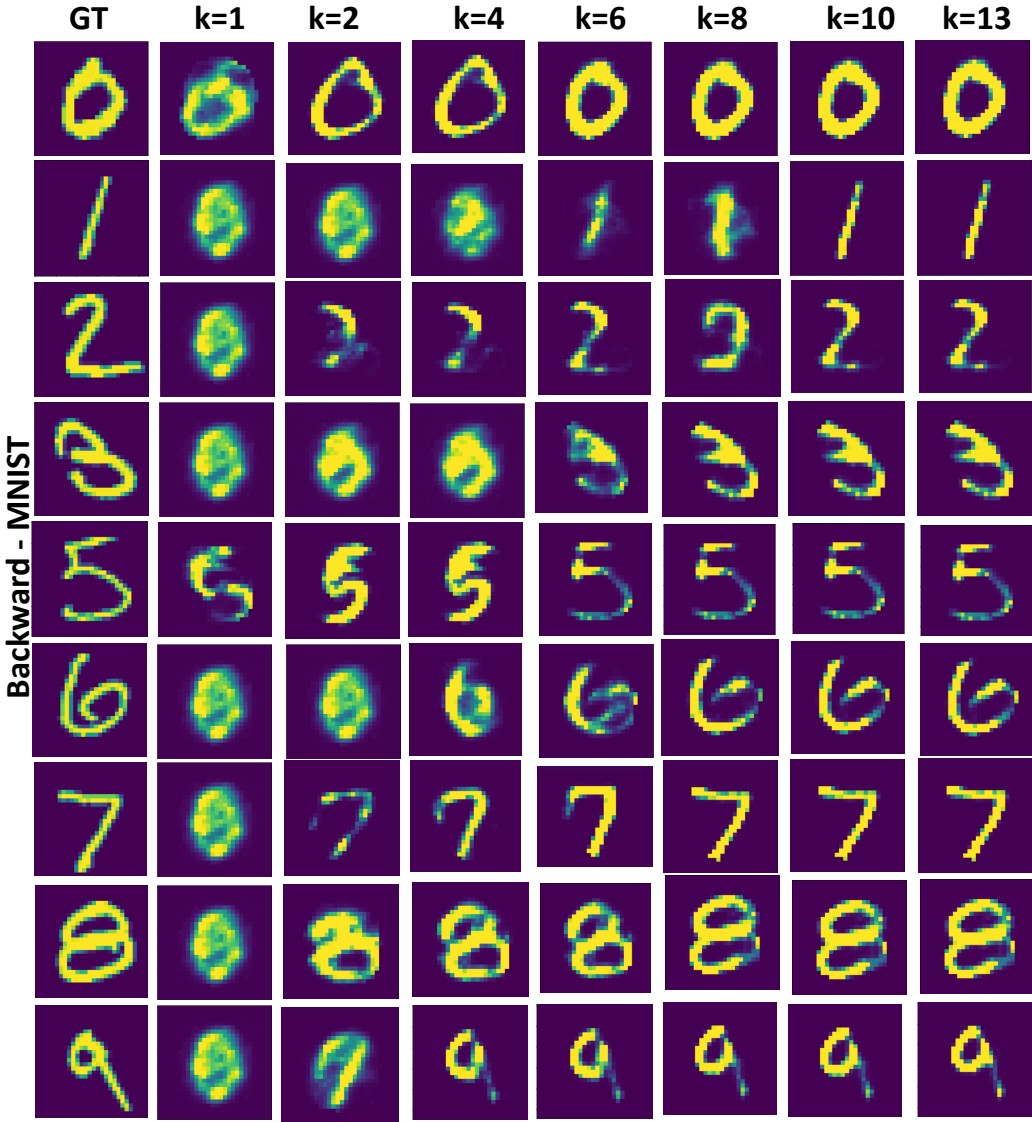

Figure 29: Reconstructed inputs when only a small amount *k* of units in the *PC* layer are considered (Dynamic Routing)

Table 10: Quantitative analysis of RMA scores obtained by **aggregating** the top-*k* (5, 15, 30, and 200) the relevant units across all datasets over the threshold of 0.5.

| Class | MNIST | | | | PASCAL | | | | SVHN | | | | CelebA | | | |
|---|---|---|---|---|---|---|---|---|---|---|---|---|---|---|---|---|
| | 5 | 15 | 30 | 200 | 5 | 15 | 30 | 200 | 5 | 15 | 30 | 200 | 5 | 15 | 30 | 200 |
| 0 | 3 ± 6 | 3 ± 5 | 3 ± 5 | 3 ± 6 | 3 ± 2 | 3 ± 3 | 3 ± 3 | 3 ± 3 | 6 ± 3 | 6 ± 3 | 5 ± 3 | 5 ± 3 | 3 ± 2 | 3 ± 2 | 3 ± 1 | 3 ± 1 |
| 1 | 3 ± 3 | 3 ± 5 | 3 ± 7 | 3 ± 7 | 3 ± 2 | 3 ± 2 | 3 ± 2 | 3 ± 1 | 5 ± 3 | 5 ± 3 | 5 ± 2 | 5 ± 2 | 4 ± 2 | 4 ± 2 | 3 ± 2 | 3 ± 1 |
| 2 | 3 ± 5 | 3 ± 5 | 3 ± 6 | 3 ± 6 | 3 ± 2 | 3 ± 1 | 2 ± 1 | 2 ± 0.8 | 6 ± 3 | 6 ± 2 | 5 ± 2 | 5 ± 2 | | | | |
| 3 | 3 ± 5 | 3 ± 4 | 3 ± 5 | 3 ± 6 | 3 ± 1 | 3 ± 2 | 3 ± 2 | 3 ± 2 | 6 ± 3 | 6 ± 3 | 5 ± 3 | 5 ± 3 | | | | |
| 4 | 3 ± 2 | 3 ± 3 | 3 ± 3 | 3 ± 4 | 3 ± 2 | 3 ± 3 | 3 ± 5 | 3 ± 7 | 6 ± 3 | 6 ± 3 | 6 ± 2 | 6 ± 2 | | | | |
| 5 | 3 ± 2 | 3 ± 3 | 3 ± 5 | 3 ± 5 | 3 ± 2 | 3 ± 2 | 4 ± 1 | 4 ± 0.05 | 6 ± 2 | 6 ± 3 | 6 ± 3 | 6 ± 3 | | | | |
| 6 | 3 ± 2 | 3 ± 1 | 3 ± 1 | 3 ± 1 | 3 ± 1 | 3 ± 1 | 3 ± 1 | 3 ± 0.04 | 6 ± 4 | 6 ± 3 | 6 ± 3 | 6 ± 3 | | | | |
| 7 | 3 ± 2 | 3 ± 2 | 3 ± 1 | 3 ± 1 | 3 ± 1 | 3 ± 1 | 3 ± 1 | 3 ± 1 | 6 ± 3 | 6 ± 3 | 6 ± 2 | 6 ± 2 | | | | |
| 8 | 3 ± 4 | 3 ± 2 | 3 ± 3 | 3 ± 2 | 3 ± 1 | 3 ± 1 | 3 ± 1 | 3 ± 1 | 6 ± 3 | 6 ± 3 | 6 ± 2 | 6 ± 2 | | | | |
| 9 | 3 ± 3 | 3 ± 2 | 3 ± 4 | 3 ± 4 | 3 ± 1 | 3 ± 1 | 3 ± 1 | 3 ± 1 | 7 ± 3 | 7 ± 2 | 7 ± 2 | 7 ± 2 | | | | |
| Avg | 3 ± 3 | 3 ± 3 | 3 ± 4 | 3 ± 4 | 3 ± 4 | 3 ± 2 | 3 ± 2 | 3 ± 2 | 6 ± 3 | 6 ± 3 | 6 ± 2 | 7 ± 2 | 2 ± 2 | 4 ± 2 | 3 ± 2 | 3 ± 1 |

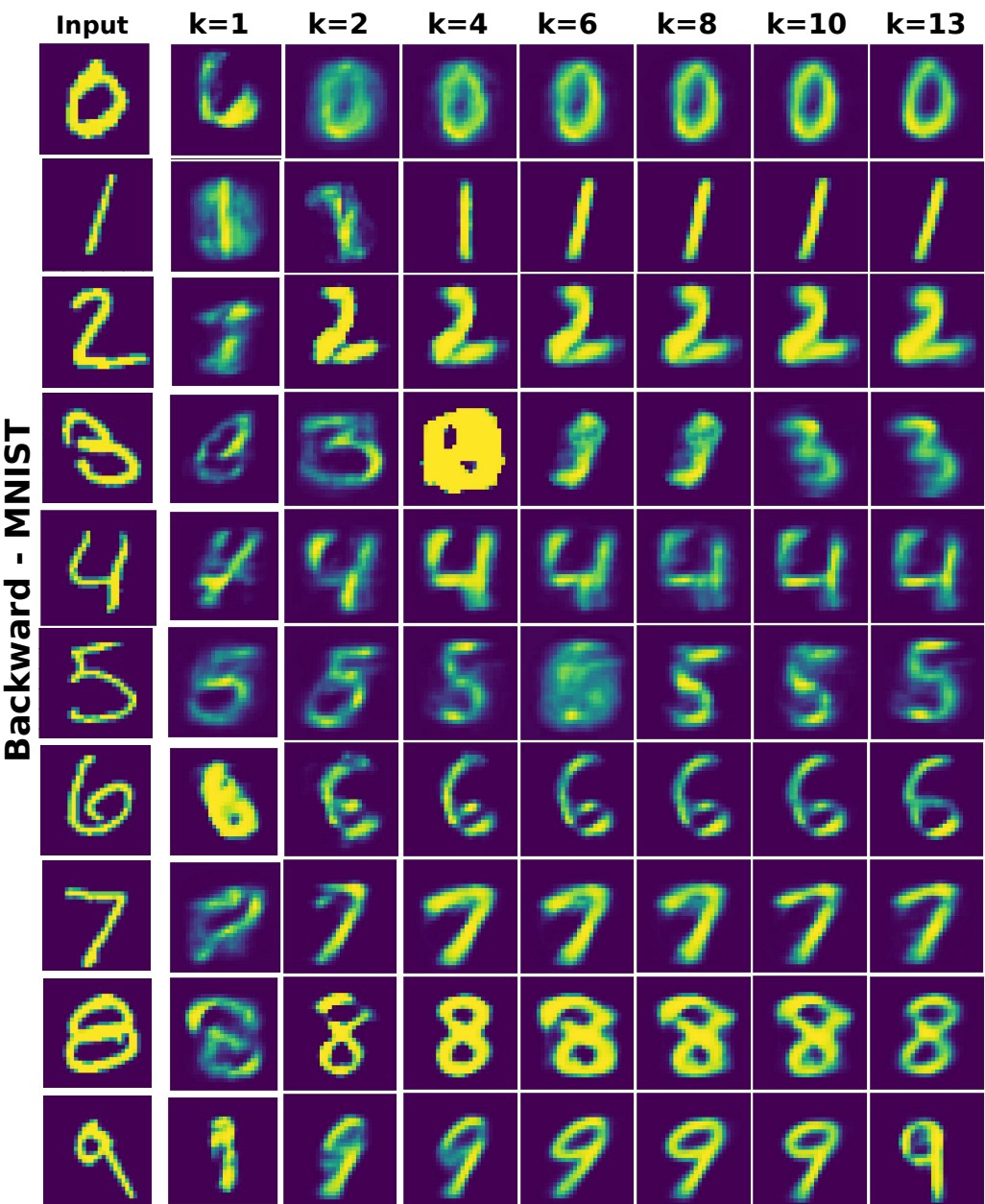

Figure 30: Reconstructed inputs when only a small amount $k$ of units in the *PC* layer are considered (EM Routing)

Table 11: Quantitative analysis of RMA scores obtained by **aggregating** the top-$k$ (5, 15, 30, and 200) the relevant units across all datasets over the threshold of 0.1..

| Class | MNIST | | | | PASCAL | | | | SVHN | | | | CelebA | | | |
|---|---|---|---|---|---|---|---|---|---|---|---|---|---|---|---|---|
| | 5 | 15 | 30 | 200 | 5 | 15 | 30 | 200 | 5 | 15 | 30 | 200 | 5 | 15 | 30 | 200 |
| 0 | 21±12 | 21±13 | 21±13 | 21±13 | - | - | - | - | 42±12 | 42±11 | 42±11 | 42±11 | 42±18 | 40±18 | 41±18 | 41±18 |
| 1 | 24±15 | 24±15 | 42±12 | 24±13 | - | - | - | - | 43±12 | 42±12 | 42±12 | 42±12 | 44±18 | 44±18 | 44±18 | 44±18 |
| 2 | 17±11 | 18±11 | 18±10 | 19±11 | - | - | - | - | 41±12 | 40±12 | 40±12 | 40±12 | | | | |
| 3 | 18±11 | 17±11 | 18±11 | 18±11 | - | - | - | - | 43±12 | 42±12 | 42±11 | 43±11 | | | | |
| 4 | 21±13 | 21±14 | 21±13 | 21±13 | - | - | - | - | 41±12 | 40±12 | 40±12 | 40±12 | | | | |
| 5 | 21±13 | 20±13 | 20±13 | 20±13 | - | - | - | - | 43±11 | 43±11 | 43±11 | 43±11 | | | | |
| 6 | 22±12 | 22±12 | 22±12 | 22±12 | - | - | - | - | 43±13 | 42±13 | 42±13 | 41±13 | | | | |
| 7 | 23±12 | 22±13 | 22±13 | 22±13 | - | - | - | - | 41±13 | 37±12 | 37±12 | 37±12 | | | | |
| 8 | 23±12 | 23±15 | 22±13 | 23±15 | - | - | - | - | 41±12 | 41±12 | 42±11 | 42±12 | | | | |
| 9 | 25±15 | 25±15 | 22±15 | 25±15 | - | - | - | - | 42±12 | 42±12 | 42±12 | 42±12 | | | | |
| Avg | 22±13 | 21±13 | 21±13 | 22±13 | - | - | - | - | 42±11 | 41±12 | 41±12 | 41±12 | 43±18 | 42±18 | 43±18 | 43±18 |

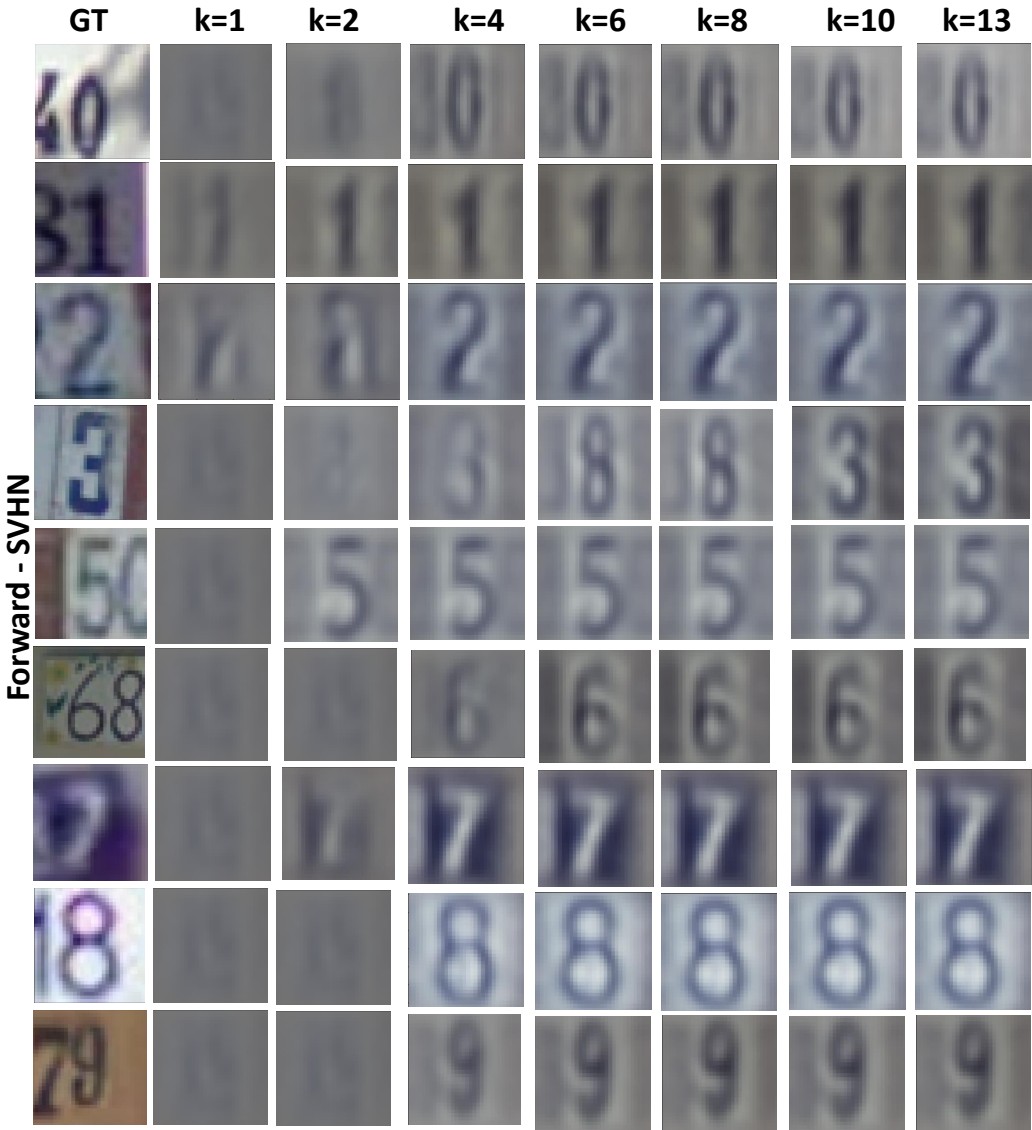

Figure 31: Reconstructed inputs when only a small amount *k* of units in the *PC* layer are considered

Table 12: Quantitative analysis of RMA scores obtained by **aggregating** the top-*k* (5, 15, 30, and 200) the relevant units across all datasets over the threshold of 0.25..

| Class | MNIST 5 | 15 | 30 | 200 | PASCAL 5 | 15 | 30 | 200 | SVHN 5 | 15 | 30 | 200 | CelebA 5 | 15 | 30 | 200 |
|---|---|---|---|---|---|---|---|---|---|---|---|---|---|---|---|---|
| 0 | 7±5 | 8±4 | 7±4 | 7±4 | - | - | - | - | 14±6 | 14±6 | 15±6 | 14±6 | 14±1 | 13±1 | 13±1 | 13±1 |
| 1 | 8±5 | 8±5 | 8±4 | 8±5 | - | - | - | - | 14±6 | 14±6 | 14±6 | 14±6 | 14±1 | 14±1 | 14±1 | 14±1 |
| 2 | 6±3 | 6±3 | 6±3 | 6±4 | - | - | - | - | 14±6 | 14±6 | 13±6 | 13±6 | | | | |
| 3 | 6±3 | 6±3 | 6±4 | 6±4 | - | - | - | - | 15±6 | 14±6 | 15±6 | 14±6 | | | | |
| 4 | 7±4 | 7±4 | 7±4 | 7±5 | - | - | - | - | 14±6 | 13±6 | 14±6 | 14±6 | | | | |
| 5 | 7±5 | 7±4 | 7±4 | 7±5 | - | - | - | - | 15±6 | 14±6 | 13±5 | 15±6 | | | | |
| 6 | 8±4 | 8±4 | 7±4 | 7±4 | - | - | - | - | 15±6 | 13±6 | 14±6 | 14±6 | | | | |
| 7 | 7±4 | 7±4 | 7±4 | 7±5 | - | - | - | - | 15±6 | 12±5 | 13±5 | 12±6 | | | | |
| 8 | 7±5 | 7±5 | 7±4 | 8±5 | - | - | - | - | 15±6 | 14±6 | 14±6 | 14±6 | | | | |
| 9 | 8±5 | 8±4 | 8±3 | 7±4 | - | - | - | - | 14±6 | 14±6 | 14±6 | 14±6 | | | | |
| Avg | 7±4 | 7±4 | 7±4 | 7±5 - | - | - | - | 15±6 | 14±6 | 14±6 | 14±6 | 14±1 | 14±1 | 14±1 | 14±1 | |

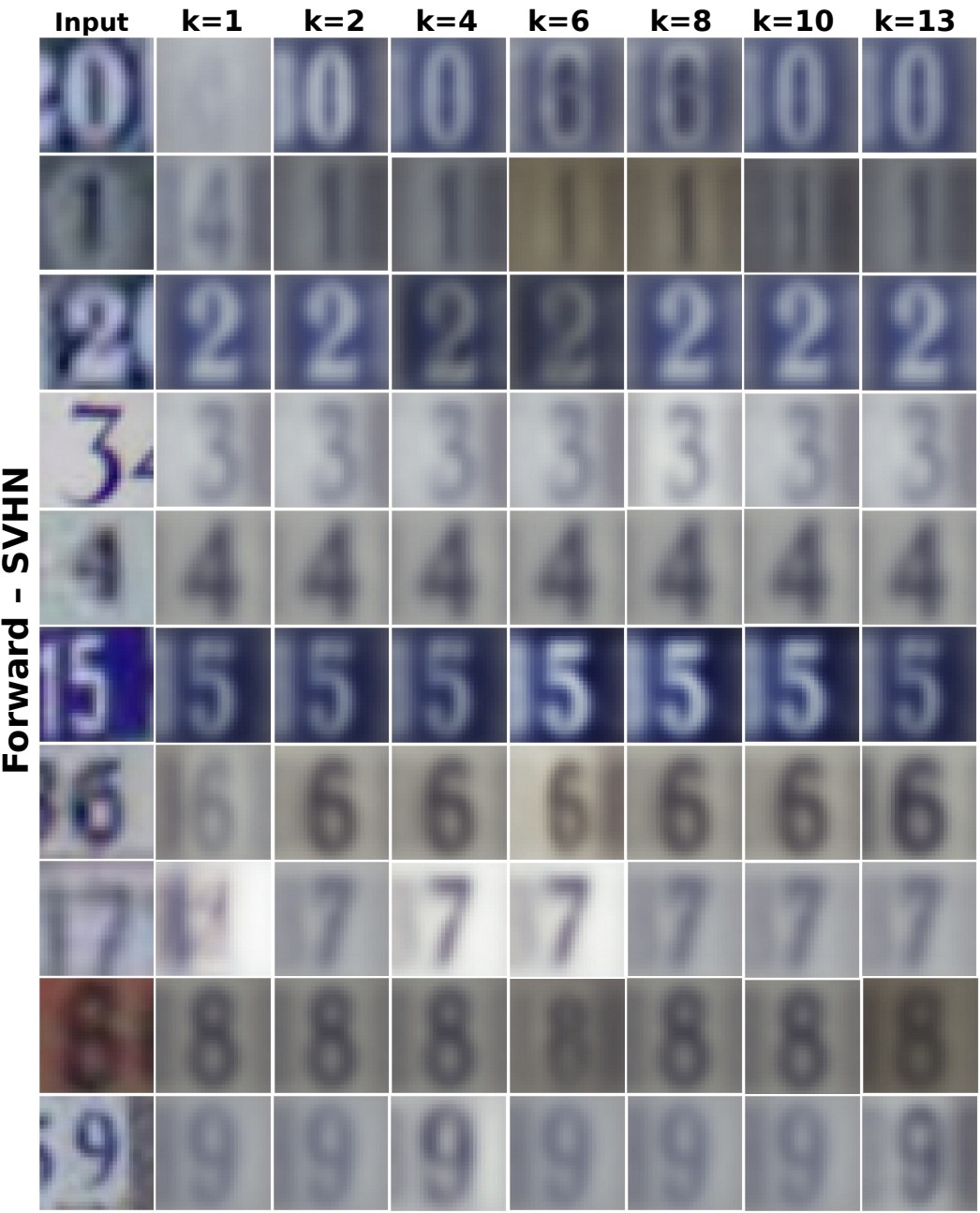

Figure 32: Reconstructed inputs when only a small amount *k* of units in the *PC* layer are considered

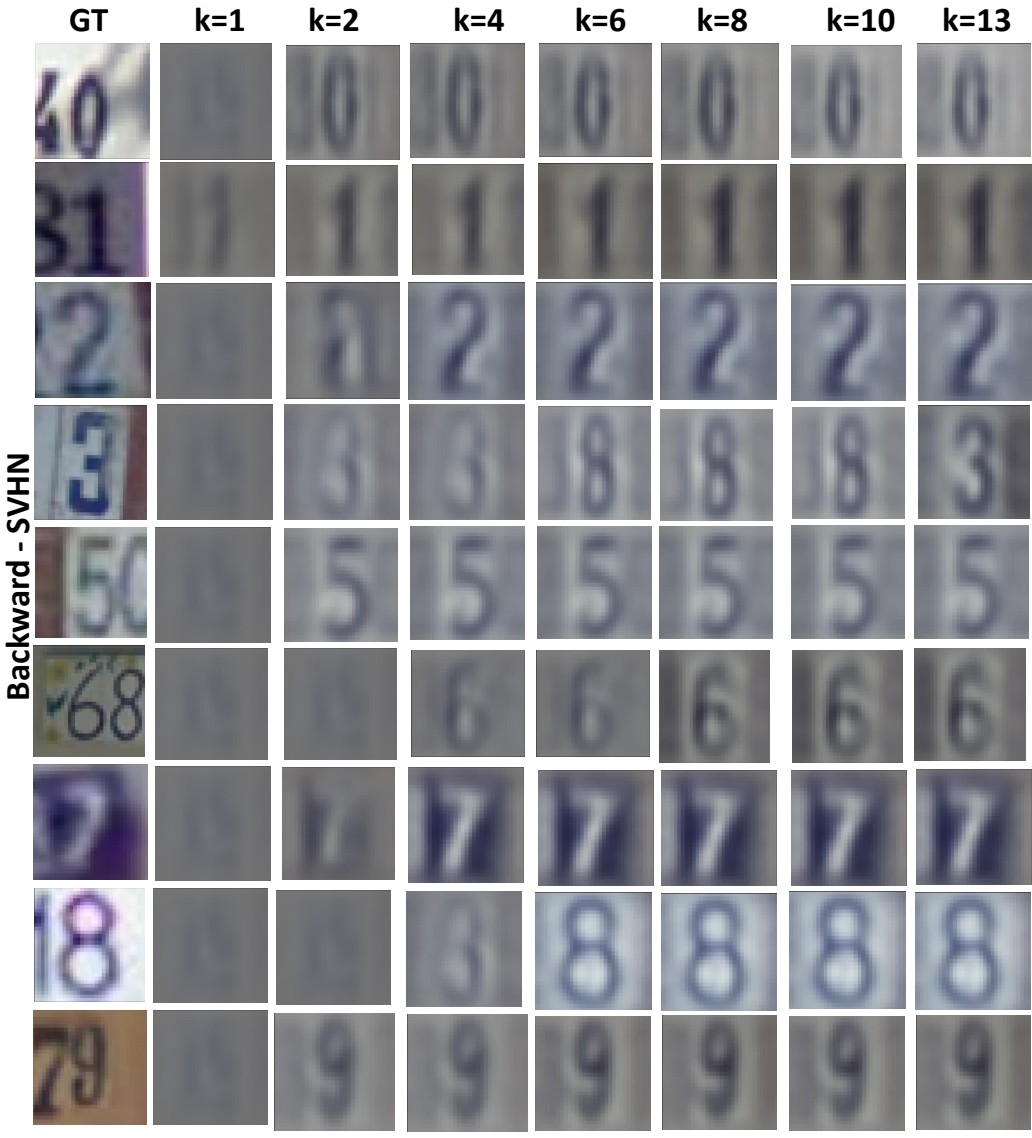

Figure 33: Reconstructed inputs when only a small amount *k* of units in the *PC* layer are considered

Table 13: Quantitative analysis of RMA scores obtained by **aggregating** the top-*k* (5, 15, 30, and 200) the relevant units across all datasets over the threshold of 0.5..

| Class | MNIST 5 | 15 | 30 | 200 | PASCAL 5 | 15 | 30 | 200 | SVHN 5 | 15 | 30 | 200 | CelebA 5 | 15 | 30 | 200 |
|---|---|---|---|---|---|---|---|---|---|---|---|---|---|---|---|---|
| 0 | 2±1 | 2±1 | 2±1 | 2±1 | - | - | - | - | 3±1 | 3±1 | 3±2 | 3±2 | 2±2 | 2±2 | 2±2 | 3±2 |
| 1 | 2±1 | 2±2 | 2±2 | 2±1 | - | - | - | - | 3±2 | 3±2 | 3±2 | 3±2 | 3±2 | 3±2 | 3±2 | 3±2 |
| 2 | 1±0.1 | 2±0.1 | 2±0.1 | 2±0.1 | - | - | - | - | 3±2 | 3±2 | 3±2 | 3±2 | | | | |
| 3 | 2±0.1 | 2±1 | 2±1 | 2±0.1 | - | - | - | - | 3±2 | 3±2 | 3±2 | 3±2 | | | | |
| 4 | 2±1 | 2±1 | 2±0.1 | 2±1 | - | - | - | - | 3±1 | 2±1 | 3±1 | 2±1 | | | | |
| 5 | 2±1 | 2±1 | 1±1 | 2±1 | - | - | - | - | 3±2 | 3±2 | 3±2 | 3±2 | | | | |
| 6 | 2±1 | 2±1 | 2±1 | 2±1 | - | - | - | - | 3±2 | 3±2 | 3±2 | 3±1 | | | | |
| 7 | 2±1 | 2±1 | 2±1 | 2±1 | - | - | - | - | 3±1 | 3±1 | 3±1 | 3±2 | | | | |
| 8 | 2±1 | 2±1 | 2±1 | 2±2 | - | - | - | - | 3±2 | 3±2 | 3±2 | 3±2 | | | | |
| 9 | 1±1 | 2±1 | 2±1 | 2±1 | - | - | - | - | 3±1 | 3±1 | 3±2 | 3±2 | | | | |
| Avg | 2±1 | 2±1 | 2±1 | 2±1 | - | - | - | - | 3±2 | 3±2 | 3±2 | 3±2 | 3±2 | 3±2 | 3±2 | 3±2 |

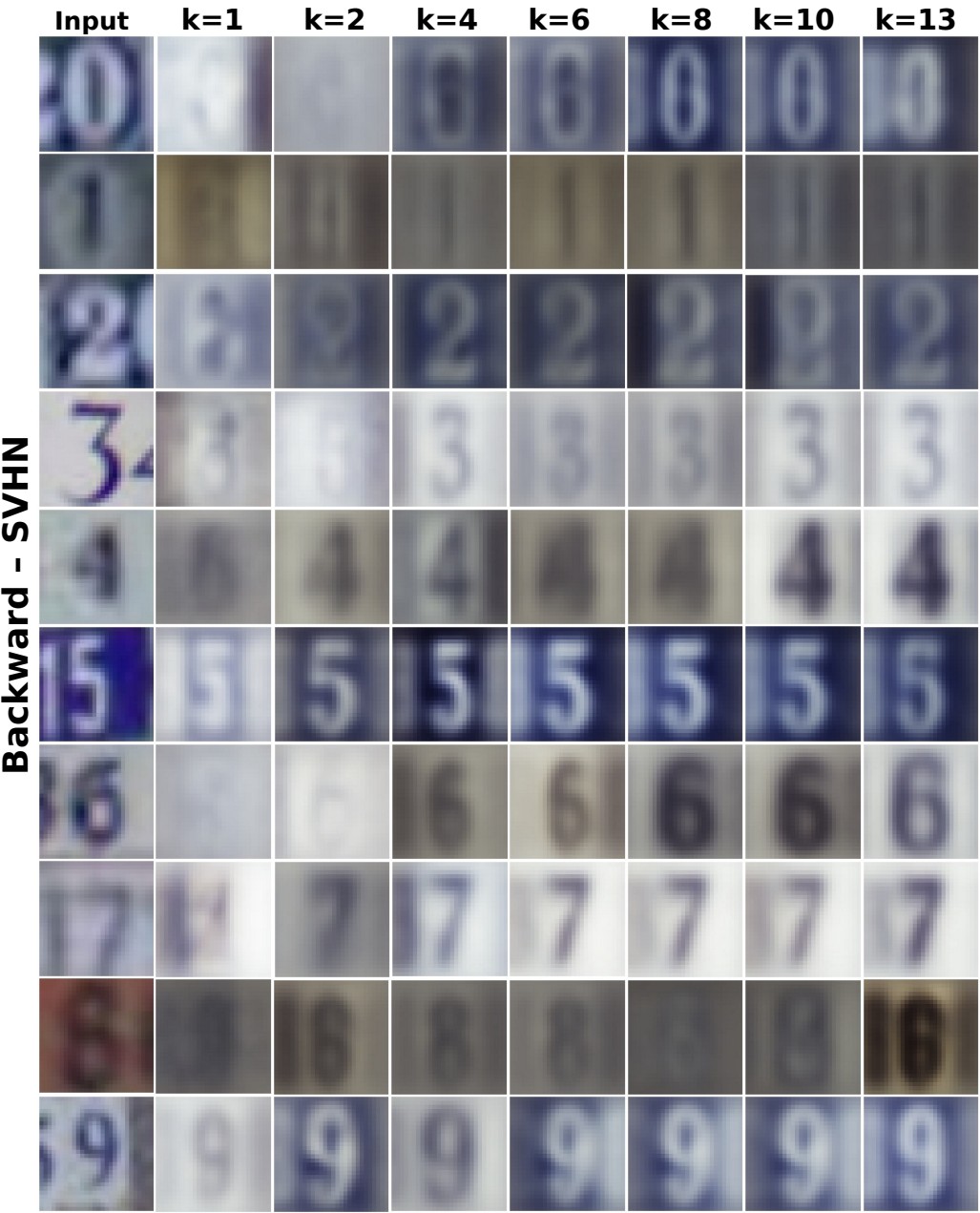

Figure 34: Reconstructed inputs when only a small amount *k* of units in the *PC* layer are considered

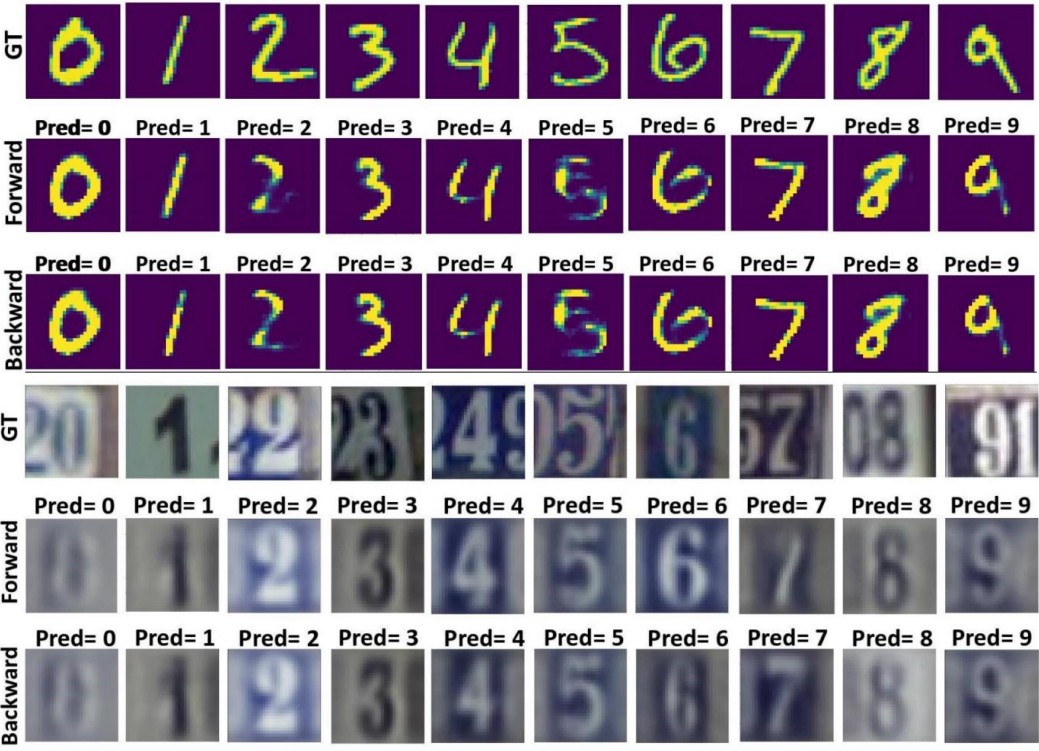

Figure 35: Reconstructed inputs when only relevant units of the network ($k$=35 *Conv* and $k$=10 *PC*) are considered.

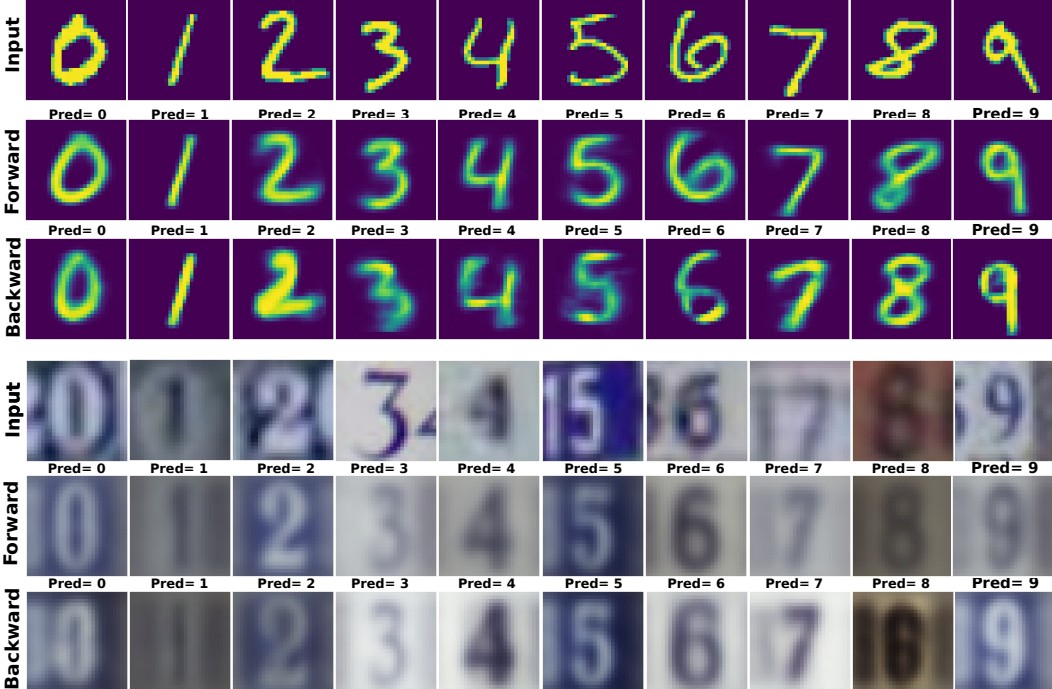

Figure 36: Reconstructed inputs when only relevant units of the network ($k$=35 *Conv* and $k$=10 *PC*) are considered.

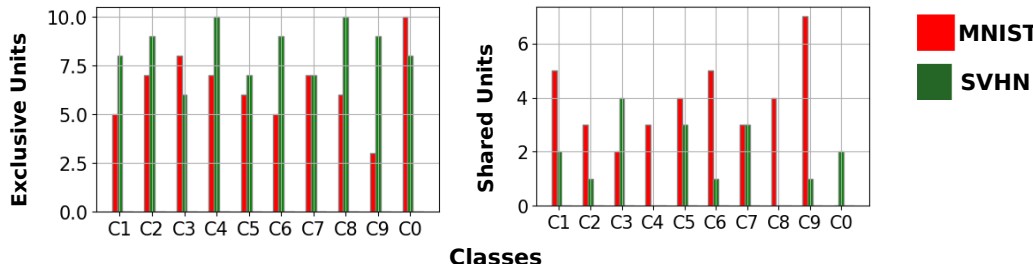

Figure 37: Number of top-$k$ units exclusive for each class (left) and shared w.r.t all the other classes (right).

---

**Algorithm 7** Measuring *Part-Whole* Relationship Encodings

---

0: $x_m \leftarrow$ input example
0: $k \leftarrow k$ relevant units
0: RMA Relevance Mass Accuracy
0: **while** $x_m$ in *dataset* **do**
0:     only $k$=200 or $k$=1000 capsule selected
0:     push the changes to find $\hat{y}$ from CapsNet(only $k$=1000)
0:     $e_{ij} \leftarrow$ CapsNet($\hat{y}$)
0:     $e_{ij} \leftarrow$ ranked($e_{ij}$)
0:     select relevant units ($k$=200)
0:     Produce heatmap of each unit ($h^l$)
0:     min&max values from all the heatmaps used to Normalize the heatmap
0:     Produce heatmap of $CC$ layer ($h^{l+1}$)
0:     computes RMA($h^l$ and $h^{l+1}$)
0:     **if** RMA < threshold **then**
0:         visualize(heatmap on top of example)
0:     **else**
0:         ignore them
0:     **end if**
0: **end while** =0

---

Table 14: Quantitative analysis of RMA scores obtained by top-$k$ (1) the relevant units separately across all datasets over different thresholds (EM routing)..

| Class | MNIST 0.1 | 0.25 | 0.5 | PASCAL 0.1 | 0.25 | 0.5 | SVHN 0.1 | 0.25 | 0.5 | CelebA 0.1 | 0.25 | 0.5 |
|---|---|---|---|---|---|---|---|---|---|---|---|---|
| 0 | 26±16 | 9±7 | 3±2 | - | - | - | 51±17 | 18±9 | 4±3 | 48±21 | 17±11 | 4±3 |
| 1 | 24±15 | 8±5 | 2±1 | - | - | - | 51±16 | 19±9 | 4±3 | 56±22 | 21±13 | 4±3 |
| 2 | 18±11 | 6±4 | 1±0.09 | - | - | - | 50±17 | 21±10 | 4±3 | | | |
| 3 | 18±11 | 6±3 | 2±0.09 | - | - | - | 53±18 | 18±9 | 5±3 | | | |
| 4 | 26±15 | 9±6 | 2±2 | - | - | - | 49±14 | 17±10 | 4±3 | | | |
| 5 | 27±17 | 9±7 | 2±1 | - | - | - | 52±16 | 18±9 | 4±3 | | | |
| 6 | 22±12 | 6±4 | 2±1 | - | - | - | 59±16 | 20±10 | 4±3 | | | |
| 7 | 22±13 | 9±6 | 3±2 | - | - | - | 51±17 | 19±9 | 4±3 | | | |
| 8 | 26±16 | 11±8 | 2±2 | - | - | - | 50±16 | 19±10 | 4±3 | | | |
| 9 | 31±18 | 7±4 | 3±2 | - | - | - | 51±15 | 19±9 | 4±4 | | | |
| Avg | 22±14 | 8±5 | 2±1 | - | - | - | 52±16 | 19±9 | 4±3 | 52±21 | 19±12 | 4±3 |

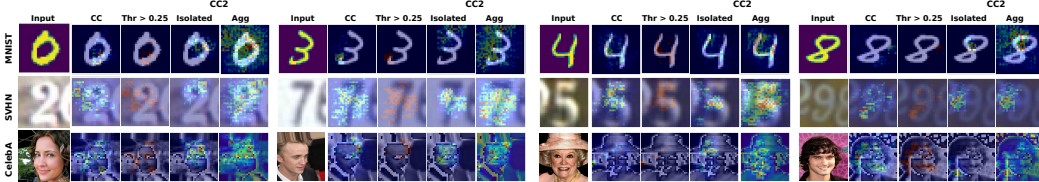

Figure 38: Qualitative examples of the responses considered for the computation of relevance mass accuracy (RMA) with 0.25 threshold (Thr) (EM routing). The CC2 refers to ConvCaps2 in EM routing experiments.

Table 15: RMA scores obtained by the top-200 relevant units when analyzed separately (left - DR and EMR) and aggregated (right - DR and EMR) over different thresholds (DR: dynamic routing and EMR: EM routing).

| | DR | | | | | | EMR | | | | | |
|---|---|---|---|---|---|---|---|---|---|---|---|---|
| | Isolated | | | Aggregated | | | Isolated | | | Aggregated | | |
| Dataset/Thr | 0.1 | 0.25 | 0.5 | 0.1 | 0.25 | 0.5 | 0.1 | 0.25 | 0.5 | 0.1 | 0.25 | 0.5 |
| MNIST | $31\pm6$ | $12\pm4$ | $3\pm1$ | $31\pm5$ | $12\pm3$ | $3\pm4$ | $22\pm14$ | $8\pm5$ | $2\pm1$ | $22\pm13$ | $7\pm5$ | $2\pm1$ |
| PASCAL | $45\pm5$ | $18\pm4$ | $3\pm1$ | $45\pm4$ | $16\pm3$ | $3\pm2$ | - | - | - | - | - | - |
| SVHN | $60\pm10$ | $26\pm8$ | $6\pm3$ | $55\pm10$ | $25\pm7$ | $7\pm2$ | $52\pm16$ | $19\pm9$ | $4\pm3$ | $41\pm12$ | $14\pm6$ | $3\pm2$ |
| CelebA | $44\pm7$ | $18\pm5$ | $4\pm2$ | $33\pm6$ | $12\pm3$ | $3\pm1$ | $52\pm21$ | $19\pm12$ | $4\pm3$ | $43\pm18$ | $14\pm1$ | $3\pm2$ |

