# OpenReview forum: "Towards the Characterization of Representations Learned via Capsule-based Network Architectures"
_ICLR.cc/2024/Conference — Submitted to ICLR 2024_

### Official Review · Reviewer_nJf8 · 2023-10-23

**Soundness:** 3 good
**Presentation:** 3 good
**Contribution:** 3 good
**Rating:** 6
**Confidence:** 2

**Summary:**

The paper focuses on Capsule Networks (CapsNets), which have been reintroduced as a compact and interpretable alternative to deep neural networks. While previous research has highlighted their compression capabilities, this paper aims to conduct a systematic and principled study to assess the interpretability properties of CapsNets, specifically examining the encoding of part-whole relationships within learned representations.

To evaluate interpretability, the authors analyze several capsule-based architectures using MNIST, SVHN, PASCAL-part, and CelebA datasets. The findings suggest that the representations encoded in CapsNets may not be as disentangled or strictly related to part-whole relationships as commonly claimed in the literature.

The contributions of the paper lie in conducting a thorough and rigorous investigation into the interpretability of CapsNets. By challenging the prevailing notion of highly disentangled representations and strict part-whole relationships, the authors provide valuable insights that facilitate a better understanding of the limitations and characteristics of CapsNets as interpretability-focused models.

**Strengths:**

+ The paper stands out for conducting a systematic and principled study to evaluate the interpretability properties of Capsule Networks (CapsNets).

+ The authors analyze multiple datasets (MNIST, SVHN, PASCAL-part, CelebA) and employ various capsule-based architectures, providing a comprehensive evaluation of interpretability in CapsNets. This extensive analysis strengthens the robustness of their conclusions and allows for a broader understanding of the limitations in terms of part-whole relationships.

**Weaknesses:**

- The study focuses on a specific type of neural network architecture (CapsNets) and evaluates interpretability properties on a limited set of datasets (MNIST, SVHN, PASCAL-part, CelebA). Maybe large-scale datasets should be also considered, such as ImageNet.

**Questions:**

Please refer to paper weakness.

---

> ### Author Response · Authors · 2023-11-17
> **Experiments on the small-scale dataset**
>
> In the literature, MNIST, CIFAR10, affNIST, or SVHN are the datasets frequently used by the community for conducting research on CapsNets-based architectures. Therefore, we opted to conduct our analysis on a variety of these datasets so that our method and reported results could compared against by future methods. Beyond these standard datasets, we conducted experiments on CelebA and PASCAL datasets which are considered much complex in comparison with MNIST and CIFAR10.
>
> At this point, conducting experiments on ImageNet or similar large-scale datasets will only contribute to indicate whether the observations made in the paper remain on such big data settings. Due to the large scale of these datasets, the density of our analysis, our reduced computational resources and the limited discussion period, conducting an experiment on ImageNet might not be feasible (even partially). We will attempt to conduct an experiment on the CIFAR100 dataset, which possesses a significantly larger number of classes when compared to the datasets considered in the paper and do our best to provide results during the discussion period. We believe this should provide insights on the observations made in our work under a larger scale dataset.

---

> > ### Comment · Reviewer_nJf8 · 2023-11-20
> > **Experiments on the small-scale dataset**
> >
> > Thank your for the response. Please provide the experimental results on CIFAR100 and I will keep the original score.

---

### Official Review · Reviewer_66ge · 2023-10-24

**Soundness:** 3 good
**Presentation:** 2 fair
**Contribution:** 2 fair
**Rating:** 6
**Confidence:** 4

**Summary:**

This work introduces a principled approach for assessing the properties of capsule networks, focused around the investigation of relevant units in the network. This serves as an initial investigation in the lacking related literature concerning CapsNets. Qualitative and quantitative evaluations on benchmark datasets (MNIST, SVHN, PASCAL-Part, CelebA and CelebAMask-HQ) reveal a potential entanglement of the emerging representations, contradicting previous findings/claims in the related CapsNet literature.

**Strengths:**

This work considers an analysis of the disentanglement of the representations (features) in the context of CapsNets. To the best of my knowledge, this constitutes one of the first attempts towards analysis of CapsNets in this setting. The motivation is clear and the considered setting is sound. This approach introduces an appropriate interval for pertubation analysis via first-order statistics The authors consider a variety of standard benchmark architecture, similar to related recent methods, and assess different configurations and settings.

**Weaknesses:**

Due to the fact that the authors consider multiple settings and configurations, some parts of the paper are not easy to read; some missing definitions and notation inconsistency further interrupt the flow.

Starting with the definition of the first-order statistics in the perturbation analysis section, the authors introduce $A_c = [a_{1,c}^l, a_{2,c}^l, \dots, a_{s,c}^l]$, where $A_c \in \mathbb{R}^{S' \times A'}$. However: (i) $A_c$ seems to be layer-specific; omitting this breaks the flow of the paper, and (ii) at the same time, $S'$ and $A'$ are never defined. Is $S'$ the set of examples corresponding to class $c$? and what is $A'$? I would assume that $A' = w \times h \times d$, but it is never defined.

In a similar vein, why are the first order statistics again $\in \mathbb{R}^{S'\times A'}$? Since a reduction takes places, the dimensionality should be different. The authors then introduce $A_{all} = [A_1;A_2;\dots ; A_1^M] \in \mathbb{R}^{D \times A'}$. I am assuming that $A_1^M$ is a typo that should be $A_M$. Again this is layer specific and misses the $l$ superscript (which appears in the $\alpha$ definition afterwards). Does this matrix comprise the first order statistics or $A_c$ themselves? The authors define $A_{all}$ as the concatentation of the $A_c$ matrices, but note that this is composed of the first order statistics. In this context, wouldn't an $\alpha$ value based only on $A_c$ make more sense for the sensitivity analysis? Different classes activate $v_j$ with different magnitudes; when an entry for one example is small and is altered with a $\xi$ that is very larger, it can easily lead to massive changes in the reconstruction. I fully understand the need for a more principled definition of $A_c$; I am not $100$% certain that this formulation captures the subtle differences in the activations between the examples. What are the values that the other works consider?

Moving on to the experimental section, for the perturbation analysis, I re-iterate my point about the magnitude of the perturbation. Even though the argument that the authors could easily hold, I am still concerned about the impact of the perturbation magnitude is the decoding process. Since the decoder it's not just a simple linear layer, a large change (compared to the original magnitudes of the vector) in the entry of vector $v_j$ can lead to misleading results.

The visualization in Fig. 2 is not clear, a description of what each color represents is important for understanding what is happening.

Further details are also necessary in the caption of Fig. 4. Without running back and forth to the relevant section, it is not clear what the different plots depict. What are the $D_0-D_9$ legends? I suppose they are the digits of the datasets.

I can't see how the Relevance Mass Accuracy metric is an appropriate  proxy for measuring the part-whole relationship. This is a metric for measuring the spatial overlap between ground truth masks and a 2D positive valued image with a single channel.  How does this relate to heatmaps arising from spatially re-arranged responses of capsules is not clear to me. It is possible that I am missing the intuition and the formulation behind this construction. I could understand the approach for activation maps for convolutional maps but not for capsules.

Figure 6 is not very clear. A more detailed caption can help clarify what each column depicts without the need to re-look at the text.

The authors note that "Overall, the mean of RMA is lower than what was anticipated". What was anticipated and how this conclusion was reached? Potentially this ties to the previous point. After the analysis, the authors themselves note that "the observed low overlap may have its origin in other sources". I personally don't find this particular analysis to be adequate to draw conclusions about the part-whole relationship of CapsNets. It may very well be true as the authors claim, but further investigation is needed.

**Questions:**

Please see the Weaknesses section.

---

> ### Author Response · Authors · 2023-11-17
> **Clarity on the Perturbation Analysis (Sec. 4.1)**
>
> Regarding the clarity of presentation of Section 4.1, We checked the section and we agree with you regarding the following; $A\prime$  where indeed refers to the flattened activation obtained from all examples of a given layer.  Where its dimension $S\prime$ refers to the number of examples corresponding to a class $c$ and $A\prime$ refers to the total number of flattened activations. This matrix represents all the activations of all the units in layer l across all the examples of class c. We agree that to preserve the flow the superindex l should be in place ($A_c^l$) in order to preserve its connection to its corresponding layer. From there we compute a matrix $\eta_c^l = [min(A^l_c) ; max(A^l_c) ; mean(A^l_c); std(A^l_c)] \in\mathbb{R}^{[4 {\times} A\prime]}$ is computed which represents the first-order statistics for the activations of every unit within layer l for class $c$. A similar approach is conducted with a larger matrix $A^l_{all}$ containing activations from examples of all classes in the dataset. It is from these $A$ that the empirical activation range ($\alpha$) is computed at the unit level. In the text of the paper have introduced an additional matrix $\eta$ to refer to the first-order statistics computed from their corresponding $A$ matrices. We are considering droping these $\eta$ matrices for clarity. In the revised version of the paper, we have addressed these notation unclarities plus other typos highlighted by the reviewer. We thank the reviewer for the meticulous revision of this part of the manuscript we believe the provided feedback improves the clarity and flow of the content.

---

> ### Author Response · Authors · 2023-11-17
> **Magnitude of the applied perturbation**
>
> We thank the reviewer for highlighting this very important aspect, we are currently conducting some experiments to further elaborate on this aspect. We hope to provide more insights on this before the end of the discussion period.

---

> > ### Author Response · Authors · 2023-11-21
> > **Magnitude of the applied perturbation**
> >
> > We thank the reviewer for the patience while while waiting for this  update. In the literature [1-4], a perturbation range $\alpha=[-0.25,025]$ is considered with steps of $0.05$. Perturbation values from this range are systematically added (as in summation) to the values present in the vector $v_j$. While widely adopted in the literature, there has not been a proper justification of why this is a method that is valid to be followed.
> >
> > Different from existing efforts which seem to define the range in a rather arbitrary manner, we do it following the method described in Sec. 4.1 which takes into account the activations of the units that define the model being analyzed. Moreover, instead of just adding to the existing values of $v_j$ the perturbation values, we replace the existing values by the estimated perturbation values. This allows us to have a more complete coverage of the empirical activation range of the units of interest and a more comprehensive analysis of their behaviour. Please note, that this coverage is not guaranteed by the standard method followed in the literature.
> >
> > In Fig. 2 of the submitted version (Fig. 10 center of the revised version) of the paper, we reported the performance of the perturbation analysis following the standard (heuristic) approach and ours. For the case of ours, we adopted a single class-agnostic range, i.e. $[-0.3,0.3]$ with steps of $0.05$, which was estimated as the mean range across that of the different classes of interest.
> >
> > In this regard, the reviewer raises the very valid point that specific units may activate on a specific range when contributing to the prediction of a given class. We agree with the reviewer on that perturbation values diverging significantly from the original activation value of a given unit may lead to misleading observations/reconstructions. To verify this, we have estimated the class-specific activation ranges (presented below). As we can notice the class-specific activation ranges across classes are well within the $[-0.3,0.3]$ range followed in our experiments.
> >
> >
> > $ \qquad \qquad \qquad MNIST  \qquad \qquad \qquad SVHN$
> >
> > $Class0 -> [-0.44, 0.40] \qquad \qquad [-0.3, 0.25]$
> >
> > $Class1 -> [-0.24, 0.3] \qquad \qquad [-0.6, 0.6]$
> >
> > $Class2 -> [-0.2, 0.3] \qquad \qquad [-0.25, 0.28$
> >
> > $Class3 -> [-0.28, 0.3] \qquad \qquad [-0.33, 0.29]$
> >
> > $Class4 -> [-0.33, 0.33] \qquad \qquad [-0.6, 0.58]$
> >
> > $Class5 -> [-0.55, 0.5] \qquad \qquad [-0.3, 0.3]$
> >
> > $Class6 -> [-0.1, 0.47] \qquad \qquad [-0.5, 0.57]$
> >
> > $Class7 -> [-0.3, 0.3] \qquad \qquad [-0.25, 0.3]$
> >
> > $Class8 -> [-0.45, 0.22] \qquad \qquad [-0.2, 0.25]$
> >
> > $Class9 -> [-0.27, 0.29] \qquad \qquad [-0.4, 0.4]$
> >
> >
> > For completeness, we have conducted an additional experiment, where we consider overarching class-agnostic ranges that cover the largest range across all classes. This range covers the entire space produced by the training and will help paint a more complete picture of plausible behaviours of the units under study. In practice, following the values presented above, this was set to $[-0.55, 0.5]$ for MNIST and $[-0.6,0.6]$ for SVHN. In order to have the same number of cells in the figure, we used the steps 0.09 and 0.1 for MNIST and SVHN, respectively. As we can see in Fig. 2, looking at the effect of the units from this larger range paints a completely different picture from that one produced by the standard (heuristic) practice.
> >
> > For completeness we have repeated our qualitative analysis considering the wider class-agnostic perturbation range (Fig. 9, top) and the class-specific ranges (Fig. 9, bottom). It is noticeable that in both settings, our observations regarding the level of disentanglement of the representation encoded in $v_j$ still hold. That is, modifying a single unit leads to the modification multiple visual features, i.e. the representation is not disentangled.
> >
> >
> >
> > References:
> > [1] Sabour. et al.,  "Dynamic routing between capsules. "NeurIPS" (2017)
> > [2] Shahroudnejad et al., "Improved explainability of capsule networks: Relevance path by agreement."ICIP" (2018)
> > [3] Choi, Jaewoong, et al. "Attention routing between capsules. "IEEE/CVF" (2019)
> > [4] Nair. et. al. "Pushing the limits of capsule networks. "arXiv"(2021)

---

> > > ### Comment · Reviewer_66ge · 2023-11-22
> > >
> > > I thank the authors for their response and particularly the analysis on the applied perturbation magnitude. Even though I still feel that there is some potential missing from this work, especially with respect to using larger datasets, I will raise my score to 6.

---

> ### Author Response · Authors · 2023-11-17
> **Clarity/Intelligibility of the presentation of some figures and tables.**
>
> Thanks for bringing the issue with the missing color bar (Fig.2/Sec. 6.1) to our attention, we already updated the color bar where the yellow indicates to higher classification accuracy. Similarly, we updated Fig.4 where $D0$ ${−}D9$ represents the classes as you stated. For clarity, we also updated the title of both Fig.4 and Fig.6.

---

> ### Author Response · Authors · 2023-11-17
> **Assessing Part-Whole Relationships via Relevance Mass Accuracy (RMA)**
>
> The description that the reviewer provided for the operation of RMA is accurate. It indeed measures the spatial coverage between a heatmap (matrix of continuous values) and a matrix of positive values. (reference) In our work, we estimate the level to which features encoded in a trained CapsNet encode part-whole relationships by measuring the spatial overlap between the responses of two units from different levels in the architecture that have a strong connection between them. The main intuition is that given the hierarchical representation that a CapsNet is expected to encode across layers, it is expected that more granular elements (parts) are encoded at earlier layers of the architecture and coarser elements (whole) are expected to be encoded at latter layers as a composition of the part elements. Therefore, spatially-speaking, features coming from a part element is expected to lie within the extent the whole element it contributes to compose. Hence, we measure the level to which this type of encoding is present by measuring the spatial coverage that responses at earlier capsule layers in the architecture have wrt. latter layers.  Following the formulation of RMA, the $2D$ positive matrix is obtained by thresholding the response of the latter layers (whole) and the continuous matrix is defined by the response of the earlier layer (parts).
> In order to obtain the $2D$ response of a given layer, we apply the methods from [Simoyan et al., 2014] which, given a unit of interest,  an input $x$ and a predicted class label $y$, generates a heatmap highlighting the spatial regions in the input $x$ which determine the prediction $y$.

---

> ### Author Response · Authors · 2023-11-17
> **Expected results from the  Relevance Mass Accuracy (RMA) analysis (Sec. 6.3)**
>
> The output of the RMA metric is a value in the range $[0,1]$ indicating the coverage between the two input matrices with $1$ indicating a high coverage. The results reported in our analysis (Table 3) indicate that, when a threshold of $0.5$ is used to produce the positive reference matrix,  the RMA score is on average $~0.04$ among the considered datasets. We notice this score increases as we use less strictive threshold values (e.g. $0.1$), however at this level the spatial extent of the candidate part unit is so coarse that hinders a local analysis. The lower layer did encoded parts of the elements encoded at deeper layers we would have expected a layer spatial coverage/overlap.

---

### Official Review · Reviewer_FL72 · 2023-10-30

**Soundness:** 2 fair
**Presentation:** 3 good
**Contribution:** 2 fair
**Rating:** 5
**Confidence:** 3

**Summary:**

The paper studies the interpretability properties of Capsule Neural Networks. It mainly focuses on the part-whole relationships encoded within the learned representations. The analysis results point out that capsule-based networks may not be related to parts-whole relationships as stated in the literature.

**Strengths:**

i) The paper conducts extensive analysis and visualization of the capsule network.

ii) The proposed permutation-based analysis and relevant unit selection are reasonable, and the analysis results support the paper's conclusion.

**Weaknesses:**

i) The experiments are mostly conducted on the small-scale dataset, such ass MINIST and SVHN, and the image resolution is also relatively small, which makes the results not convincing, and the visual difference between the baseline method and the proposed method is not obvious.

ii) The experiments are all conducted based on ConvNets. Does the conclusion hold based on a transformer-based network?

iii) The discussed related works are mostly before 2020. There have been many works about capsule networks in recent years that have not been discussed.

**Questions:**

Refer to the weakness

---

> ### Author Response · Authors · 2023-11-17
> **Experiments on ConvNets/ Transformer based models**
>
> Regarding conducting experiments based on other types of networks, while our methodology is applicable to other architectures, the aim of this paper lies on the interpretability of CapsNets-based architectures.

---

> ### Author Response · Authors · 2023-11-17
> **Input size of the considered dataset**
>
> Regarding the input size of the datasets used in our analysis, on the one hand, given our limited computational resources, we opted to conduct our analysis on a smaller input size. On the other hand, we believe that using larger input sizes could have helped us improve the level of detail of the provided qualitative results.

---

> ### Author Response · Authors · 2023-11-17
> **Experiments on the small-scale dataset**
>
> In the literature, MNIST, CIFAR10, affNIST, or SVHN are the datasets frequently used by the community for conducting research on CapsNets-based architectures. Therefore, we opted to conduct our analysis on a variety of these datasets so that our method and reported results could compared against by future methods. Beyond these standard datasets, we conducted experiments on CelebA and PASCAL datasets which are considered much complex in comparison with MNIST and CIFAR10.
>
> At this point, conducting experiments on ImageNet or similar large-scale datasets will only contribute to indicate whether the observations made in the paper remain on such big data settings. Due to the large scale of these datasets, the density of our analysis, our reduced computational resources and the limited discussion period, conducting an experiment on ImageNet might not be feasible (even partially). We will attempt to conduct an experiment on the CIFAR100 dataset, which possesses a significantly larger number of classes when compared to the datasets considered in the paper and do our best to provide results during the discussion period. We believe this should provide insights on the observations made in our work under a larger scale dataset.

---

> ### Author Response · Authors · 2023-11-17
> **Regarding related recent efforts**
>
> In [1], interpretation capabilities of CapsNets were investigated by analyzing coupling coefficients that link the connections between capsule layers. This analysis helps understand how CapsNet predicts amyotrophic lateral sclerosis (ALS) or Healthy, it reveals which primary capsules were activated when the model predicts ALS, similar to how we interpret predictions in our methodology. In addition, coupling coefficients were averaged, and the results are presented in heatmaps/histograms, highlighting the differences in capsule activation between both classes. In our case, we produce heatmaps that highlight the spatial regions in the input which determine the prediction. Similar to our work, [2] conducted a perturbation analysis of the predicted vector which was used to link learned representation to visual attributes such as thickness, orientation, etc. by reconstructing the perturbed vector using handwritten-digit images. Similar to us, this work was also conducted using a perturbation analysis ([-0.2,0.2] with steps of 0.06).  Different from our work, where a methodology is put forward for the extraction of this perturbation range, the range considered in [2] seems to have been selected in a rather arbitrary manner. [3] proposed an interpretable multimodal fusion method (IMCF) aiming at getting insights on how modalities interact and which ones are significant for predictions made by the model. Similar to our work, routing coefficients are considered to guide the interpretation process. In contrast, [3] introduces this interpretation capability by the modification of the CapsNet architecture via the integration of  BiLSTM/LSTM components. [4] proposes a CapsNet for the classification of imagined phonemes and words in EEG signals. This effort differs from ours on that it uses the learned representation of the output vectors in the class capsule layer to construct activity maps (similar to heatmaps) of brain activity based on different categories. These maps explain the most active parts of the brain during the process and they are produced based on statistical features obtained from EEG speech imagery signals. However, in our methodology, we produce gradient-based heatmaps that explain the predictions made by the model.
>
>
> [1] Predicting the prevalence of complex genetic diseases from individual genotype profiles using capsule networks. "Nature Machine Intelligence" (2023)
>
> [2] Quantum Capsule Networks. Liu, Zidu, et al. "Quantum Science and Technology" (2022)
>
> [3] Interpretable Multimodal Capsule Fusion. Wu, Jianfeng, Sijie Mai, and Haifeng Hu. "IEEE/ACM Transactions on Audio, Speech, and Language Processing" (2022)
>
> [4] Interpretation of a deep analysis of speech imagery features extracted by a capsule neural network. Macías-Macías, José M., et al. "Computers in Biology and Medicine" (2023)

---

> > ### Comment · Reviewer_FL72 · 2023-11-20
> >
> > Thanks for the author's response. After reading the author's response, the experiment results still can not convince me. So I keep my original score.

---

### Author Response · Authors · 2023-11-17

The main goal of our work is to analyze representations learned via capsule-based network architectures (CapsNets) and assess the level to which part-whole relationships are encoded therein. This is the main motivation why our experimentation is purely centered around this type of architecture and is not covering other popular architectures, e.g. ResNet, VGG, ViT and other transformer-based models.  Moreover, in line with salient work in the subject, we have conducted part of our experiments on relatively small datasets (MNIST and SVHN). Taking into account the simplicity of these datasets, we increased the level of complexity by also considering the CelebA and PASCAL-Parts datasets in our experiments.

Including larger datasets, e.g. ImageNet (ILSVRC'12), could further contribute to assess whether the observations made in our experiments remain in large scale scenarios. However, due to the large scale of this dataset, the density of our analysis, our reduced computational resources and the limited discussion period, conducting an experiment on ImageNet might not be feasible (even partially). We will attempt to conduct an experiment on the CIFAR100 dataset and do our best to provide results within the discussion period.

Below we provide more detailed response to the feedback raised by each reviewer. Finally, we would like to thank the reviewers with the provided feedback and pointers regarding the clarity of the content of the paper. We believe the provided feedback will help strengthen our paper.

---

### Meta-Review · Area_Chair_TLQ4 · 2023-12-06

**Metareview:**

The authors provide a principled approach for probing the properties of capsule networks. The authors provide qualitative and quantitative results on small-scale datasets (MNIST, SVHN, Celeb-A, ...) which are at odds with the existing findings in the related literature. In particular, the findings suggest that CapsNet representations may not be as disentangled or strictly related to part-whole relationships as commonly claimed.

This is a borderline submission. The motivation of the work is clear and the results are indeed interesting. The perturbation analysis via first-order statistics also makes sense in this context. To make the message stronger, reviewers wanted to see more results on (1) transformer based models, and (2) larger-scale datasets, (3) CapsNet related work in the last 3 years. Both (1) and (2) are important to establish whether the results generalize beyond the small datasets and beyond ConvNets. Given the strength of the message this work is communicating to the CapsNet community, I feel that this analysis is not optional, and even if the results don't hold for (1) and (2), it would still be a meaningful contribution  to the community.

During the rebuttal phase the authors did provide some additional results, but the reviewers remain unconvinced that the strong sentiment brought forward in this work generalizes beyond the setting analized. I suggest the authors carefully assess the claims in the context of (1) and (2) and improve the clarity of exposition related to the perturbation analysis and submit the results to some of the following venues.

**Justification For Why Not Higher Score:**

The message is too strong given the analysis, and a major revision of the empirical section is needed.

**Justification For Why Not Lower Score:**

N/A

---

### Decision · Program_Chairs · 2024-01-16

Reject